

# NorCPM1 and its contribution to CMIP6 DCPP

Ingo Bethke[1], Yiguo Wang[2], François Counillon[2,1], Noel Keenlyside[1,2], Madlen Kimmritz[3], Filippa Fransner[1], Annette Samuelsen[2], Helene Langehaug[2], Lea Svendsen[1], Ping-Gin Chiu[1], Leilane Passos[2,1], Mats Bentsen[4], Chuncheng Guo[4], Alok Gupta[4], Jerry Tjiputra[4], Alf Kirkevåg[5], Dirk Olivié[5], Øyvind Seland[5], Julie Solsvik Vågane[5], Yuanchao Fan[6], Tor Eldevik[1]

[1]Geophysical Institute, University of Bergen, Bjerknes Centre for Climate Research, 5007 Bergen, Norway

[2]Nansen Environmental and Remote Sensing Center and Bjerknes Centre for Climate Research, 5006 Bergen, Norway

[3]Alfred Wegener Institute for Polar and Marine Research, Bremerhaven, Germany

[4]NORCE Norwegian Research Centre, Bjerknes Centre for Climate Research, 5007 Bergen, Norway

[5]Norwegian Meteorological Institute, P.O. Box 43, Blindern, 0313 Oslo, Norway

[6]Center for the Environment, Faculty of Arts & Sciences, Harvard University, Cambridge, MA 02138, USA

*Correspondence to*: Ingo Bethke (Ingo.Bethke@uib.no)

**Abstract.** The Norwegian Climate Prediction Model version 1 (NorCPM1) is a new research tool for performing climate reanalyses and seasonal-to-decadal climate predictions. It combines the Norwegian Earth System Model version 1 (NorESM1)—which features interactive aerosol-cloud schemes and an isopycnic-coordinate ocean component with biogeochemistry—with anomaly assimilation of SST and T/S-profile observations using the Ensemble Kalman Filter (EnKF).

We first describe the Earth system component and the data assimilation (DA) scheme, highlighting implementation of new forcings, bug-fixes, re-tuning and DA innovations. Notably, NorCPM1 uses two anomaly assimilation variants to assess the impact of sea ice initialisation and climatological reference period: The first (i1) uses a 1980–2010 reference climatology for computing anomalies and the DA only updates the physical ocean state; the second (i2) uses a 1950–2010 reference climatology and additionally updates the sea ice state via strongly coupled DA of ocean observations.

We then assess the baseline, reanalysis and prediction performance with output contributed to the Decadal Climate Prediction Project (DCPP) as part of the sixth Coupled Model Intercomparison Project (CMIP6). The non-assimilation experiments exhibit a moderate historical global surface temperature evolution and tropical climate variability characteristics that compare favourably with observations. The climate biases of NorCPM1 using CMIP6 external forcings, are comparable to, or slightly larger than those of the original NorESM1 CMIP5 model, with positive biases in Atlantic meridional overturning circulation (AMOC) strength and Arctic sea ice thickness, too cold subtropical oceans and northern continents, and a too warm North Atlantic and Southern Ocean. The biases in the assimilation experiments are mostly unchanged except for a reduced sea ice thickness bias in i2 caused by the assimilation update of sea ice, generally confirming that the anomaly assimilation synchronises variability without changing the climatology. The i1 and i2 reanalysis/hindcast products overall show comparable performance. The benefits of initialisation are seen globally in the first year of the prediction over a range





of variables, also in the atmosphere and over land. External forcings are the primary source of multi-year skills, while added benefit from initialisation is demonstrated for the subpolar North Atlantic (SPNA) and its extension to the Arctic. Both products show limited success in constraining and predicting surface ocean biogeochemistry variability. However, observational uncertainties and short temporal coverage make biogeochemistry evaluation uncertain while potential predictability is found to be high. For physical climate prediction, i2 performs marginally better than i1 for a range of variables, especially in the SPNA and in the vicinity of sea ice, with notably improved sea level variability of the Southern Ocean. Despite similar skills, i1 and i2 feature very different drift behaviours, mainly due to their use of different climatologies in DA; i2 exhibits an anomalously strong AMOC that leads to forecast drift with unrealistic warming in the SPNA, whereas i1 exhibits a weaker AMOC that leads to unrealistic cooling. In polar regions, the reduction in climatological ice thickness in i2 causes additional forecast drift as the ice grows back. Posteriori lead dependent drift correction removes most hindcast differences; applications should therefore benefit from combining the two products.

The results confirm that the large-scale ocean circulation exerts strong control on North Atlantic temperature variability, implying predictive potential from better synchronisation of circulation variability. Future development will therefore focus on improving the representation of mean state and variability of AMOC and its initialisation. Other efforts will be directed to refining the anomaly assimilation scheme—to better separate between internal versus forced signals, to include land and atmosphere initialisation and new observational types—and improving biogeochemistry prediction capability. Combined with other models, NorCPM1 may already contribute to skilful multi-year climate prediction that benefits society.

# 1 Introduction

Retrospective predictions have demonstrated potential of forecasting seasonal-to-decadal climate variations. Particularly for the North Atlantic (Keenlyside et al., 2008; Yeager et al., 2017) and partly also for the North Pacific (Mochizuki et al., 2009) models show robust benefit from initializing the internal climate variability in forecasting the upper ocean state several years ahead. Prediction skill in the ocean gives rise to skill in the atmosphere and over land by affecting the atmospheric circulation or atmospheric transport of anomalous heat and moisture. The level of internal climate variability, and thus potential benefit from initialization, is especially high on regional scale, where it has numerous socioeconomic applications (Kushnir et al., 2019). Comparison of initialized retrospective predictions with the observed climate evolution not only provides forecast quality information, but also informs climate change attribution and Earth system model (ESM) evaluation. Initialized retrospective predictions were part of the Coupled Model Intercomparison Project phase 5 (CMIP5; Taylor et al., 2012) that provided input to the Intergovernmental Panel on Climate Change assessment IPCC AR5 report (Kirtman et al., 2013). They are also included in the latest CMIP6 (Eyring et al., 2016), as part of the Decadal Climate Prediction Project (DCPP; Boer et al., 2016), feeding into the upcoming IPCC AR6 report.

Current climate prediction systems are thought to not fully realise the predictive potential on multi-year times scales, although the practical limits of predictability themselves and their regional variations are poorly known. The skill of climate





prediction depends on the initialisation of internal climate variability state, the representation of the dynamics and processes that lead to predictability, and the representation of the climate responses to external forcings. Dynamical climate prediction systems typically use ESMs (initially developed to provide uninitialized long-term climate projections) for representing the dynamics and the responses to external forcings. Importantly, the dynamical prediction systems add initialisation capability to the ESMs, adopting a wide range of initialisation strategies (see Section 2.2.1). A better understanding of the three

aspects—initialisation, model dynamics, forcing responses—is fundamental for better exploiting the climate predictive potential and improving estimates of climate predictability. The existing climate prediction systems undersample effects of model and initialisation uncertainty and are not necessarily well suited to address questions related to changes in the observing system. The benefits from using advanced data assimilation for initialisation, especially in an ocean density coordinate framework, are not well explored.

The Norwegian Climate Prediction Model version 1 (NorCPM1) is a new climate prediction system with coupled initialization capability that features innovations aiming to reduce initialisation shock and forecast drift, and to rigorously account for observational uncertainties. NorCPM1 contributes to CMIP6 DCPP using two variants of an anomaly initialisation method (see Section 2.1 for details), enriching the CMIP6 DCPP repository in terms of model and initialisation diversity as well as simulation ensemble size. Specifically, it provides output from CMIP standard experiments (including a

30-member ensemble of no-assimilation *historical* simulations), two sets of initialised DCPP hindcast simulations, and two sets of DCPP coupled reanalysis simulations that also provide the initial conditions for the hindcasts. The output is suited for multi-model studies that address model and initialisation uncertainty in climate prediction or aim at combining multiple models to achieve better predictions, and for benchmarking future versions of NorCPM.

The Norwegian Earth System Model version 1 (NorESM1; Bentsen et al., 2013; Iversen et al., 2013), the backbone of

NorCPM1, has previously contributed to CMIP5 with climate projections and distinguished itself with realistic El Niño–Southern Oscillation (ENSO) variability (Lu et al., 2018) and a modest historical global warming trend that favourably compares to observations. It also includes a physical-biogeochemical ocean component with a vertical density coordinate and an atmosphere component with specialised aerosol-cloud schemes. While not included in this version, current development efforts are directed to improving the regional climate representation in the sub-Arctic and Arctic and to

exploring benefits for climate prediction from bias-reduction techniques (Toniazzo & Koseki, 2018, Counillon et al., 2021), model parameter estimation (Gharamti et al., 2017, Singh et al., in preparation), upgrades of model physics and resolution (Seland et al., 2020), improved ocean biogeochemistry (Tjiputra et al., 2020), and coupling of multiple ESMs (Shen et al., 2016).

NorCPM1 further stands out in that it uses an Ensemble Kalman Filter (EnKF; Evensen, 2003) based anomaly DA scheme

that updates unobserved variables by utilizing the state-dependent covariance information derived from the simulation ensemble, and also has a rigorous treatment of observation measurement and representation errors (see Appendix A for more information on the choice of DA scheme). To date, few climate prediction systems use assimilation schemes of similar complexity and their implementations differ significantly from the one used here (see Section 2.1.1 for details). NorCPM's





DA capability is subject to continuous development and the system serves as a tool and testbed for new science innovations
in the field of DA. Reliable ensemble prediction requires an accurate representation of uncertainty in the initial conditions
and the EnKF provides a mean to achieve this. The EnKF further allows assimilation of raw observations of various types
and controls the assimilation strength depending on observational error, their spatial coverage and evolution of the
covariance with the state of the climate. In a Monte Carlo manner, it propagates uncertainty from the previous assimilation,
providing a complete spatiotemporal uncertainty estimate. The method generates a spread in hindcast initial conditions that
reflects uncertainties in the initial conditions, which typically evolve in time and space as the observational network changes.
This makes NorCPM1 a suitable tool for assessing the impact of observation system changes on climate prediction. It also
limits artefacts due to over-assimilation of sparse and uncertain observations in the early instrumental era. By utilizing initial
conditions from a coupled reanalysis that assimilates observational anomalies into the same ESM as used in the predictions,
the system reduces initialisation shock and ensures consistency of initialisation anomalies across variables and with the
model dynamics.

NorCPM1 has been developed from a series of prototypes. In a perfect model framework, Counillon et al. (2014) tested
EnKF anomaly assimilation of synthetic SST observations into the low-resolution version of NorESM1 and found the system
to constrain well oceanic variability in the tropical Pacific and subpolar North Atlantic. The system was successively
upgraded to the medium-resolution NorESM1-ME and other features such as the use of real-world SST observations
(Counillon et al., 2016; Wang et al. 2019, Dai et al., 2020), assimilation of temperature and salinity profiles (Wang et al.,
2017) and optional assimilation of sea ice concentration observations with strongly coupled ocean-sea ice state update
(Kimmritz et al., 2018, 2019). The version described in this paper includes further upgrades of the external forcings to
comply with CMIP6, code fixes, retuning of the physics, activation of ocean biogeochemistry and modifications to the
anomaly assimilation scheme. These are detailed in Section 2.

This paper sets out to technically describe NorCPM1 and its contribution to CMIP6 DCPP and then assess the model's
fitness of purpose through a broad evaluation of its baseline climate, and climate reanalysis and prediction performance. The
paper intends to inform science studies that use the model's CMIP6 DCPP output, to provide a synthesis of past model
development and to serve as a baseline for future development. While presenting a comprehensive reference of NorCPM1,
the paper is organised in a way that makes it easy to navigate through for readers with focused interest.

The following section describes the ESM component, assimilation scheme and CMIP6 simulations performed with
NorCPM1. Section 3 evaluates the reanalysis and hindcast performance of NorCPM1. Section 4 summarizes and concludes
the paper.

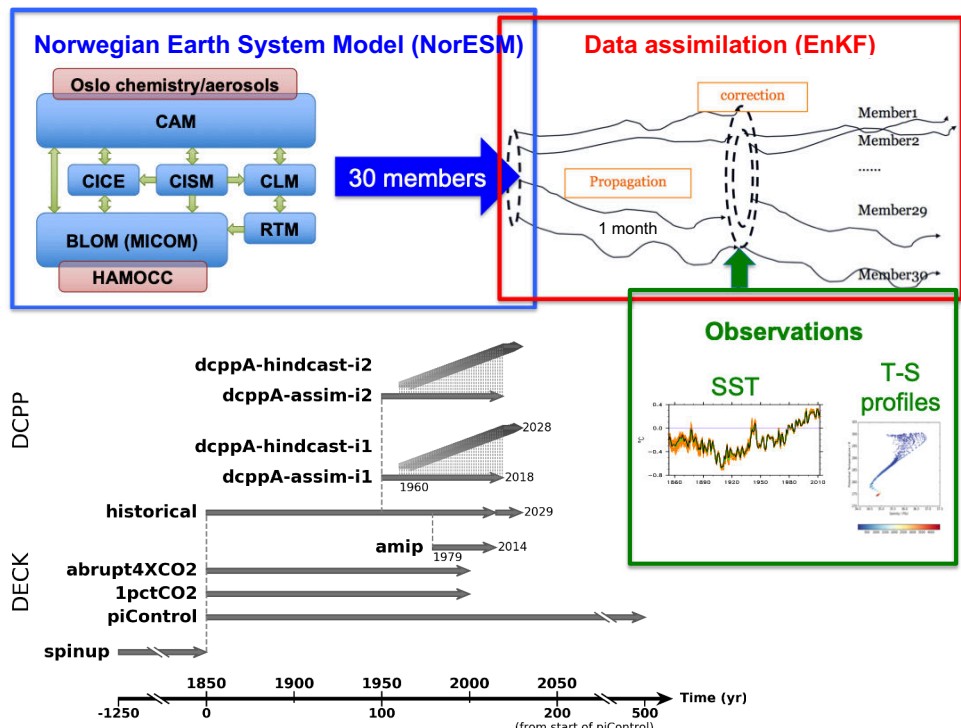

**Figure 1: Schematic of NorCPM1 and its contribution to CMIP6.**

## 2 Prediction system and simulations

This section describes the physical model, DA approach and simulations produced for CMIP6. The prediction setup and simulations are summarized in a schematic diagram in Figure 1.

### 2.1 Norwegian Earth System Model (NorESM)

The physical model used in NorCPM1 builds on the medium resolution NorESM1-ME that includes a complete carbon cycle representation, which allows the model to be run fully interactively with prescribed $CO_2$ emissions. However, we use prescribed atmospheric greenhouse gas concentrations in NorCPM. While previous NorCPM prototypes (e.g., Counillon et al., 2014, 2016) used the original CMIP5 version, NorCPM1 uses a modified version that has been subject to CMIP6 forcing updates, minor code changes and re-tuning (see Section 2.1.3). In the following subsections, we will summarize the main features of the original NorESM1-ME and then detail the differences to the version used in NorCPM1.



### 2.1.1 General description

NorESM1-ME (Bentsen et al., 2013; Tjiputra et al., 2013) is based on the Community Earth System Model (CESM1.0.4; Hurrell et al., 2013). Its atmosphere component CAM4-OSLO replaces the original prescribed aerosol formulation of the Community Atmosphere Model (CAM4; Neale et al., 2010) with a prognostic aerosol life cycle formulation using emissions and new aerosol-cloud interaction schemes (Kirkevåg et al., 2013). It also uses a different ocean component—the Bergen Layered Ocean Model (BLOM formerly NorESM-O; Bentsen et al., 2013; Danabasoglu et al., 2014)—that originates from the Miami Isopycnic Coordinate Ocean Model (MICOM; Bleck and Smith, 1990; Bleck et al., 1992). The vertical density coordinate of the ocean component minimizes spurious diapycnal mixing, improving conservation and transformation of tracers and water masses. BLOM transports biogeochemical tracers of the ocean carbon cycle component—the Hamburg Ocean Carbon Cycle model (HAMOCC; Maier-Reimer et al., 2005)—which has been coupled to the physical ocean model and optimised for the isopycnic coordinate framework (Assmann et al., 2010; Tjiputra et al., 2013). The Community Land Model (CLM4; Lawrence et al., 2011) and the Los Alamos Sea Ice Model (CICE4; Bitz et al., 2012) with five thickness categories and the elastic–viscous–plastic rheology (Hunke and Dukowicz, 1997), are adopted from CESM in their original form.

The atmosphere and land components are configured on NCAR's finite-volume 2° grid (f19), which has a regular 1.9°×2.5° latitude-longitude resolution. The atmospheric component comprises 26 hybrid sigma-pressure levels extending to 3 hPa. The ocean and sea-ice components are configured on NCAR's gx1v6 horizontal grid, which is a curvilinear grid with the northern pole singularity shifted over Greenland and a nominal resolution of 1° that is enhanced meridionally towards the equator and both zonally and meridionally towards the poles. The ocean component comprises a stack of 51 isopycnic layers, with a bulk mixed layer representation on top consisting of two layers with time-evolving thicknesses and densities.

### 2.1.2 CMIP6 forcing implementation

This section details the CMIP6 external forcing implementation into NorCPM1. Special note is made where the model setup deviates from the CMIP6 protocol. The updates of external forcing from CMIP5 to CMIP6 are expected to moderately alter the model's climate mean state, variability and anthropogenic trends. A detailed assessment of the impacts of the individual forcing upgrades is beyond the scope of this overview paper and needs to be addressed in separate studies.

The update that affects most the anthropogenic climate trend in NorCPM1 compared to the original NorESM1-ME is likely the change in anthropogenic emissions of aerosols and aerosol precursors (see Section 2.1.1 in Kirkevåg et al. 2013 for details of NorESM1-ME's CMIP5 aerosol implementation and emission datasets). We updated the emissions of $SO_2$, $SO_4$, fossil fuel and biomass burning of black carbon (BC) and organic matter (OM) to the CMIP6 preindustrial and historical forcing (Hoesly et al. 2018). We used the SSP2-4.5 scenario forcing, i.e., the "middle of the road" scenario of the SSP2 socioeconomic family, with an intermediate 4.5 W m$^{-2}$ radiative forcing level by 2100 (Gidden et al., 2019) for the post-2014 period in accordance with the DCPP protocol (Boer et al. 2016). BC emissions from aviation, omitted in the CMIP5





implementation, are now included. The representations of natural aerosol emissions of biogenic OM and secondary organic aerosol (SOA) production, dimethyl sulfide (DMS), tropospheric background $SO_2$ from volcanoes, mineral dust and sea salt are kept unchanged.

We updated prescribed atmospheric greenhouse gas concentrations (except ozone) to Meinshausen et al. (2017) for the preindustrial and historical period and to SSP2-4.5 (Gidden et al., 2019) for the post-2014 period. We applied globally uniform concentrations of the five equivalent greenhouse gas species of $CO_2$, $NH_4$, $N_2O$, CFC-11 and CFC-12. The forcing data is at annual resolution and linearly interpolated between years by the model. Due to a bug in the merging of historical and future scenario forcing, values for 2015 and 2016 were erroneously set to 2014 values while from 2017 all values

correctly follow the scenario forcing. This results in a $CO_2$ concentration error of less than 4 ppm, which has a negligible impact on the radiative forcing evolution but may impact ocean-atmosphere $CO_2$ flux prediction.

We updated prescribed atmospheric ozone concentrations to Hegglin et al. (2016) (see also Checa-Garcia et al., 2018) for the preindustrial, historical and post-2014 periods. After most simulations had been completed, we discovered that the date in our historical and post-2014 ozone input files was erroneously shifted by 23 months (e.g., the January 2000 observation is

applied in February 1998). As a result, the model anticipates anthropogenic ozone changes approximately two years too early. The 1-month shift in the seasonal cycle may have dynamical implications particularly for the stratosphere if compared against the preindustrial simulation that does not contain the shift.

We updated the solar forcing to the CMIP6 product (Matthes et al., 2017) as well as the stratospheric volcanic forcing (Revell et al., 2017; Thomason et al., 2018). In NorESM1-ME used in CMIP5, stratospheric volcanic aerosol loadings were

prescribed, and the model then computed the resulting radiative forcing assuming certain aerosol properties and particle growth. In CMIP6, pre-computed optical parameters are provided instead and prescribed directly to the radiation code of the models in order to reduce inter-model spread in responses. NorCPM1 prescribes zonally uniform space-time varying extinction coefficient, single scattering albedo and hemispheric asymmetry factor for 14 solar (i.e., shortwave covering infrared, visible and ultraviolet) and 16 terrestrial (i.e., thermal longwave) wavelength bands. Despite significant changes

between volcanic forcing implementations, we found only minor differences when comparing the radiative forcing to the Mt Pinatubo 1991 eruption, with the CMIP6 implementation producing a less distinct peak and a wider tail compared to the CMIP5 implementation (not shown). Additionally, the CMIP6 experimental protocol now requires the use of a stratospheric volcanic background forcing (monthly climatology computed from historical 1850-2000 volcanic forcing) during preindustrial and future, whereas the use of such background forcing was optional in CMIP5 and not implemented in the

original NorESM1-ME.

We updated the land surface types and transient land-use to be consistent with the Land-Use Harmonization version 2 (LUH2) dataset (Lawrence et al., 2016). For the post-2014 period, NorCPM1 deviates from the DCPP protocol as it uses land-use data from SSP3-7.0 scenario (which was the only LUH2-version land use scenario data for CLM4 available to us at that time) instead of the recommended SSP2-4.5. For CMIP6 DCPP, the main interest is in the historical period 1850-2014.

From the future scenario only the period prior 2030 is of interest for DCPP decadal outlooks, during which time the





differences between the SSP scenarios are still small. We expect this deviation to have a minimal impact on the outcomes of NorCPM1's near-future climate outlooks (note that the greenhouse gases concentrations still follow the SSP2-4.5 scenario). Data users who specifically investigate near-future land-use related climate feedbacks are, however, advised to either exclude NorCPM1 from their analysis or take the land-use differences between SSP2-4.5 and SSP3-7.0 into consideration. A

supporting simulation experiment revealed that the update to LUH2 caused an unrealistic land-cryosphere cooling trend over the historical period in NorCPM1 (Fig. C3, C4 and text in Appendix C). The cause and ramifications are subject to further investigation.

Other forcing not mentioned above (e.g., nitrogen deposition) are kept the same as in the CMIP5 model setup.

### 2.1.3 Code changes, retuning and equilibration

This section describes code changes unrelated to forcing upgrades, and re-tuning of NorCPM1 relative to NorESM1-ME that was necessary due to forcing and code changes.

An error in the aerosol code that caused an overestimation of the BC load was identified in NorESM1-ME and a correction has been proposed (details in Graff et al., 2019). The correction of this error is applied in NorCPM1 and causes a slight cooling of the climate with a -0.5 °C difference in the Arctic (Fig. C4).

NorESM1-ME featured too thick sea ice on the shelf seas of the eastern Eurasian Arctic due to spurious variability in ocean velocities enhancing ice formation in the region (Seland and Debernard, 2014; Graff et al., 2019). Increasing the built-in velocity damping applied to shallow ocean regions in MICOM reduces the regional thickness bias in NorCPM1.

NorESM1-ME's ocean biogeochemistry output has been subject to substantial grid noise. The noise was traced back to a local tracer mass correction that was applied because surface freshwater fluxes do not change the ocean column mass in the

model. For instance, a positive surface freshwater flux into the ocean – assuming tracer concentrations of this flux to be zero – will reduce the ocean tracer concentrations. Without a compensating increase in column water mass such a reduction in concentrations inevitably leads to a reduction (i.e., non-conservation) in column-integrated tracer mass. The correction in NorESM1-ME locally scales the tracer concentrations such that the column-integrated tracer mass is conserved for each grid cell. This correction scheme has the weakness that it produces considerable spatial noise at the surface and artificial temporal

variability and trends in the deep ocean. These problems are mitigated in NorCPM1 by replacing the local scaling with a global scaling (i.e., the same correction scale factor is used for all grid cells) that enforces global instead of local tracer conservation.

Using the original parameter settings of NorESM1-ME, the surface climate of the physical component of NorCPM1 drifts towards an unrealistic cold state with exacerbated biases as a consequence of introducing stratospheric background volcanic

forcing, changing the land surface boundary conditions and correcting the bug in the aerosol code. To avoid a deterioration of climate performance and to re-equilibrate the climate we therefore re-tuned NorCPM1 relative to NorESM1-ME. Specifically, we increased the condensation threshold for low clouds (from 90.05 % to 90.08 %) and also decreased the snow



albedo over sea ice by adjusting parameters that affect snow metamorphosis (from r_snw=0, dt_mlt_in=1.5, rsnw_mlt_in=1500 to r_snw=-2, dt_mlt_in=2.0, rsnw_mlt_in=2000).

After the re-tuning, NorCPM1 neither shows obvious climate improvements nor global scale deterioration compared to NorESM1-ME, though some regional differences exist (see Appendix C). Since the model characteristics did not substantially change, we performed only a short preindustrial spin-up of 250 years for NorCPM1—using the year-1000 state of NorESM1-ME's spin-up (corresponding to the year 100 state of its CMIP5 preindustrial control simulation) as initial conditions—in order to allow the upper ocean, sea ice and land surface to equilibrate to the model code and forcing changes.

**2.2 Data assimilation (DA)**

The decadal hindcasts are initialised from two coupled reanalyses of NorCPM1 in which monthly anomalies of sea surface temperature (SST) and of hydrographic profiles are assimilated into NorESM using anomaly EnKF DA over the period 1950–2018. The same ESM is used for generating the reanalysis and performing the decadal hindcasts, limiting adjustments that occur after the model system is initialized. The following subsections will present the assimilated data, the DA method,

its general implementation and the treatment of ocean biogeochemistry during assimilation. A rationale behind the choice of the DA method is presented in Appendix A.

**2.2.1 Assimilated data**

For the period 1950-2010, SST data are taken from the HadISST2 dataset of the Met Office Hadley Centre (HadISST2.1.0.0; Kennedy et al., personal communication; Rayner et al., personal communication) that has also been utilized in the

construction of the coupled reanalysis CERA-20C (Laloyaux et al., 2018). HadISST2 provides 10 realisations of monthly gridded SST over 1850-2010 with a 1° resolution. The spread between the realisations, which depends on time and space, is designed to reflect uncertainties in gridding and combining SST in-situ observations, retrievals from AATSR (Advanced Along-Track Scanning Radiometer) reprocessing and AVHRR (Advanced Very High Resolution Radiometer) retrievals. We consider the average and variance of these 10 realisations as the observations and their error variance. We use monthly SST

data from the National Oceanic and Atmospheric Administration (NOAA) Optimum Interpolation SST version 2 (OISSTV2; Reynolds et al., 2002) for the period 2011-2018, when HadISST2 data are not available. OISSTV2 provides weekly SST and weekly observation error variance, in addition to monthly SST. The observation error variance of the monthly data is estimated as the harmonic mean of weekly error variances provided by OISSTV2. We have confirmed through a separate reanalysis and set of hindcasts overlapping between 2006 and 2010 that the transition from HadISST2 to OISSTV2 does not

cause discontinuities nor significant change of prediction skill (not shown). SST data in the regions covered by sea ice are not assimilated; these regions are identified using the sea ice mask in HadISST2 or OISSTV2.

Subsurface ocean temperature and salinity (hydrographic) profile observations are taken from the EN4 dataset (EN4.2.1; Good et al., 2013). The EN4 dataset consists of profile data from all types of ocean profiling instruments, including from the World Ocean Database, the Arctic Synoptic Basin Wide Oceanography project, the Global Temperature and Salinity Profile





Program, and Argo. The EN4 profile data are available from 1900 to the present, including data quality information and bias corrections (Gouretski and Reseghetti, 2010). Data that lie within the mixed layer of NorCPM's first ensemble member are not assimilated, in order to maximize the impact of SST assimilation in the mixed layer. The uncertainty of observed hydrographic profiles is not available and we have used the estimate provided by Levitus et al. (1994a, 1994b) and Stammer et al. (2002).

### 2.2.2 DA method

The EnKF (Evensen, 2003) is an advanced, ensemble-based and recursive DA method. One advantage of the EnKF is its probabilistic nature that provides model uncertainty quantification through Monte-Carlo ensembles (Fig. 1; red box). Moreover, the EnKF provides multivariate and flow-dependent updates, meaning that information is propagated from the observed variables to the unobserved variables dependent on the evolving state of the climate system; this is crucial to

capture shifts in regimes (Counillon et al., 2016). In order to work efficiently, the EnKF needs an ensemble size sufficiently large to span the model subspace dimension (Natvik and Evensen 2003; Sakov and Oke 2008). Localisation reduces the spatial domain of influence of observation which reduces drastically the need of a large ensemble size. With the recent improvements of high-performance computing, the use of the EnKF for seasonal to decadal climate prediction has emerged (Zhang et al., 2007; Karspeck et al., 2013; Counillon et al., 2014; Brune et al., 2015, Sandery et al., 2020). Because

NorCPM1 performs monthly assimilation updates, the numerical cost for performing the updates is small compared to the cost of integrating the model.

NorCPM1 uses a deterministic variant of the EnKF (DEnKF; Sakov and Oke, 2008). The DEnKF updates the ensemble perturbations around the updated ensemble mean using an expansion of the expected correction to the forecast. This yields an approximate but deterministic form of the traditional stochastic EnKF that outperforms the latter, particularly for small

ensembles (Sakov and Oke, 2008).

### 2.2.3 DA implementation

In order to generate the coupled reanalysis, we assimilate in the middle of the month all observations available during that month and update the instantaneous model state. Assimilation of monthly SST data implies that the innovation (i.e., observations minus model state) compares variability of an instantaneous model snapshot with that of monthly averaged

observations. An alternative has been investigated, where data has been assimilated at the end of the month comparing the monthly averaged model output with the SST data. However, the latter approach shows poorer performance for reanalysis and no improvements during prediction (Billeau et al., 2016). This suggests that comparing model snapshots with monthly data is not a critical approximation for our system.

We perform anomaly assimilation in which the climatology of the observations is replaced by the model climatology.

Considering the impact of the choice of the climatology reference period on the performance of reanalysis, NorCPM1 contributes two coupled reanalysis products to CMIP6 DCPP, labelled *assim-i1* and *assim-i2* (see Fig. 1; Section 2.3 for





experiment overview). In *assim-i1*, the climatology is defined over the reference period 1980-2010 when assimilating EN4.2.1 hydrographic profile data and HadISST2 data, but over the period 1982-2010 when assimilating OISSTV2 data (i.e., beyond 2010). The model climatology is calculated from the ensemble mean of NorCPM1's 30-member no-assimilation

historical experiment (Section 2.3). The observed climatology for assimilating hydrographic profile data is computed from EN4 objective analysis (Good et al., 2013). In *assim-i2*, the climatology reference period is 1950-2010. For the hydrographic profile and HadISST2 data, the climatology is computed for the longer reference period. However, the climatology for the OISSTV2 data (i.e, after 2010) is calculated from a concatenated database of HadISST2 for 1950-1981 (when OISSTV2 is not available) and OISSTV2 for 1982-2010.

Together with changing the climatology reference period, we test two versions of the DA system. Time and resources constraints prevented us from testing these two aspects separately. In *assim-i1*, we only update the ocean state based on oceanic observations. In this case the system belongs to the category of weakly coupled DA system (WCDA; Penny et al., 2017), where the update in the ocean component of the system only influences the other components during model integration. In *assim-i2*, we allow the oceanic observations to update the ocean and the sea ice components. In this case the

system is a strongly coupled DA system (SCDA; Penny et al., 2017), where the oceanic observations influence the sea ice component of the system both at the DA step and during the model integration. The approach assures a more consistent initialisation across components and exploits the longer temporal coverage of oceanic observations relative to sea ice observations (see also Appendix A). To update the sea ice state, we follow Kimmritz et al. (2018), where an optimal way to update the sea ice state was identified: the EnKF updates the sea ice concentrations of the individual thickness categories,

while the other sea ice state variables (volume per thickness category, top surface temperature, snow and energy of melting) are post-processed to ensure physical consistency and maximize the benefit of the updates in the sea ice concentrations. In particular, the volume of the individual sea ice category is scaled proportionally to the updated individual concentration so that the prior individual category thickness is preserved. This approach ensures that the individual thickness values remain in its prescribed range, but still allows a large reduction of total ice thickness error (Kimmritz et al., 2018).

The DA scheme updates all ocean physical state variables. In an isopycnal coordinate ocean model, the layer thickness (a time-varying ocean state variable) is by definition always strictly positive. Due to normality assumptions the linear analysis update of the EnKF may return unphysical (negative) values. To solve this issue, we use the aggregation method proposed by Wang et al. (2016), in which we iteratively aggregate layers in the vertical until no unphysical value is returned by the EnKF. This scheme does not significantly increase the computational cost of DA, but avoids the drift in heat content, salt

content and mass that would otherwise be caused.

The reanalysis system uses 30 ensemble members. The ensemble size is relatively small compared to the dimension of the system. In order to limit spurious correlation caused by sampling error, we use localization (Houtekamer and Mitchell, 1998). We use the local analysis framework (Evensen, 2003) in which DA is performed for each horizontal grid cell and that uses only observations around the targeted grid cell to limit spurious correlation as ocean covariance decays with distance.

This also reduces the dimension of the problem. In order to avoid discontinuity in the increment at the edge of the local





domain, we use the reciprocal of the Gaspari and Cohn function (a function of the distance between observation location and the target model grid; Gaspari and Cohn, 1999) to taper observation error variance (i.e., to reduce the influence of observations). We taper innovation and ensemble perturbations with the square root of the Gaspari and Cohn function, which is equivalent to the tapering of observation error variance. The localization radius used in NorCPM1 is a bimodal Gaussian

function of latitude with a local minimum of 1500 km at the equator where covariances become anisotropic, a maximum of 2300 km in the mid-latitudes, and another minimum in the high latitudes where the Rossby radius is small (Wang et al., 2017).

Observation errors are assumed to be uncorrelated. For the SST product, this assumption clearly fails because the SST data is the result of an analysis. We have therefore decided to only assimilate the nearest SST data. For the observed hydrographic

profile, the independence of observation errors is more plausible. The observation error for the profile is considered to be the sum of the instrumental error (defined as in Levitus et al., 1994a, 1994b and Stammer et al., 2002) and the representativity error accounting for the model unresolved processes and scales. As detailed in Wang et al. (2017), the representativity error is estimated offline from the innovation and the ensemble spread of the 30-member historical experiment, to ensure that the reliability of the ensemble is preserved (i.e., the truth and the ensemble members can be considered to be drawn from the

same underlying probability distribution function). The profile observation error is inflated by a factor of three in sea ice covered regions where the observation climatology critical for anomaly assimilation is highly uncertain because of the lack of observations. When there are several observations falling within the same grid cell, these observations are "superobed": all observations falling within the same grid cell are averaged and the instrumental error variance is reduced as the harmonic sum of the individual instrumental error variances (Sakov et al., 2012). Note that the representativity error term mainly

relates to the capability of the model to represent the truth and is thus not reduced by the superobed technique.

As further detailed in Section 2.3, the initial ensemble used at the start of the reanalyses (year 1950) is branched from a 30-member historical experiment. The historical experiment was initialized in 1850 from the end of a pre-industrial spinup simulation (Section 2.1.3), with initial ensemble spread being generated by adding small random noise $O(10^{-10}$ K) to the ocean temperatures, and then integrated for 100 years allowing the spread to grow. This approach ensures that the initial

ensemble spans sufficient spread in the interior of the ocean needed for a well calibrated EnKF and that each member is synchronised with respect to the timing of the external forcing. To avoid an abrupt start of the assimilation, the observation error variance is inflated by a factor of eight during the first assimilation update; every two assimilation updates, the factor is decreased by one until it reaches one, as suggested by Sakov et al. (2012). The ensemble spread is sustained in the course of the reanalysis using the following inflation techniques. The DEnKF (Section 2.2.2) limits the need for inflation to some

extent. We use the moderation technique of Sakov et al. (2012)—while the ensemble mean is updated with the observation error variance, the ensemble spread is updated with the observation error variance by a factor of four. We also use pre-screening of the observation; i.e. the observation error variance is inflated so that the analysis remains within two standard deviations of the forecast error from the ensemble mean of the forecasts.





### 2.2.4 Treatment of ocean biogeochemistry

Fransner et al. (2020) showed with perfect model predictions using NorESM1-ME that the initial state of the biogeochemical tracers has a negligible impact on the predictability of ocean biogeochemistry beyond lead year one. During the assimilation process, the thickness of the isopycnal layers changes while the tracer concentrations on the layers remain unchanged, meaning that we allow assimilation to change the mass at every location. However, this does not introduce a drift as long as the analysis is unbiased (i.e., the assimilation does not systematically pull the model climate in one direction). This was

verified with a ten-year long twin-experiment where SST from a preindustrial control run was assimilated every month into a run with 30 members. The total change in the biogeochemical tracer mass over this period was negligible; the largest drift was found for silicate that corresponded to 0.5 % of its global mass. With this approach the global near-surface primary production approached that of the control run, showing that there is a good potential for constraining biogeochemical variability by assimilating SST only in our model setup. This might be improved by the additional assimilation of sea ice and

temperature and salinity profiles. Similarly, there are several studies that have shown that assimilation of ocean physics improves the representation of ocean biogeochemistry (e.g., Seferian et al., 2014; Li et al., 2016).

### 2.3 CMIP6 simulations

Figure 1 provides a schematic overview of NorCPM1's simulations prepared for CMIP6, including their temporal coverage

and initialisation relations. We will base our model verification and evaluations on these simulations. They can be summarized in four groups.

The Diagnostic, Evaluation and Characterization of Klima (DECK) baseline experiments comprise a coupled control experiment with fixed pre-industrial forcings (*piControl*), an idealised 1% per year $CO_2$ increase experiment (*1pctCO2*), an abrupt four times $CO_2$ experiment (*abrupt4XCO2*) and a forced atmosphere experiment with prescribed observed evolutions

of SST and sea ice (*amip*). NorCPM1's *piControl* features three realisations to better allow time-evolving assessment of model drift. The second and third realisations start from the same initial conditions as the first realisation (taken from the end of a long spin-up), but with small random noise $O(10^{-10}$ K) added to the atmospheric temperature field. *amip* features ten realisations (matching the ensemble size of the decadal hindcasts) with slightly perturbed atmospheric initial states. *1pctCO2* and *abrupt4XCO2* feature one realisation each.

The *historical* experiment features 30 realisations that are used for initialising NorCPM1's assimilation experiments, for constructing the climate anomalies of the assimilation experiments and also serve as a benchmark for the initialised hindcasts. The simulations are initialized from the same restart from *piControl*, with ensemble spread generated by adding small perturbations to the mixed layer temperatures (details in Section 2.2.3). In that way, we avoid contaminating influence of model drift on the ensemble spread that would occur if the restart conditions of piControl were sampled. *historical-ext*

extends the historical simulations from 2015 to 2029 using SSP2-4.5 scenario forcing (Section 2.1.2) to cover the time


period of the hindcast and future outlook experiments. Hereafter, *historical* refers to the combined *historical* and *historical-ext* experiment.

The DCPP simulations comprise two sets of assimilation simulations (*dcppA-assim*), hereafter referred to as *assim-i1* and *assim-i2*, with 30 ensemble members per set. The simulations are initialised from the January 1, 1950 states of *historical* and

integrated until January 15, 2019.

The DCPP simulations further comprise two sets of decadal hindcast simulations (*dcppA-hindcast*), hereafter referred to as *hindcast-i1* and *hindcast-i2*, that each feature 10 ensemble members per start date, with one start date per year from 1960 to 2018. The October 15 states of the first 10 members of *assim-i1* and *assim-i2* are used to initialise corresponding members of *hindcast-i1* and *hindcast-i2*. However, we will in the following refer to November 1 as the initialisation day, because the

assimilation update on October 15 uses observations from the entire October month. The hindcast simulations are integrated for a total of 123 months to cover 10 complete calendar years.

## 3 Verification and evaluation

In this section, we evaluate NorCPM1's reanalysis performance (Section 3.1) and hindcast performance (Section 3.2) based on the CMIP6 output. We measure skill and skill differences with anomaly correlation coefficients (ACCs) and anomaly

correlation coefficient differences (ΔACCs) (for details see Appendix B). Additional evaluation of the ESM, focusing on its climatology and variability characteristics, is presented in Appendix C.

### 3.1 Reanalysis performance

We evaluate the performance of the *assim-i1* and *assim-i2* reanalyses that span the period 1950–2018 and provide the initial conditions for the decadal hindcast experiments *hindcast-i1* and *hindcast-i2*. The following subsections cover global

assimilation statistics, impact of assimilation on the model mean states and synchronization of variability for the different components of the climate system.

### 3.1.1 Global assimilation statistics

We use the innovation to monitor the performance of assimilation over time (Sakov et al., 2012; Counillon et al., 2016), which is defined as the ensemble mean of the model forecast state (at assimilation time on the observational grid) minus the

observation. In combination with the ensemble spread and the observation error standard deviation, it can be used to assess the reliability of the ensemble system (Sakov et al., 2012). Ideally, the reliability is checked for each grid cell. Under an ergodicity assumption, we define global statistics based on innovation as follows,

$$w_i = \frac{a_i}{\sum_j a_j}, \tag{2}$$

$$\bar{d} = \sum_i w_i d_i, \tag{3}$$





$$\hat{d} = \sqrt{\sum_i w_i d_i^2}, \tag{4}$$

$$\overline{\sigma^f} = \sum_i w_i d_i^f, \tag{5}$$

$$\overline{\sigma^o} = \sum_i w_i d_i^f, \tag{6}$$

$$\overline{\sigma^t} = \sqrt{\overline{\sigma^f}^2 + \overline{\sigma^o}^2}, \tag{7}$$

where $a_i$ is the area of the model grid cell $i$, where the gridded observation is located, $w_i$ is the area-weight, $d_i$ is the

innovation, $\sigma_i^f$ is the ensemble spread (standard deviation) of forecasts, $\sigma_i^o$ is the standard deviation of observation error at

the grid cell $i$ at a given time. The observations are binned onto the model grid and into 42 depth bins that are also used to

bin the model data. In a perfectly reliable system, the RMSE $\hat{d}$ matches $\overline{\sigma^t}$, i.e., the forecast ensemble spread combined with

the observational error. Figure 2 shows the time evolutions of the innovation statistics for SST, ocean temperature and

salinity in *assim-i1* (the evolutions in *assim-i2* are similar to those in *assim-i1* and therefore not shown).

For SST (Fig. 2a), $\hat{d}$ is stable with an accuracy of approximately 0.5 K. The bias $\bar{d}$ is stable as well, fluctuating around zero.

This is expected as we use anomaly assimilation (with the bias estimated from the *historical* experiment that does not use

assimilation). It also indicates that the assimilation with a monthly cycle largely eliminates the conditional bias, caused by

model error in the sensitivity to the forcing, and thus corrects the forced long-term trends. The ensemble spread

$\overline{\sigma^f}$ is also relatively stable. There is a drop in observation error standard deviation $\overline{\sigma^o}$ in 1982 with the emergence of satellite

measurements and in 2011 with the transition from HadISST2 to OISSTV2 (see Section 2.2.2). The reliability of the system

is good until 1982 (compare blue and magenta curves), but then $\overline{\sigma^t}$ drops slightly below $\hat{d}$ indicating that the introduction of

satellite data excessively reduces the observational error estimates applied during assimilation. When the observation error

reduces, the accuracy of our model did not increase accordingly, most likely because it fails to represent the features seen in

the observations. Adding a representativity error during the satellite era to improve the reliability should be explored in

future development.

For ocean temperature (Fig. 2b), the RMSE $\hat{d}$ decreases over time from 1.5 K to 1.2 K. The bias $\bar{d}$ is positive prior 1970, but

near zero afterwards. The distribution of the observations prior to 1970 is considerably uneven with a predominance in the

North Atlantic region and the bias $\bar{d}$ does not reflect the globally averaged bias. The total error standard deviation $\overline{\sigma^t}$ is

smaller than the RMSE, suggesting that the ensemble system overestimates its accuracy (i.e., the ensemble spread is too

small). For ocean salinity (Fig. 2c), the RMSE $\hat{d}$ is stable prior 2000 and after 2005. The decrease in the RMSE $\hat{d}$ in the

period 2000-2005 is due to the introduction of ARGO floats. There is a negative bias $\bar{d}$ in salinity prior 2000. The bias $\bar{d}$

remains negative but is relatively small after 2000. As for ocean temperature, there is a mismatch between the RMSE $\hat{d}$ and

total error standard deviation $\overline{\sigma^t}$ indicating that the system is overconfident.

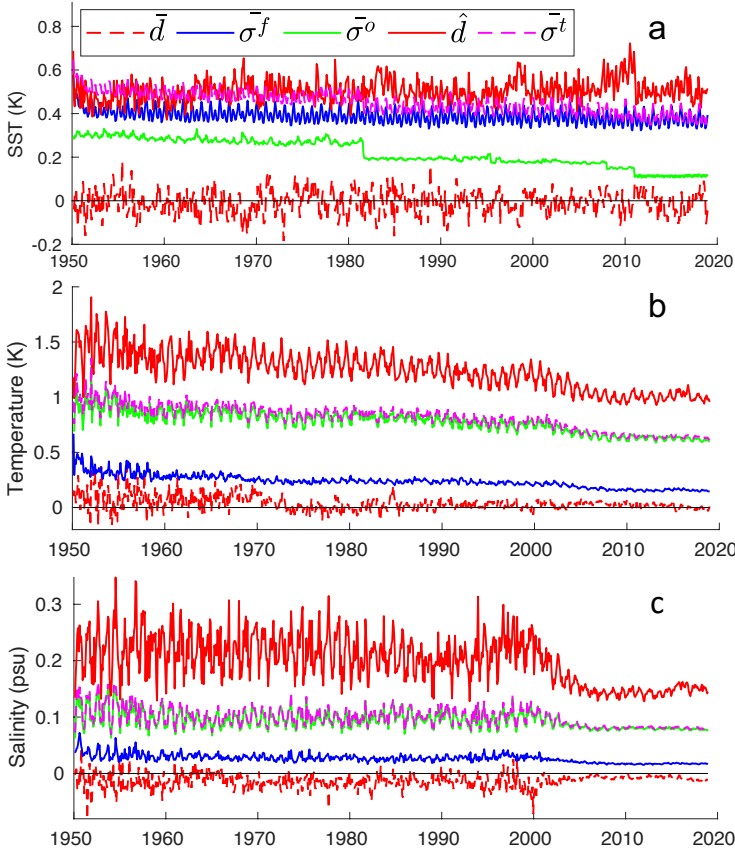

**Figure 2: Assimilation statistics. Bias $\overline{d}$ (red dashed lines), ensemble spread $\overline{\sigma^f}$ blue lines), observation error $\overline{\sigma^o}$ (green lines), RMSE $\hat{d}$ (red solid lines) and the total error $\overline{\sigma^t}$ (pink lines) for SST (a), ocean temperature (b) and ocean salinity (c).**

### 3.1.2 Effect of assimilation on mean state

Anomaly assimilation should by design have a negligible effect on the climate mean state. Nonlinear propagation of the assimilation updates between the assimilation updates can, however, yield a post-assimilation change in the mean state in regions where there are no observations Furthermore, *assim-i1* and *assim-i2* are not using the same reference period (1980-2010 versus 1950-2010) and thus differences in the mean state can occur as a consequence of different sampling of internal multidecadal climate variability in the observations and due to errors in the model's forced climate trend. Additionally, in the computation of observational profile anomalies we subtracted the climatology of the objective EN4 analysis, which is inaccurate in regions with sparse data coverage. This can further impact mean states of the reanalyses.

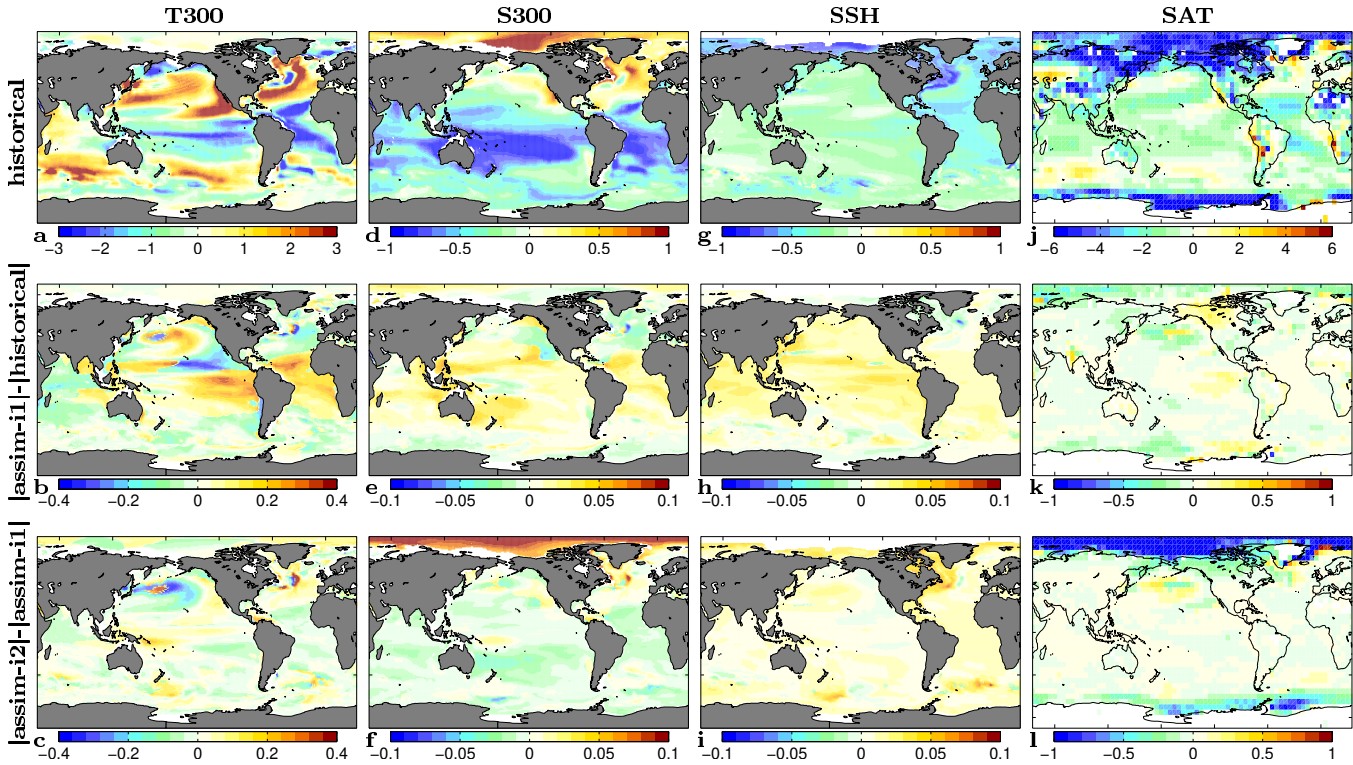

**Figure 3: Annual-mean climatological biases for T300 (a–c), S300 (d–f), SSH (g–i) and SAT (j–k). Biases of *historical* (top row), differences between absolute biases in *assim-i1* and *historical* (middle row), differences between absolute biases in *assim-i2* and *assim-i1* (bottom row). Cold colours imply bias improvement. The EN4.2.1 objective analysis (Good et al., 2013) is used to estimate the biases of T300 and S300 over 1950–2018. The Global ARMOR-3D L4 Reprocessed dataset (Larnicol et al., 2006) is used to estimate the biases of SSH over 1993–2018. HadCRUT4 (Morice et al., 2012) is used to estimate the biases of SAT over 1950–2018.**

We verify the effect of DA on the climatology by comparing mean state biases of our two assimilation products with those of the *historical* experiment (Fig. 3). The mean state changes in upper ocean temperature (T300) and salinity (S300) averaged over the top 300 m, sea surface height (SSH) and surface air temperature (SAT) due to assimilation are generally an order of magnitude smaller than the absolute biases of *historical*. The relative impact of DA on the biases is thus mostly below 10 % of its absolute magnitude. An exception is the Arctic, where the assimilation of *assim-i2* increases the S300 bias and decreases the SAT bias. This is consistent with that the assimilation of *assim-i2* tends to remove sea ice mass, which leads to higher SAT because of the thinner ice and higher surface salinity because the model tries to grow back sea ice, ejecting salt during that process. Despite assimilating climate anomalies, the sea ice update in *assim-i2* largely reduces the climatological sea ice thicknesses towards more realistic values whereas the climatology of *assim-i1* remains unchanged (Fig. 4). In a similar NorCPM version with climatological too thick Arctic sea ice, Kimmritz et al. (2019) found anomaly assimilation of observed sea ice concentration (updating the area in different thickness categories of the model using SCDA)





to yield large reductions in total ice thickness error. Here we show that similar bias reduction is achieved by strongly coupled update of the sea ice states using ocean observations. The exact reason for this behaviour is subject to further investigation.

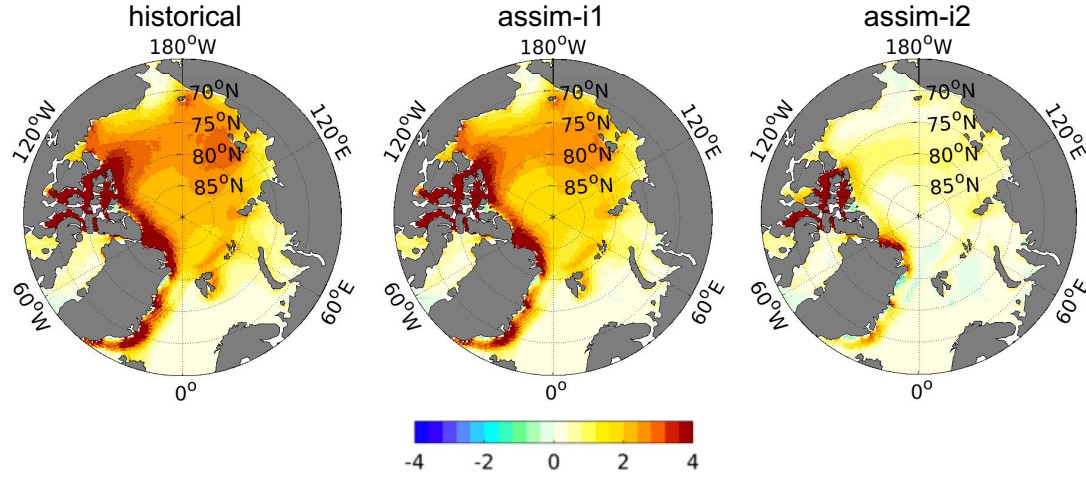


**Figure 4: November-March climatological biases of sea ice thickness (SIT) in *historical* (left), *assim-i1* (middle) and *assim-i2* (right). The observational reference combines C2SMOS (Ricker et al., 2018), Cryosat2 (Hendricks et al., 2018a) and Envisat (Hendricks et al., 2018b) over the period 2002–2018.**

The effect the assimilation has on the mean state of nutrients was assessed by investigating the difference between the
ensemble means of *historical* and *assim-i1* (Fig. 5). From previous studies (While et al., 2010; Park et al., 2018) we know that the equatorial regions are the most susceptible to errors originating from assimilation of physical variables. However, since sea ice, an efficient blocker of sunlight, is updated by weakly coupled DA, some differences in the polar region are also expected. There is indeed an increase in primary production in the polar regions in the respective summers of each hemisphere. On average there is an increase in nutrients in the Arctic, indicating that part of the increase in productivity is
caused by an increase in mixing as the ocean is exposed to the atmosphere. There are very small differences in the mean nutrient in the Southern Ocean.



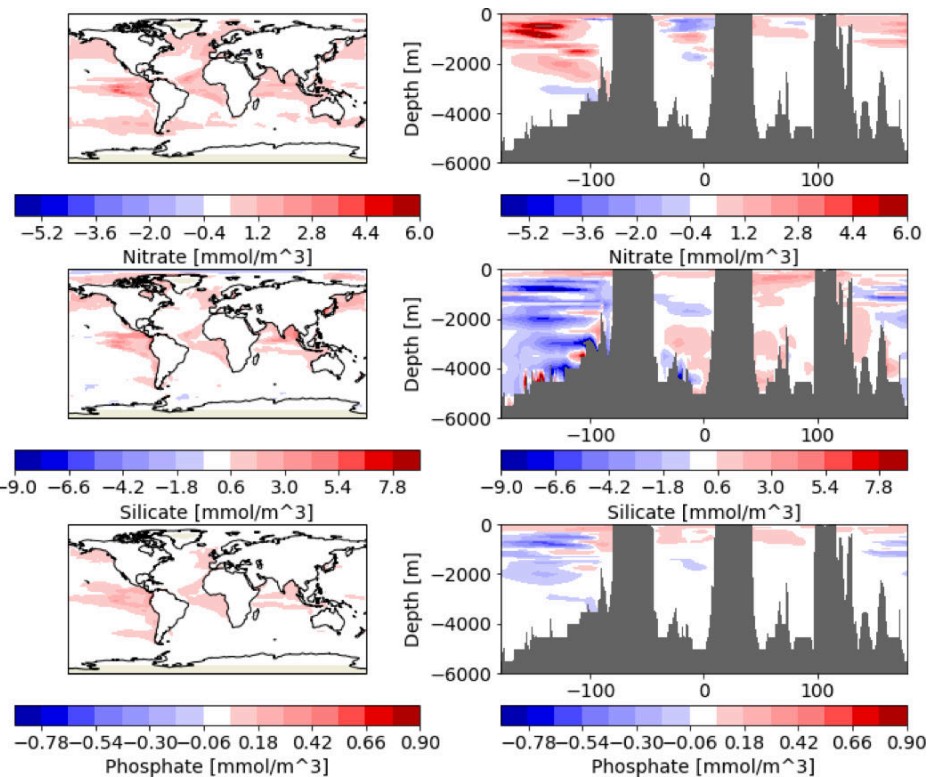

**Figure 5: Difference between the three nutrients nitrate, silicate and phosphate between *assim-i1* and *historical*. Positive values**
**means that the assimilation run has increased values. Left column shows the difference at 100 meters depth and the right column**
**shows the difference at a section along the equator. The plots are based on the mean from the period 1950–2018.**

Some changes are also seen in the surface waters of the tropical oceans, these changes do not have a pronounced seasonal variation. The largest changes to the surface nitrate and phosphate occurred in the eastern Pacific, while for silicate there was also an increase in the concentration in the Bay of Bengal. The increase in silicate in the Bay of Bengal occurs throughout
the water column, there is also a similar increase in the water column of the Western Tropical Pacific. For nitrate and phosphate, the increase in concentration is confined to the upper 500–1000 m. At the surface and down to about 1000 m all three nutrients have increased concentrations along the equator. Below 1000 m in the eastern equatorial Pacific nitrate has increased concentration while silicate and phosphate have decreased concentrations compared to the historical run. An increase in nitrate with a simultaneous decrease in silicate indicates that there is some movement in the water masses that
leads to decreased silicate and phosphate and at the same time an increase in oxygen in *assim-i1* (Fig. 6); this reduces the denitrification that occurs below the thermocline in the tropical Pacific. Furthermore, we compared the magnitude of the computed differences between *assim-i1* and the historical run along the equator and the variability of the ensemble of the historical run. The changes are always within one standard deviation of the ensemble variability—i.e., small relative to the



internal variability—except for oxygen in a small region at around 2000 m in the equatorial Atlantic where there is a large

increase in oxygen. We therefore conclude that the changes to nutrients in the *assim-i1* run are caused by changes to circulation and temperature and not by unphysical mixing caused by the assimilation.

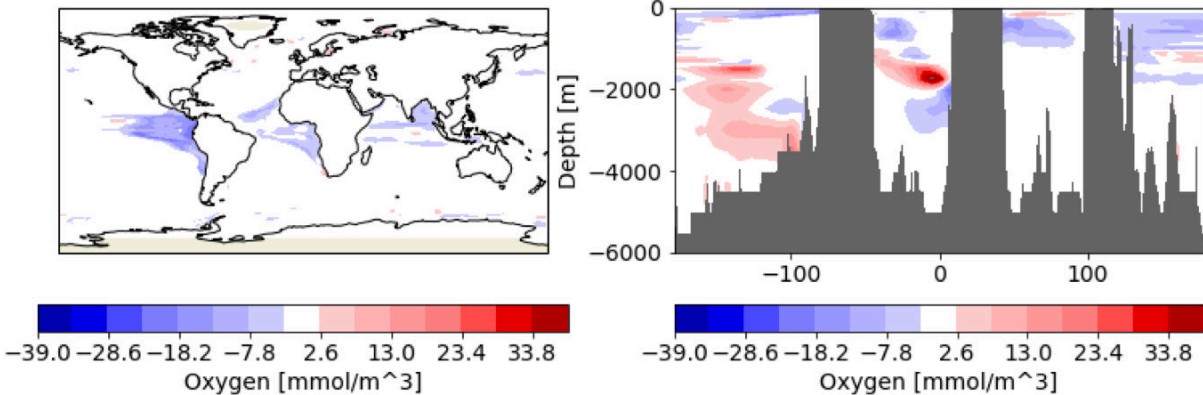

**Figure 6: The figure shows the oxygen difference between *assim-i1* and *historical*. Positive values means that the assimilation run has increased values. Left column shows the difference at 100 meters depth and the right column shows the difference at a section**

**along the equator. The plots are based on the mean from the period 1950–2018.**

### 3.1.3 Physical ocean variability

We first evaluate the synchronisation of physical ocean variability globally at grid scale interpolated to 5°x5°. Figure 7 shows ACCs for annual SST, T300, S300 and SSH for *assim-i1* along with ΔACCs for *assim-i1 - historical* and *assim-i2 - assim-i1*. The ACCs for *assim-i1* are high and statistically significant across variables in most regions. The ΔACCs for

*assim-i1 - historical* show that the assimilation of ocean data significantly improves the synchronization of SST, T300 and S300 with observations in most regions. Significant improvements for T300 are located in the Pacific and North Atlantic. The improvements for S300 are smaller than those for T300, likely because there are considerably fewer subsurface salinity observations than subsurface temperature observations. For SSH, ACCs for SSH are increased in the subpolar North Atlantic (SPNA), tropical Pacific and Indian oceans, but decreased in the South Atlantic due to the fact that the SSH long-term trend

is degraded by the weakly coupled DA in the *assim-i1* system (not shown). Missing contributions from land ice in the model play possibly a role in the degradation. The small ΔACCs for *assim-i2 - assim-i1* suggest that the choice of the climatology reference period does not play an important role for the overall performance of the reanalysis in terms of variability. Significant differences appear close to the sea ice covered areas and are thus likely related to the sea ice state updated via SCDA in *assim-i2*. However, we have limited confidence in the EN4 objective analysis that we used for validation in ice

covered regions where subsurface observations are sparse.





**Figure 7:** ACC for annual SST (a), 0-300m temperature (b), 0-300m salinity (c) and sea surface height (d) for *assim-i1*. ΔACC for *assim-i1 - historical* (e–h), *assim-i2 - assim-i1* (i–l). Temporal coverage is 1950–2018 of SST (ERSSTv5; Huang et al., 2017) and temperature and salinity (EN4.2.1; Good et al., 2013) observations, and 1993–2018 of sea surface height (ARMOR-3D; Larnicol et al., 2006). Hatched areas are locally insignificant, dotted areas are field significant.



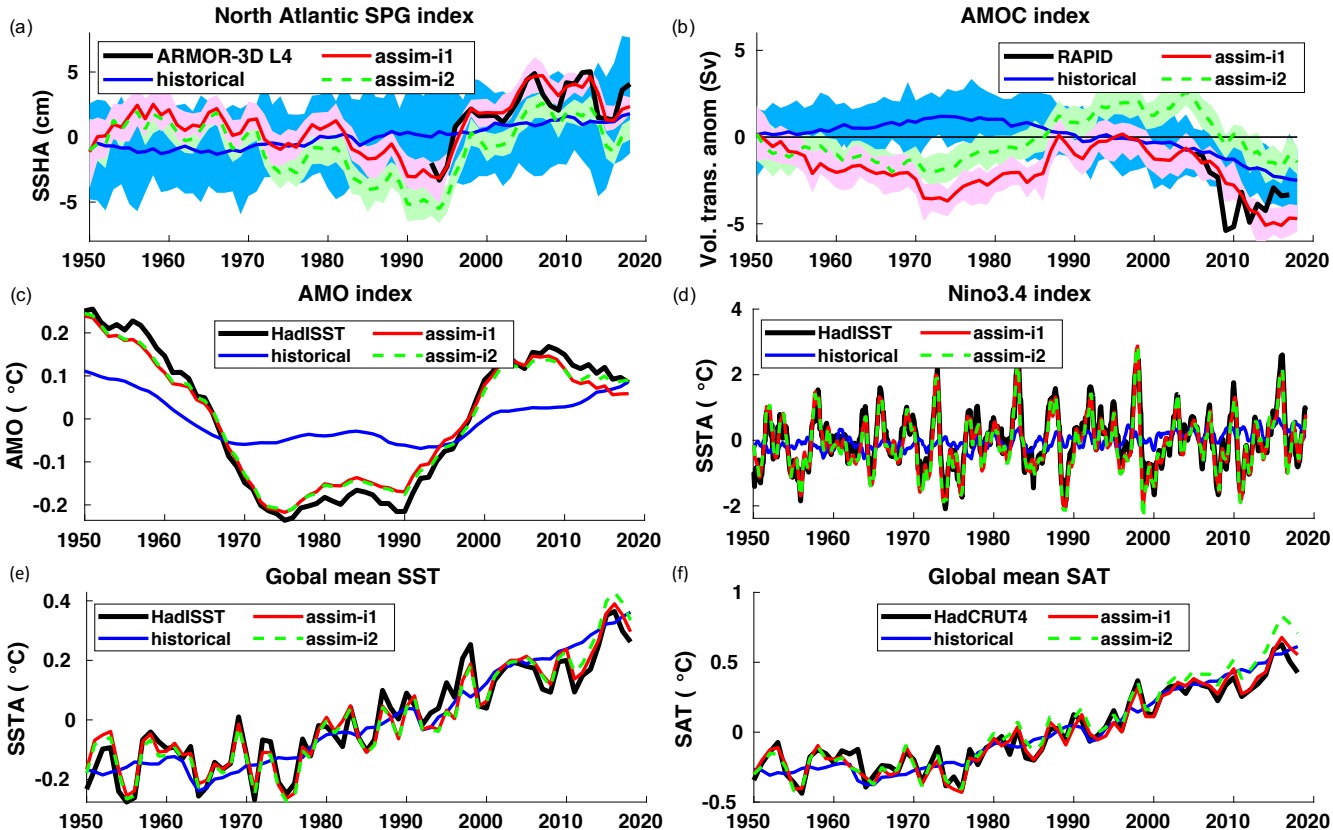

**Figure 8: Anomaly time-series for selected large-scale indices. (a) Annual-mean Subpolar gyre ([60–15 °W, 48–65 °N]) SSH with ARMOR-3D L4 observations (Larnicol et al., 2006). (b) Annual-mean AMOC strength at 26.5 °N with RAPID observations (Johns et al., 2011). (c) Monthly Niño-3.4 index with HadISST observations (Rayner et al., 2003). (d) AMO index computed as the 10-year running mean of detrended SST averaged over the North Atlantic ([0–80 °W, 0–65 °N]), with HadISST observations. (e) Global-mean SST with HadISST observations (Rayner et al., 2003). (f) Global-mean SAT with HadCRUT4 observations (Morice et al., 2012). In all panels, the 1950–2018 climatology of *historical* is removed from *historical, assim-i1* and *assim-i2*. Observations in (a) and (b) are shifted to align their time-mean with *assim-i1*. Observations in (c), (d), (e) and (f) are relative to 1950–2018 climatology.**

We evaluate the effect of assimilation on large-scale climate indices of leading modes of variability (Fig. 8). The North Atlantic subpolar gyre (SPG) circulation exerts strong control on subpolar North Atlantic (SPNA) temperature variations (e.g., Häkkinen and Rhines, 2004), affects the Atlantic meridional overturning circulation (AMOC) by regulating the poleward transport of Atlantic water (Hátún et al., 2005) and has a wide range of marine environmental impacts (e.g., Hátún et al. 2016). The SPG index is here defined as the anomalous SSH averaged over the SPNA box [60–15 °W, 48–65 °N]

(Lohmann et al., 2009). A positive (negative) SPG index reflects a weak (strong) barotropic mass transport in the SPNA region that usually coincides with a warm (cold) SPNA. Figure 8a shows the SPG index over 1950-2018 in *historical*, *assim-i1*, *assim-i2* and observations (altimetry data available from 1993). The observed SPG index exhibits an abrupt shift from a



strong to a weak circulation around 1995, that has been linked to direct North Atlantic Oscillation (NAO) influence (Häkkinen and Rhines, 2004; Yeager and Robson, 2017) and NAO-related preconditioning of the ocean circulation state

(e.g., Lohmann et al., 2009, Robson et al., 2012). The ensemble mean of the *historical* ensemble does not show the shift, but a slow long-term increase likely related to anthropogenic global sea level rise. The min-max range of the *historical* ensemble nevertheless bounds the observed SPG index, suggesting that the model range of variability is not inconsistent with the observed trajectory. The ensemble means of *assim-i1* and *assim-i2* show pronounced strong and weak SPG index phases and match well the observed SPG index changes during 1993-2018. Their simulated weak phase during 1950-1970 and strong

phase during 1980-1997 are also in good agreement with other model studies (e.g., Msadek et al., 2014). The ensemble ranges of *assim-i1* and *assim-i2* are much smaller than that of *historical*, indicating the ensemble members are well synchronised by the assimilation. Despite showing similar decadal scale variability, *assim-i1* and *assim-i2* have different means and long-term trends. The stronger SPG circulation of *assim-i2* goes in tandem with a stronger AMOC, and it is likely that these two are related (Eden and Willebrand, 2001; Eden and Jung, 2001; Böning et al., 2006).

The strength of AMOC is measured continuously from April 2004 at 26.5 °N by a joint US-UK Rapid Climate Change – Meridional Overturning Circulation and Heat flux Array (RAPID-MOCHA; Johns et al., 2011). Accordingly, we define the AMOC index as the yearly anomalies of overturning transport maximum at 26.5 °N. Figure 8b shows the AMOC indices of *historical*, *assim-i1* and *assim-i2* and observations. The ensemble mean of *historical*, a measure for the simulated anthropogenic trend, rises before the mid-70s and then slowly declines. In contrast, the two assimilation products show a

weakening before the mid-70s, followed by a strengthening that is consistent with a dominantly positive observed NAO during that period (Robson et al., 2012; Yeager and Robson, 2017; Zhang et al., 2019). The simulated AMOC strongly declines after 2005, though not as rapidly as in the observations, and flattens after 2010. Similar results have been shown in previous studies (e.g., Keenlyside et al., 2008; Karspeck et al., 2017). As for SPG circulation, *assim-i1* and *assim-i2* show similar multi-year AMOC variations but different long-term trends. Most notable, *assim-i1* stays below the ensemble mean

of *historical* over the entire period, while *assim-i2* surpasses *historical* around 1990, which is more consistent with the anomalously strong AMOC during the mid-90s SPG shift. Results from a supporting experiment suggest that the stronger circulation in *assim-i2* is primarily caused by the different climatological period but also partly by the SCDA update of sea ice (Fig. D1 and related text in Appendix D).

The Atlantic Multidecadal Oscillation (AMO)—or Atlantic Multidecadal Variability—refers to large-scale, low-frequency

SST variations in the North Atlantic, with linkages to AMOC variability (Keenlyside et al., 2015; Yeager and Robson, 2017). Following Enfield et al., (2001), we define the AMO index as the 10-year running mean of linearly detrended SSTs averaged over the entire North Atlantic [0–80 °W, 0–65 °N]. Figure 8c shows the index in observations, *historical*, *assim-i1* and *assim-i2*. In agreement with observations, the indices of all three experiments are in a warm phase during 1950-1965 and 1995-2018 and a cold phase during 1965-1995. However, the *historical* ensemble mean (representing the forced response of

the model) underestimates the amplitude, exhibits a longer cold phase as well as an upward trend after 2010, when observations show a downward trend. As a result of assimilating SST observations, the AMO indices of *assim-i1* and *assim-*





*i2* both follow the observed index with only minor departures. *assim-i2* shows a slightly weaker post-2000 downward trend than *assim-i1* and observations, either related to differing sea ice behaviour or differences in AMOC.

While ocean dynamics in the Atlantic basin give rise to multi-year climate predictability, ENSO variability is an important
source for seasonal and interannual predictability. The ESM features realistic ENSO characteristics (Fig. C5, C6 and text in Appendix C). But how well do monthly DA updates synchronise the model's ENSO variability with the observed one? Figure 8d shows the monthly Niño 3.4—computed as the average of SST in the region [120–170 °W, 5 °S–5 °N]—for *historical*, *assim-i1* and *assim-i2* and HadISST. Both *assim-i1* and *assim-i2* accurately reproduce the observed index, showing a perfect match of the large 1998 event but slightly underestimate other peaks. We attribute the good performance
to that DA in NorCPM1 constrains well thermocline depth (equivalent to warm water volume) in the equatorial Pacific that is critical to develop ENSO events (Meinen and McPhaden, 2000; Wang et al., 2019). The Niño 3.4 indices of *assim-i1* and *assim-i2* are almost identical, meaning that the climatology reference period defined in anomaly assimilation and jointly updated sea ice state have little impact on the equatorial Pacific. The ensemble mean of *historical* has a smaller amplitude but is nevertheless positively correlated with the observed index (r=0.2), suggesting a small contribution from external
forcing.

Last, we consider the effect of assimilation on the global mean SST representation. Figure 8e shows the anomalies of global mean SST evolutions for *historical*, *assim-i1*, *assim-i2* and HadISST. *historical* captures the long-term warming trend and some shorter volcanic cooling events (e.g., after the 1963-Mt Agung and 1991-Mt Pinatubo eruptions). *assim-i1* and *assim-i2* additionally capture the high-frequency variability on top of the forced signal. The assimilation experiments show minor
discrepancies with respect to observations, such as a too weak post-Mt Pinatubo recovery and a seemingly underestimated 1998-El Niño imprint on global mean SST. *assim-i2* exhibits a slightly more positive trend after 2010 compared to *assim-i1*, which likely is the imprint of the more positive trend in AMO on global mean SST. The behaviour of global mean SAT (Fig. 8d) is similar to that of SST and will be further addressed in Section 3.1.6.

### 3.1.4 Ocean biogeochemistry variability

The correlation skills for the analysis runs of annual-mean Primary Production (PP), pCO$_2$ and air-sea CO$_2$ fluxes are shown in Figure 9. For PP, the total skill (with contribution from external forcing) is high and field significant in the tropical Pacific and Indian Oceans, with some skill in the subtropical oceans. The ΔACCs between *assim-i1* and *historical*, measuring assimilation benefit, are not field significant and smaller in value than the ACCs of *assim-i1*, indicating that most skill comes from the external forcing. Still, large regions in the tropical Pacific and Indian Oceans feature high ΔACCs that are locally
significant. The ΔACCs between *assim-i2* and *assim-i1* are generally small. The largest differences are found in the polar regions, although precaution should be taken when evaluating the PP in these regions due to the low coverage of satellite data.





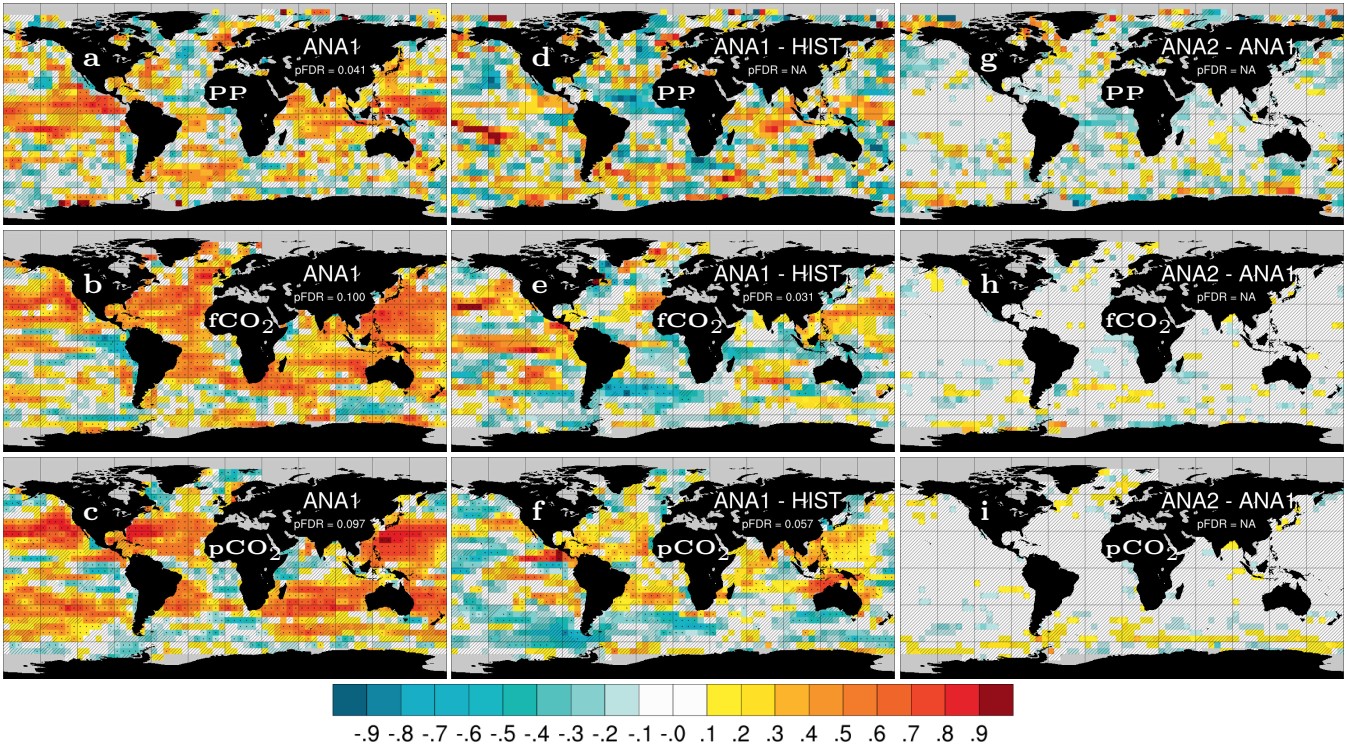

**Figure 9: ACC for annual primary production (a), CO₂ flux (b) and surface pCO₂ (c) for *assim-i1*. ΔACC for *assim-i1 - historical***
**(d–f), *assim-i2 - assim-i1* (g–i). Temporal coverage is 1998–2018 for observed primary production (GlobColour; Garnesson et al.,**
**2019) and 1982–2017 for CO₂ flux and surface pCO₂ (SOCCOM; Landschützer et al., 2019). Hatched areas are not locally**
**significant, dotted areas are field significant.**

For the $CO_2$ fluxes and $pCO_2$, the total skill is high and field significant over the tropical and subtropical oceans. Exceptions
are eastern part of the tropical Pacific, and the southern subtropical Pacific for the $CO_2$ fluxes. For $CO_2$ fluxes, there is also
high skill in the southern part of the Southern Ocean, and in the Nordic Seas. This is not the case for $pCO_2$, which suggests
that part of the $CO_2$ flux skill might be related to successful synchronisation of sea-ice variability. As for PP, the ΔACCs
relative to *historical* are considerably smaller than the ACCs of *assim-i1*, despite the linear detrending that was applied to the
$CO_2$ fields before the ACC computation. The ΔACCs remain field-significant in parts of the subtropical and tropical oceans,
although with a reduced westward extension of the skilful areas. Contrary to expectation, the subpolar North Atlantic shows
little skill. As for PP, skill differences for $CO_2$ fluxes and $pCO_2$ are small between *assim-i1* and *assim-i2*.

### 3.1.5 Sea ice variability

We evaluate the success of our assimilation in phasing sea ice variability. We use ACC maps of annual mean sea ice
concentration and HadISST (Reyner et al., 2003) data from 1950-2018 as a benchmark (Fig. 10).







**Figure 10: ACC for annual sea ice concentration in Arctic (a) and Antarctic (b) for *assim-i1*. ΔACC for *assim-i1 - historical* (c–d), *assim-i2 - assimi-i1* (e–f). Observations are from HadISST (Rayner et al., 2003) over the period 1950–2018. The data are interpolated to a regular 2°×2° grid. Hatched areas are not locally significant, dotted areas are field significant.**

Over the Arctic, *assim-i1* features overall high skill. While much of this skill is from the externally forced trend, positive *assim-i1 - historical* ΔACCs show that ocean DA considerably improves the agreement in the marginal ice zones. Positive

ΔACCs for *assim-i2 - assimi-i1* show that updating the sea ice state via SCDA of ocean observations further improves the agreement, including over the central Arctic.

Over the Antarctic, *assim-i1* shows modest to high skill and only isolated negative ACCs. Strikingly, the *assim-i1 - historical* ΔACCs are as high or higher than the absolute ACCs of *assim-i1*. This means that assimilation correct for the negative trend in the historical ensemble SCDA again improves the skill (Fig. 10f), especially close to the coast where the

ACCs of *assim-i1* are low or negative (Fig. 10b).



**Figure 11: ACC for annual 2 m temperature (a), precipitation (b), sea level pressure (c), and 500 hPa geopotential height (d) for** *assim-i1*. ΔACC **for** *assim-i1 - historical* **(e–h),** *assim-i2 - assim-i1* **(i–l). Temporal coverage is 1950–2018 for observed 2 m**
**temperature (HadCRUT4; Morice et al., 2012), precipitation (CRU TS4.03; Harris et al., 2020), sea level pressure (NCEP reanalysis; Kalnay et al., 1996) and geopotential (extended ERA5; Harris et al., 2020). Hatched areas are not locally significant, dotted areas are field significant.**

### 3.1.6 Atmosphere variability

Because our DA is weakly coupled with respect to the atmosphere, we expect a partial synchronisation of atmospheric
variability from the combined influence of the ocean surface-sea ice states and the external forcings. The reanalysis performance provides a hypothetical upper bound for the achievable atmospheric-land prediction skill with our system, assuming close to perfect prediction of ocean variability and skilful prediction of sea ice variability. We assess the synchronisation of atmospheric variability with ACCs of annual-mean SAT, precipitation over land (PR), sea level pressure



(SLP) and 500 hPa geopotential height (Z500) for *assim-i1* (Fig. 11a-d). We also consider ΔACCs for *assim-i1 - historical*

and *assim-i2 - assim-i1* to isolate skill contribution from assimilation and skill differences between two reanalysis products.

For SAT, the ACCs of *assim-i1* are high over both ocean and land. Most of the assimilation benefit is located over the oceans, as revealed by the ΔACCs for *assim-i1 - historical*, with benefits over land mainly found in the tropical regions and also over northwest North America i.e., regions that are strongly affected by ENSO variability. *assim-i2* does not show any significant skill improvement over *assim-i1*, despite the sizable improvements in sea ice variability when updating the sea ice

state via SCDA. This is likely because the improvements in sea ice extent (Fig. 10) occur mostly during summer when they have little impact on surface temperatures (Deser et al., 2010). For global scale SAT synchronisation, the global warming hiatus at the beginning of the 21$^{st}$ Century, which has been attributed to both internal variability and external forcing, makes an interesting test case. Figure 8f shows that global mean SAT anomaly of *assim-i1* reproduces well the flat post-2000 trend of the observations, while *assim-i2* and *historical* continue to warm, consistent with their AMO and AMOC evolutions. The

better match of *assim-i1* with observed global mean SAT does not necessarily imply that *assim-i1* is more correct than *assim-i2*. It is possible that *assim-i1* makes up for a missing post-2000 cooling signal over the continents by an unrealistic low reduction of winter sea thickness during that period, something that warrants further investigation.

For PR over land, the ACCs of *assim-i1* are overall positive. The ΔACCs for *assim-i1 - historical* show similar strength and pattern, indicating a limited contribution to the ACCs of *assim-i1* from the anthropogenically driven spin-up of the

hydrological cycle. The ΔACCs for *assim-i2 - assim-i1* do not suggest statistically significant performance differences between the two products.

For SLP, the ACCs of *assim-i1* are most positive over the low- and high-latitudes and less positive over the mid-latitudes, with slightly negative values over the Southern Ocean and Eurasia. The ΔACCs for *assim-i1 - historical* suggest that a large portion of the positive skill can be attributed to the assimilation, including benefits over the North Pacific that stretch over

North America and also over the SPNA, consistent with ENSO influence. However, assimilation seems to cause degradation over the subtropical North Atlantic, Central Europe, Siberia and East Asia. The ΔACCs for *assim-i2 - assim-i1* reveal that updating sea ice improves SLP performance over the Arctic. Assimilation also seems to partly mitigate the skill deficit over Central Europe while degrading skill further east.

For Z500, the correlation skill of *assim-i1* is virtually saturated over the tropics, decreases towards the mid-latitudes and

again slightly increases towards the poles. While modest ΔACCs for *assim-i1 - historical* indicate that external forcing contributes significantly to high tropical skill, assimilation leads to consistent skill enhancement in those regions. One should note that a change in correlation from 0.6 to 0.9 equates to a change in explained variance from 36 % to 81 % i.e., more than a doubling. Hence, the benefit from assimilation is more substantial than the ΔACCs alone would suggest. Significant skill enhancement is also present over the mid-to-high latitudes, presumably related to ENSO influence on the extratropical

atmospheric circulation. The ΔACCs for *assim-i2 - assim-i1* indicate weak improvement over the polar regions, albeit not statistically significant, and no signs of degradation, as consequence of updating the sea ice during the assimilation.





### 3.2 Hindcast performance

This section evaluates retrospective predictions with NorCPM1 that are initialised on November 1 (i.e., no observations after October 31 are utilized in the initialisation) of the years 1960–2018. We demonstrate skill benefits from forecast
initialisation as well as from using a dynamic prediction system. To assess skill degradation with forecast lead time, we consider the different forecast ranges lead year 1 (LY1), lead years 2–5 (LY2–5) and lead years 6–9 (LY6–9). We compare against the skill of NorCPM1's reanalyses, uninitialized prediction (constructed from *historical*) and persistence forecast (defined in Appendix B). We also highlight performance differences between the two hindcast products *hindcast-i1* and *hindcast-i2*. The following subsections present skill evaluations for the physical ocean, marine biogeochemistry, sea ice and
atmosphere.





**Figure 12: Prediction skill for SST.** ACC of hindcast-i1 (a), ΔACC of hindcast-i1 - persistence (b), ΔACC of hindcast-i1 - historical (c) and ΔACC of hindcast-i2 - hindcast-i1 (d) for LY1. Middle and right column show the same but for LY2–5 (e–h) and LY6–9 (i–l). Observations use ERSSTv5 (Huang et al., 2017) with coverage 1960–2018. Hatched areas are not locally significant, dotted areas are field significant.





### 3.2.1 Physical ocean variability – globally

SST prediction has the most direct application for near-term climate impact assessment. We evaluate NorCPM1's capability to predict interannual to multi-year SST variations with ACC skill maps for *hindcast-i1* along with skill difference maps for *hindcast-i1–assim-i1*, *hindcast-i1–persistence*, *hindcast-i1–historical* and *hindcast-i2–hindcast-i1* (Fig. 12). For LY1, *hindcast-i1* exhibits generally positive ACCs, exceeding 0.8 over extended areas, that are both locally and field significant except for limited regions in the eastern Pacific and at high latitudes (Fig. 12a). The system loses information of the initial condition over time, resulting in notably smaller ACCs compared to the *assim-i1* reanalysis (Fig. 12b). Significant benefits from initialization, as diagnosed from the ΔACC of *hindcast-i1–historical*, are concentrated in the Pacific and Atlantic sectors of the tropics and Southern Ocean, and also in the subpolar North Atlantic (SPNA) and extending from there into the Eurasian Arctic (Fig. 12d). Consistent with other prediction systems (e.g., Yeager et al., 2018), the SPNA stands out as the region that benefits most from initialisation. However, *hindcast-i1* does not outperform *persistence* in the SPNA (Fig. 12c), indicating that the benefit of initialisation primarily offsets poor performance of the uninitialized dynamical prediction of *historical* in that region. *hindcast-i2* shows improved skill over *hindcast-i1* in sea ice covered regions and in a small part of the SPNA (Fig. 12e). These skill differences are not field significant, but the fact that the two systems differ in their sea ice treatment adds confidence that skill improvements in the polar regions are real. Much of the LY1 skill, in particular in the tropics, is likely related to skilful initialisation of ENSO in NorCPM (Fig. D2 and text in Appendix D), which has been studied in detail using a similar model configuration (Wang et al., 2019).

The LY2–5 and LY6–9 multi-year SST skill patterns (Fig. 12, middle and right columns) resemble that of LY1, but with some notable differences. Large regions in the eastern central North Atlantic, tropical Indian Ocean and West Pacific show elevated skills that exceed 0.9. The same regions show, however, negligible gains relative to uninitialized prediction of *historical* (Fig. 12i,n). Thus, the skill increase relative to LY1 is likely due to the forced trend having more weight, as the 4-year averaging effectively filters out interannual internal variability, and less due to the presence of more predictable internal climate variability on multi-year time scales or forecast shock that more strongly impacts LY1. Despite limited initialisation benefit, the initialised predictions globally outperform persistence except for in the Southern Ocean. Since we expect the persistence forecast to capture a linear trend, this may indicate a significant skill contribution from non-linearities in the forced trend. Also for multi-year prediction, the SPNA and its extension towards the Arctic stand out as the region benefiting most from initialisation, although the benefit is somewhat reduced and less statistically robust than for LY1 (Fig. 12d). Over time, the impact of initialisations in the SPNA diminishes and the system drifts back to the poorly performing simulated forced trend, causing skill deficit to emerge (Fig. 12f,k). The eastern Pacific presents another region where the skill notably deteriorates over time. The historical simulations perform better here than for the SPNA (Fig. 12i,n), suggesting a detrimental effect of initialisation on multi-year scales on Pacific SSTs notwithstanding the positive effect on LY1 prediction. Also for multi-year prediction, *hindcast-i2* performs better than *hindcast-i1* in the high latitude regions, notably





in the northwestern North Atlantic (Fig. 12j,o). However, the multi-year skill *hindcast-i1–historical* and *hindcast-i2–hindcast-i1* differences are both not field significant and we thus cannot exclude they are a sampling artefact.

Skill patterns for the upper ocean temperature and salinity averaged over the top 300 m (Fig. D3,D4), and for sea surface height (Fig. D5)—a proxy for circulation and vertically integrated behaviour—largely reflect those for SST. Skill enhancement due to multi-year averaging is less apparent than for the surface state, presumably due to less presence of interannual climatic noise below the surface. Initialisation benefit in the SPNA extends below the surface, across variables and stands out as a robust feature.

**3.2.2 Physical ocean variability – SPNA**

Initialization of the large-scale ocean circulation and the associated meridional heat transport have been identified as essential for skilful prediction of SPNA climate (e.g., Yeager and Robson, 2017). We evaluate in more detail how well NorCPM1 represents mechanisms that give rise to North Atlantic decadal predictability. This evaluation provides additional forecast quality information, better understanding of the *hindcast-i2–hindcast-i1* skill differences and of how well the
predictive potential for North Atlantic SSTs is realised in the system.

The forced evolution of the AMOC strength shows slight increase until 1980 and weakening thereafter (Fig. 13a, blue solid). *assim-i1* initializes the circulation in an anomalous weak state prior to 1990, close to neutral between 1990 and 2010, and weak again thereafter (red solid), with the initial perturbations tending to be outside the internal variability range (blue shading). After initialisation, the circulation (purple solid) rapidly relaxes towards the unperturbed ensemble-mean state
evolution of *historical* (blue solid). Because ocean heat exchange between the subtropical and the subpolar North Atlantic covaries with the variability in AMOC strength (Fig. D5e-g), the anomalies of the northward heat transport at the time initialization (Fig. 13b, red solid) roughly resemble those of the circulation, mostly showing anomalously negative transports, except during the 90s. The heat transport relaxes towards the ensemble-mean of *historical* during the hindcasts. *assim-i2* shows generally stronger circulation and heat transports with weaker long-term decline than *assim-i1* (Fig. 13d,e).
These circulation and heat transport differences are key to explaining strikingly different subpolar North Atlantic temperature evolutions in *hindcast-i1* versus *hindcast-i2* (Fig. 13c,f). *hindcast-i1* and *hindcast-i2* notably drift away from the observed SPNA-averaged temperature trajectory, suggesting that both configurations struggle to predict the observed decadal SPNA temperature trends. However, while *hindcast-i1* exhibits drift behaviour towards cooling (most pronounced during 1960-1980 and after 2005), *hindcast-i2* exhibits drift behaviour towards warming (most severe during 1980-2000).

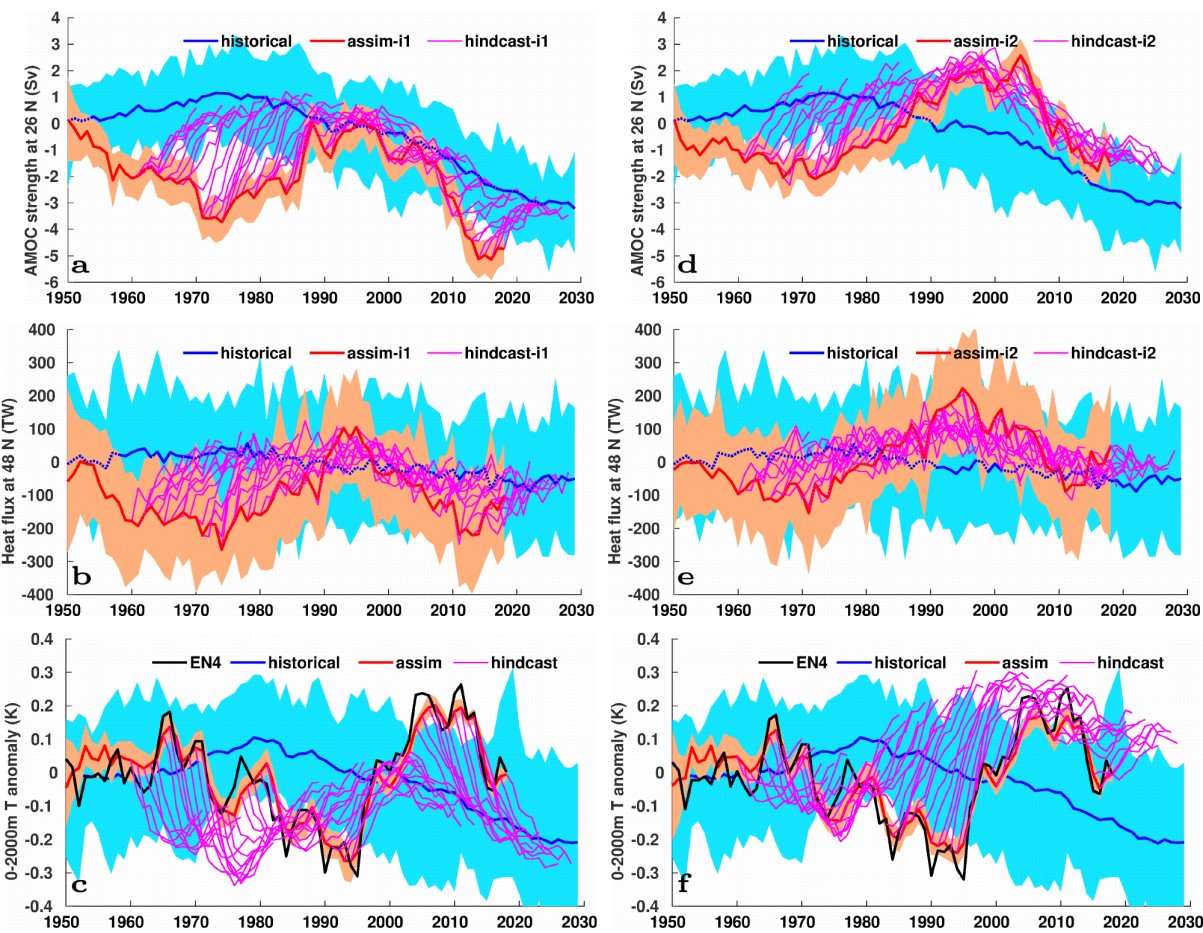

**Figure 13: AMOC strength at 26 °N, Atlantic meridional ocean heat transport at 48 °N and 0–2000 m temperature averaged over SPNA box [60–15 °W, 48–65 °N] for i1 (a–c) and i2 (d–e). Solid lines show ensemble means of *historical* (blue), *assim* (red) and *hindcast* (purple) experiments, with the 1950–2010 average of *historical* subtracted. Shading denotes ensemble minima and maxima.**

Diagnosing the hindcast SPNA temperature evolution from the anomalous ocean heat transport across 48 °N (Fig. D6a,c) or the regression of heat transport on AMOC (Fig. D6b,d) results in a very similar behaviour. The SPNA 0–2000m heat content changes are well balanced by transport changes across 48 °N and anomalous surface fluxes over the SPNA region (not shown). The latter mainly act to dampen the temperature signal, explaining the greater amplitude of the diagnosed temperature evolutions. The resemblance of diagnosed and simulated hindcast evolutions suggests that circulation exerts a strong control on the simulated SPNA temperature evolution and that poor SPNA prediction is largely a consequence of poor initialisation of AMOC and associated poleward heat transport. Errors in the simulated externally forced AMOC trend and associated heat transport likely affect the skill as well.





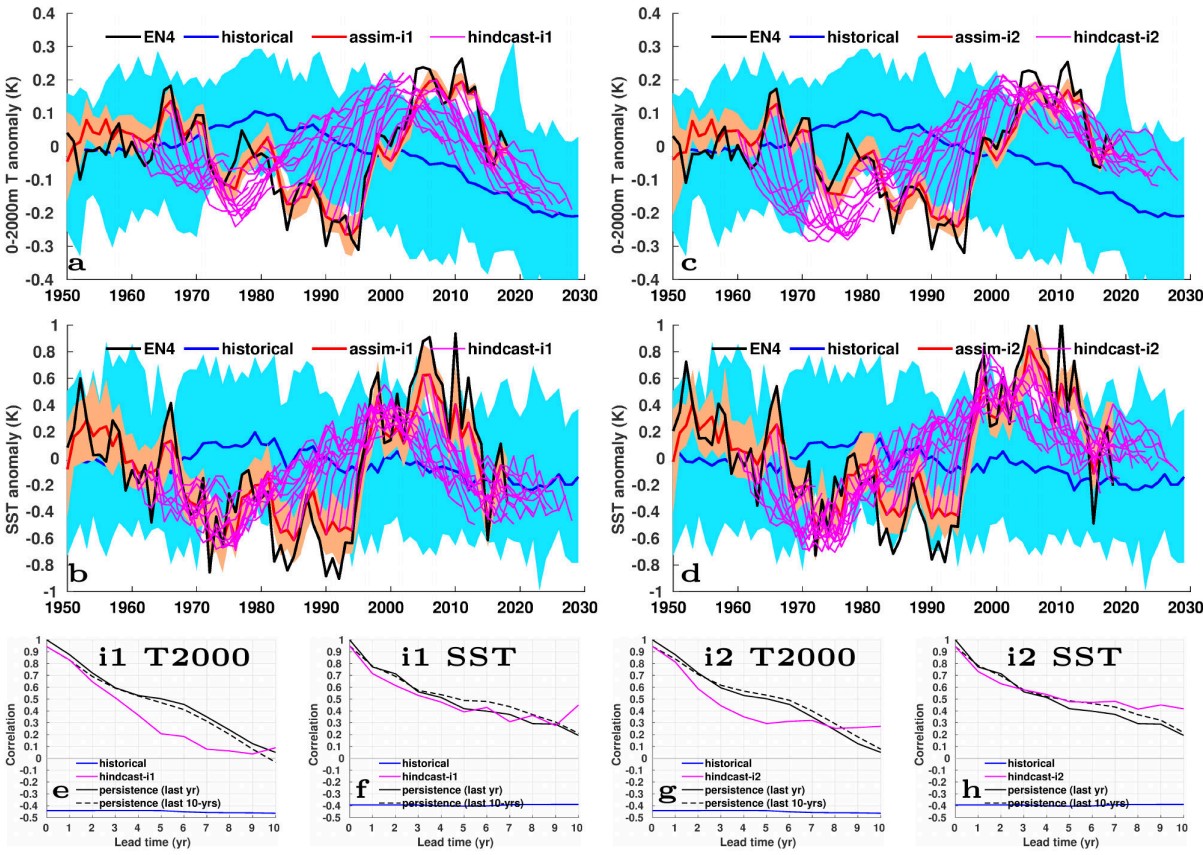

**Figure 14: Drift-corrected 0–2000m temperature (T2000) and SST averaged over SPNA box [60-15 °W, 48-65 °N] for i1 (a,b) and**
**i2 (c,d), respectively. Solid lines show ensemble means of *historical* (blue), *assim* (red) and *hindcast* (purple) experiments, with the**
**1950–2010 average of *historical* subtracted. Shading denotes ensemble minima and maxima. Also shown, ACCs as function of lead**
**time for T2000 and SST for i1 (e,f) and i2 (g,h), respectively. The persistence forecasts use the average over the last one year (solid)**
**and ten years (stippled) from the observations.**

How can *hindcast-i1* and *hindcast-i2* exhibit very different SPNA 0–2000 m temperature evolutions but similar correlation
skills? Applying lead-dependent drift correction largely removes the differences (Fig. 14a,c). Remaining differences hint at a
slight time dependence, consistent with the somewhat different long-term trends in AMOC strength in *assim-i1* and *assim-i2*
(Fig. 13a vs d). In terms of ACC skill, *hindcast-i2* performs marginally better than *hindcast-i1* for long lead times but does
not outperform persistence (Fig. 14e,g). The results for SPNA SST (Fig. 14b,d) generally resemble those for 0–2000 m
temperature but look slightly more promising, with *hindcast-i2* performing marginally better than persistence for long lead
times (Fig. 14f,h).

**Figure 15: Prediction skill for primary production (PP). ACC of hindcast-i1 (a), ΔACC of hindcast-i1 - persistence (b), ΔACC of hindcast-i1 - historical (c) and ΔACC of hindcast-i2 - hindcast-i1 (d) for LY1. Middle and right column show the same but for LY2–5 (e–h) and LY6–9 (i–l). Observations use GlobColour (Garnesson et al., 2019) with coverage 1998–2018. Hatched areas are not locally significant, dotted areas are field significant.**





### 3.2.3 Ocean biogeochemistry variability

As for the reanalyses, we evaluated ocean biogeochemistry performance for PP, surface $pCO_2$ and surface $CO_2$ flux. Figure 15 shows maps of PP prediction skill for LY1, LY2–5 and LY6–9. While the results are patchy, some coherent patterns can be distinguished. For the total LY1 skill of *hindcast-i1* (Fig. 15a), ACCs are relatively high and field significant over large

parts of the tropical Pacific and tropical Indian Oceans. The correlations stay relatively high for longer lead times (Fig. 15f,k), although their significance is reduced. When subtracting the skill of *historical* (Fig. 15d,i,n), the correlation is greatly reduced, showing that much of the total skill comes from external forcing. The only region with a coherent pattern of locally significant correlation differences is in the tropical Pacific [0-30 °S, 120-150 °W], which shows positive skill differences until LY2–5. For LY6–9, the correlation differences become statistically not significant, although the values stay relatively

high. The ΔACCs for *hindcast-i1 - assim-i1* (Fig. 15b,g,l) are negative over the tropical Indo-Pacific and large parts of the South Pacific and Southern Ocean, indicating information from initialisation is lost over time, while they are positive over the tropical Atlantic, parts of the Atlantic subpolar gyre and most parts of the extratropical Indo-Pacific. Paradoxically, the analysis used to initialize the hindcasts does not consistently outperform the hindcasts. Improvement of the initialised dynamic predictions over *persistence* can be seen for LY2–5 and LY6–9, but not for LY1. Thus, temporal nonlinearities in

the externally forced climate trend are likely to contribute to skill, as *persistence* should capture any linear trends due to forcings and most of the skill comes from the external forcing. Differences between the two sets of hindcasts lack statistical robustness (Fig. 15e,j,o).

Using satellite chlorophyll measurements for model evaluation is subject to caveats. For example, temporal data coverage is relatively short and the spatial data coverage at high latitudes is poor due to cloudiness. Following Yeager et al. (2018), we

therefore also analysed potential predictability with respect to initialisation, i.e., the model's ability to hindcast its own analysis over the period 1960–2018 (Fig. 16). The results become less patchy, and the total skill stays field significant for large parts of the global ocean until LY6–9. Removing the skill of the historical run again reveals that there are regions where the skill is improved by initialization, notably the subtropical gyres and the Nordic Seas (Fig. 16d,i,n). Note that subtracting negative *historical* ACCs leads to ΔACCs higher than the absolute ACCs of *hindcast-i1* itself. Therefore, a large

skill benefit from initialisation does not necessarily translate into a societally useful absolute skill. We analysed time series of region-averaged PP between 1970-2018 in regions of high skill, namely the subtropical gyres of the Pacific, Atlantic and Indian Oceans, as well as the Nordic Seas (not shown). The Nordic Seas is the only region with a strong positive correlation between *hindcast-i1* and *historical* (r=0.5 and 0.6 for single year and four-year means, respectively), indicating that there is a large contribution of the external forcing to the predictive skill. There, the correlation between the *hindcast-i1* and *assim-i1*

is close to 0.75 for all lead year ranges, indicating an improvement with respect to *historical*, with the largest difference for LY1. For the other regions there is considerable agreement between the *hindcast-i1* and the *assim-i1* for LY1, with correlations exceeding 0.7. For the subtropical gyres in the Pacific and South Atlantic the agreement between the hindcasts and the analysis extends to LY2–5, while the skill in the Indian and North Atlantic Oceans drops beyond LY1.



**Figure 16: Potential predictability for primary production (PP). ACC of hindcast-i1 (a), ΔACC of hindcast-i1 - persistence (b), ΔACC of hindcast-i1 - historical (c) and ΔACC of hindcast-i2 - hindcast-i1 (d) for LY1. Middle and right column show the same but for LY2–5 (e–h) and LY6–9 (i–l). Synthetic observations constructed from the ensemble mean of the first 10 members of *assim-i1* with coverage 1960–2018. Hatched areas are not locally significant, dotted areas are field significant.**

Despite the ambiguous results, the predictability of PP of a couple of years in the tropical/subtropical Pacific is in agreement with the results from perfect model experiments (Fransner et al., 2020) and Seferian et al. (2014), who found a predictability of 2–5 years when comparing with satellite-based PP in the same region. Also, Krumhardt et al. (2020) found a potential predictability of PP of a couple of years in tropical/subtropical regions when comparing to a reconstruction based on an ocean simulation forced with an atmospheric reanalysis. However, to remove the effect of external forcing they performed a linear detrending. This partly removes the effect of climate change but not of other episodic external forcing such as volcanic eruptions. Frölicher et al. (2020) found a perfect model predictability of more than 10 years in some parts of the subtropical gyres in their perfect model study.



Studies have yet to report predictability of PP in high latitudes if compared to observational data. In these regions the use of satellite observations is not reliable because of the lower data coverage and more variable chlorophyll:carbon ratio of phytoplankton (Frigstad et al., 2014). However, several recent perfect and potential predictability studies suggest that

predictability of primary production in high latitudes is low or even non-existent on interannual to decadal time scales (Fransner et al., 2020, Frölicher et al., 2020, Krumhardt et al., 2020).

For the $CO_2$ fluxes, a high total skill is found for all lead year ranges in the Atlantic, Indian and in the North Pacific Oceans (Fig. 17). Most of the skill comes from the external forcings, as revealed by small ΔACCs (of both signs) for *hindcast-i1 - historical* (Fig. 17d,i,n). The relatively high ΔACCs in the southern subtropical Pacific are a result of subtracting ACCs that

are negative in *historical*. Only in the tropical Pacific and for LY1 initialisation leads to moderately positive total skill. The low predictive skill that we find for the $CO_2$ fluxes is in agreement with what was found in Lovenduski et al. (2019), that compared hindcasts of CESM-DPLE (Yeager et al., 2018) with the same observational dataset. However, other model systems (Li et al., 2016, Ilyina et al., 2020) and perfect model studies (Seferian et al., 2018, Fransner et al., 2020) have shown a predictability of $CO_2$ fluxes up to several years, particularly in the North Atlantic subpolar gyre, suggesting that

there is room for improvement for the NorCPM1 decadal predictions.





**Figure 17: Prediction skill for surface CO$_2$ flux.** ACC of hindcast-i1 (a), ΔACC of hindcast-i1 - persistence (b), ΔACC of hindcast-i1 - historical (c) and ΔACC of hindcast-i2 - hindcast-i1 (d) for LY1. Middle and right column show the same but for LY2–5 (e–h) and LY6–9 (i–l). Observations use SOCCOM (Landschützer et al., 2019) with coverage 1982–2017. Hatched areas are not locally significant, dotted areas are field significant.



**Figure 18: ACC for sea ice concentration (SIC) for *historical* (left), *hindcast-i1* (middle) and *hindcast-i2* (right) in Arctic (top row) and Antarctic (bottom row) for LY1. Observations are from HadISST1 (Rayner et al., 2003) over the period 1960–2018. The data are interpolated to a regular 1°×1° grid.**

## 3.2.4 Sea ice variability

Previous studies have found robust initialisation benefits for sea ice prediction lasting for a couple of months (Guemas et al., 2016), with some reemergence of skill during the second year (Day et al., 2014). While these studies reported strong seasonal dependencies, the evaluation here is limited to hindcasts initialised in November. We evaluate LY1 predictions of annual-mean sea ice concentration (SIC) against HadISST1 (Rayner et al., 2003) over the period 1960-2018 that includes historical observations as well as satellite estimates (Fig. 18). In the Arctic, the uninitialized predictions (*historical*) show externally forced skill in the Barents, Kara and Chukchi Seas as well as the Canadian Archipelago (Fig. 18a). *hindcast-i1* shows consistently higher ACCs than *historical* in these regions and additionally exhibits first-year skill in sub-Arctic regions, e.g., in the Bering and Greenland Seas (Fig. 18b). *hindcast-i2* benefits from a stronger constraint on the sea ice

initial state compared to *hindcast-i1*, resulting in generally higher and more widespread skill (Fig. 18c). In the Antarctic,
*historical* shows patches of both positive and negative ACC (Fig. 18d). There are nearly no regions where *hindcast-i1*
shows negative ACC, while regions with positive ACC are limited to the east-Pacific sector of the Southern Ocean (Fig.
18e). *hindcast-i2* shows even more positive skill, that extends into the Atlantic sector (Fig. 18f), but also some negative skill
in Pacific sector, albeit less negative as in *historical*.

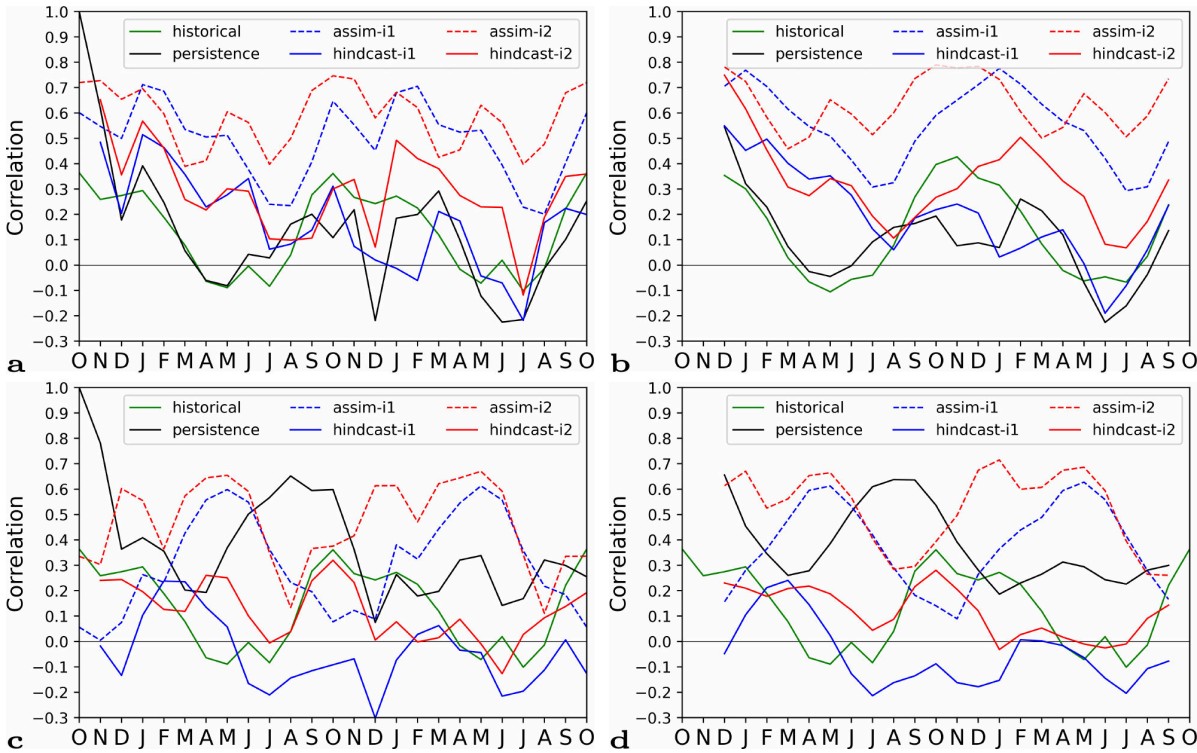

**Figure 19: ACC for Arctic (top) and Antarctic (bottom) total ice area as a function of lead month for monthly averages (left) and
3-month averages (right). The persistence forecast uses the observed October mean, while the hindcasts were initialised November
1. Observations are from HadISST1 (Rayner et al., 2003) limited to the satellite era 1979–2018.**

We address seasonal dependence and temporal forecast limit of sea ice prediction by computing the ACC of total Arctic and
Antarctic sea ice area as a function of lead month after November initialisation (Fig. 19a,c). The Arctic ACC of *persistence*
drops rapidly and both *hindcast-i1* and *hindcast-i2* show comparable or higher skill during the first winter and into spring.
From early summer, the ACCs of *hindcast-i1* remain close to zero. In contrast, *hindcast-i2* shows some reemergence of skill
from the first autumn extending into the second year. Performing 3-month pre-averaging makes the skill reemergence for
*hindcast-i2* and improvements over *hindcast-i1, persistence* and *historical* clearer (Fig. 19b,d). The uninitialized prediction
from *historical* shows some skill during autumn and winter but no skill during summer. For the Antarctic, both uninitialized



and initialised predictions perform inferior to *persistence*, with *hindcast-i1* performing worst (Fig. 19c,d). Nevertheless, *assim-i1* and *assim-i2*, which provide the initial conditions for *hindcast-i1 and hindcast-i2*, outperform *persistence* during most of the year, except in austral winter when *persistence* shows reemerging skill. This suggests that model errors are skill-limiting rather than imperfect initialisation in that region.

905

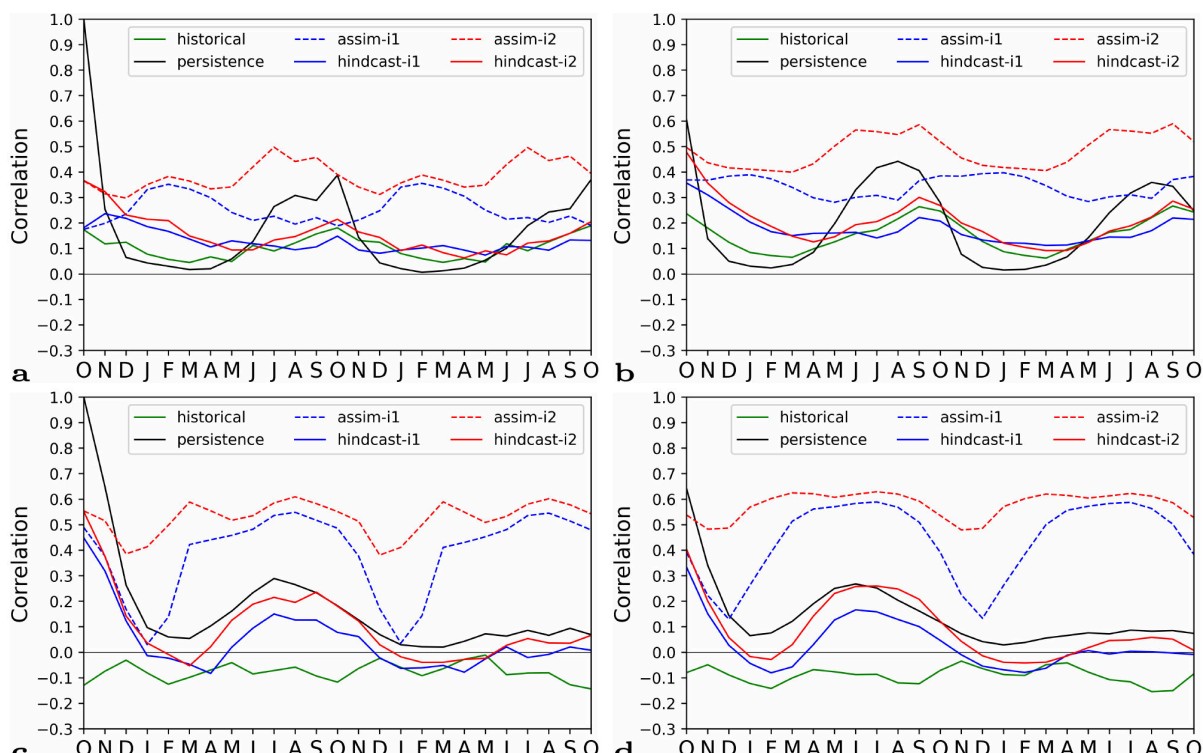

**Figure 20: Hemispheric correlation skill for Arctic (top) and Antarctic (bottom) ice area as a function of lead month for monthly averages (left) and 3-month averages (right). The data is first interpolated to a 5°×5° degree grid and correlations are then computed jointly over space and time, applying area-weighting and only considering grid cells with non-zero temporal standard deviations. The persistence forecast uses the observed October mean, while the hindcasts were initialised November 1. Observations are from HadISST1 (Rayner et al., 2003) limited to satellite era 1979–2018.**

Since regional sea ice variability is not necessarily in phase with total hemispheric sea ice area, we define a hemispherically integrated skill score for predicting local (i.e., grid cell scale) sea ice conditions (Fig. 20). We first interpolate observation and model data to a common 5°×5° grid and then reduce the space and time dimensions to a vector that is used in the ACC computation. We apply square root grid cell area weighting and only consider cells with non-zero temporal standard-deviation. The squared score gives the fraction of predicted sea ice concentration variance. A theoretical score of one would imply perfect prediction in every location (note the score depends on the resolution of the common grid). In addition to





monthly ACCs (Fig. 20a,c), we present 3-monthly ACCs (Fig. 20b,d) that are smoother. For the Arctic (Fig. 20a), the *hindcast-i2* score reaches 0.4 during the first lead month, outperforming the sharply dropping *persistence* score (with 1-month e-folding scale), and remains significantly higher than the uninitialized *historical* score throughout winter and spring and marginally higher during the remainder of the two lead years. *persistence* shows a reemergence of skill during summer and autumn that is present but weaker in *hindcast-i2*. *hindcast-i1* shows a score below 0.3 for the first lead month and no initialisation benefit after the first spring. Consistent with these differences in hindcast scores, *assim-i2* features consistently higher scores than *assim-i1*. For the Antarctic (Fig. 20c), the initialised predictions do better than the uninitialized ones (with no or negative skill) but for the most part fall behind *persistence*. *assim-i2* shows notably higher and more stable skill than *assim-i1*, explaining better performance of *hindcast-i2* over *hindcast-i1*.

We have demonstrated initialisation benefits for predicting sea ice up to two years ahead in NorCPM1, but can initialisation improve prediction of decadal trends in Arctic sea ice decline? An analysis of Northern Hemisphere integrated sea ice volume (SIV) provides little evidence for that (Fig. D6). The initialised hindcasts have a tendency to simulate a flatter trend than the *historical* experiment over the last decade, which arguably can be interpreted as an improvement. Despite the lack of initialisation benefit, the comparison between the two reanalysis products and their corresponding hindcasts is instructive and illustrates the importance of forecast drift correction. As mentioned in Section 3.1, the sea ice state update in *assim-i2* overall reduces the simulated ice volume to values closer to observations, whereas the climatology of *assim-i1* remains unaffected. Once assimilation is stopped, the sea ice in *hindcast-i2* grows back towards levels comparable to the no-assimilation *historical* experiment. As a result, the hindcast-i2 predictions all simulate strongly positive decadal sea ice volume trends, whereas *hindcast-i1* produces flat or negative trends more in line with observations. Adjusting for lead-dependent forecast drift largely eliminates differences in the decadal SIV trends between the two hindcast products.

### 3.2.5 Atmosphere variability

Transfer of skill from the ocean to the atmosphere and over land is key to societally relevant climate prediction. We assess the extent such transfer is realized in NorCPM1 from ACCs of SAT, total precipitation (PRECIP), 500 hPa geopotential height (Z500) and sea level pressure (SLP).

For SAT, *hindcast-i1* shows considerable first-year and multi-year hindcast skill that exceeds persistence skill over most land areas, except over central South America and parts of Africa and South Asia (Fig. 21a,c). The LY1 initialisation benefit (Fig. 21d) is highest over the subpolar North Atlantic, extending from there over Scandinavia and western Siberia. Siberia is also the only region where *hindcast-i2* consistently shows higher skill than *hindcast-i1* (Fig. 21e). While the ΔACCs are not field significant, it is plausible that differences in sea ice initialisation impact skill over adjacent land (Ringgaard et al., 2020). The LY1 ΔACCs relative to *historical* (Fig. 21d) hint ENSO-related initialisation benefits over low-latitude coastal regions as well as over northwest North America. For LY2–5 and LY6–9, the difference maps indicate little initialisation benefit, implying that most multi-year SAT skill over land stems from the externally forced trend in NorCPM1.






**Figure 21: Prediction skill of 2 m temperature (SAT). ACC of hindcast-i1 (a), ΔACC of hindcast-i1 - persistence (b), ΔACC of hindcast-i1 - historical (c) and ΔACC of hindcast-i2 - hindcast-i1 (d) for LY1. Middle and right column show the same but for LY2–5 (e–h) and LY6–9 (i–l). Observations use HadCRUT4 (Morice et al., 2012) with coverage 1950–2019. Hatched areas are not locally significant, dotted areas are field significant.**




**Figure 22: Prediction skill of total precipitation (PR). ACC of hindcast-i1 (a), ΔACC of hindcast-i1 - persistence (b), ΔACC of hindcast-i1 - historical (c) and ΔACC of hindcast-i2 - hindcast-i1 (d) for LY1. Middle and right column show the same but for LY2–5 (e–h) and LY6–9 (i–l). Observations use CRU TS4.03 (Harris et al., 2020) with coverage 1950–2018. Hatched areas are not**
**locally significant, dotted areas are field significant.**

For PRECIP, *hindcast-i1* exhibits positive skill over most land regions for all lead ranges (Fig. 22a,f,k). For LY1 it is highest and field significant over the western tropical Pacific and Indonesian Archipelago (Fig. 22a). The LY1 skill difference to *historical* (Fig. 22d), a measure for the benefit from initialisation, resembles the *hindcast-i1* skill itself, suggesting only a





small contribution from the externally forced trend to the first-year skill. For LY2–5 and LY6–9, the *hindcast-i1* skill over

the western tropical Pacific, Indonesian Archipelago and Australia is considerably reduced or disappears, whereas it is enhanced over North Africa, North America and Northern Eurasia (Fig. 22f,k). It is plausible to assume that the bulk of the multi-year skill is driven by the externally forced changes in rainfall patterns and hydrological cycle (Dong and Sutton, 2015), which is evidently the case over North Africa where ΔACCs relative to *historical* are small or even negative (Fig. 22i,n). However, positive ΔACCs over western North America and Northern Eurasia for all lead ranges suggest

contributions from initialisation. Most ΔACCs for precipitation are not field significant and we cannot preclude that they are a sampling artifact. This is in particular true for the *hindcast-i2* and *hindcast-i1* precipitation skill differences (Fig. 22e,j,o).

For Z500, *hindcast-i1* exhibits positive LY1 skill over most regions (Fig. 23a) that is considerably higher than persistence skill (Fig. 23c). Initialisation benefit related to ENSO is found over the tropical Pacific and also over the Southern Ocean, extratropical North Pacific and subpolar North Atlantic, and over the Canadian Arctic (Fig. 23d). For LY2–5 and LY6–9, the

absolute skill (Fig. 23f,k) is higher than for LY1 and close to saturation in some regions. Most of the LY1 initialisation benefit has disappeared, but some remains over the subpolar North Atlantic and central Siberia (Fig. 23i,n) where the absolute skill of *hindcast-i1* is low and variability is dominated by the North Atlantic Oscillation. The assimilation update of sea ice in *hindcast-i2* slightly benefits the multi-year skill over the Arctic (Fig. 23j,o), offsetting some of the skill degradation caused by initialisation in *hindcast-i1* (Fig. 23i,n).

For SLP, *hindcast-i1* exhibits positive LY1 skill over most regions except Central Eurasia, Brazil and the Southern Ocean (Fig. 24a) and generally performs better than uninitialized prediction (Fig. 24d). The benefit from initialisation is less evident for LY2–5 and LY6–9 (Fig. 24i,n), showing roughly equal proportions of positive and negative ΔACCs. For all lead-year ranges, *hindcast-i1* performs better than persistence in the Pacific and western Atlantic sectors and considerably worse in the African/Indian sector, something that warrants further investigation. The assimilation update of sea ice in *hindcast-i2* again

benefits the multi-year skill over the Arctic (Fig. 24j,o), offsetting some of the skill degradation caused by initialisation in *hindcast-i1* (Fig. 24i,n).



**Figure 23: Prediction skill of 500 hPa geopotential height (Z500). ACC of *hindcast-i1* (a), ΔACC of *hindcast-i1 - assim-i1* (b),**
**990   *hindcast-i1 - persistence* (c), *hindcast-i1 - historical* (d), and *hindcast-i2 - hindcast-i1* (e) for LY1. Middle and right column show the**
**same but for LY2–5 (f–j) and LY6–9 (k–o). Observations use extended ERA5 reanalysis (Hersbach et al., 2020) with coverage**
**1950–2019. Hatched areas are not locally significant, dotted areas are field significant.**





**Figure 24: Prediction skill of sea level pressure (SLP).** ACC of *hindcast-i1* (a), ΔACC of *hindcast-i1 – assim-i1* (b), *hindcast-i1 - persistence* (c), *hindcast-i1 - historical* (d), and *hindcast-i2 - hindcast-i1* (e) for LY1. Middle and right column show the same but for LY2–5 (f–j) and LY6–9 (k-o). Observations use NCEP reanalysis (Kalnay et al., 1996) with coverage 1950–2019. Hatched areas are not locally significant, dotted areas are field significant.





### 3.2.6 Global skill evaluation

We globally summarise first- and multi-year prediction skills by computing ACCs over time and space for the variables assessed in previous sections (Fig. 25). Skills are computed for LY1, LY2–5 and LY6–9 for the two analyses and hindcast products and benchmarked against the uninitialized historical predictions and persistence. The results are not particularly sensitive to grid cell variance normalisation and therefore similar to the globally averaged local (i.e., grid cell) ACC and also qualitatively similar to the mean-square skill score (not shown).


**Figure 25: Global correlation skill for SST (a), 0–300 m temperature (b), sea surface height (c), surface CO₂ flux (d), column-integrated primary production (e,f), 2 m air temperature (g), precipitation (h) and sea level pressure (i). The ACCs are computed over time and space after weighting with the square root of the cell area. The box plots are constructed from 1000 bootstrap ACC realisations. Potential predictability of primary production (f) is referenced to *assim-i1*.**





For SST (Fig. 25a), which is assimilated, the ACCs of *assim-i1* and *assim-i2* exceed 0.8 for all lead year ranges. After assimilation is discontinued, the values drop to 0.5 during the hindcasts. For LY1, this is still higher than and well separated from the 0.4 value of the *historical* experiment, suggesting statistically robust benefit from initialisation for dynamical prediction with NorCPM1. Consistent with better first-year skill in ice covered regions, *hindcast-i2* performs slightly better than *hindcast-i1*, and both hindcast products exhibit marginally higher skill than persistence for LY1 (differences are not

statistically significant). For LY2–5 and LY6–9, the ACCs of the two analyses and initialised hindcast products are very similar to or slightly higher than those for LY1. For multi-year prediction, the ACC of the *historical* experiment is on par with the initialised hindcast products, suggesting a major contribution from the externally forced trend and negligible initialisation benefit. The fact that persistence scores lower than the uninitialized historical experiment reveals that the skill contribution from the externally forced trend is more than what could be expected from a linear anthropogenic climate trend.

For T300 (Fig. 25b), the ACCs of the two analyses are 0.6-0.7 i.e., lower than for SST, presumably due to lower data coverage and higher observation error. Similar as for SST, a clear initialisation benefit manifests for first-year prediction and only a hint of benefit for multi-year prediction. SSH (Fig. 25c) shows initialisation benefit for first-year prediction but signs of detrimental initialisation impact for multi-year prediction. The ACC estimates for SSH are more uncertain than for T300, partly owing to the shorter evaluation period.

Surface $CO_2$ flux (Fig. 25d) and primary production (Fig. 25e) are poorly constrained by the assimilation with the two analyses exhibiting ACCs of 0.2 and below. It is therefore unsurprising that the initialised hindcasts are not skilful and at best show marginal initialization benefit over likewise unskilful uninitialized predictions of *historical*. However, Ilyina et al. (2020) found a predictability of the global air-sea $CO_2$ fluxes of up to 6 years when combining the members of the two hindcast sets, suggesting considerable sensitivity to the chosen biogeochemistry skill metric, spatial averaging, evaluation

period, and ensemble size. In contrast to the hindcasts, the *persistence* skill for $CO_2$ flux exceeds 0.6 for LY1 and 0.3 for LY2–5, and for PP is close to 0.3 for LY1. When using *assim-i1* as observational truth for primary production (Fig. 25f) the system suggests initialisation benefit for all lead years with hindcasts reaching ACCs close to 0.6 for LY1. Inherent issues in the marine ecosystem parameterization to represent realistic variability (Tjiputra et al., 2007; Gharamti et al., 2017) in combination with observational uncertainties are likely causing this discrepancy.

Assimilation in NorCPM1 updates the ocean and sea ice state but does not directly constrain the atmospheric and land states. Nevertheless, the assimilation can improve their prediction to the extent that SST and sea ice control the atmospheric state. The ACCs for surface air temperature (Fig. 25g) resemble those for SST, but are lower, in particular for the two analyses. Land precipitation exhibits ACCs of 0.4 independent of lead year range for the two analyses, and 0.2 for the hindcasts for LY1, suggesting some success in initializing ENSO. Contrary to temperature, the *historical* experiment and *persistence* both

exhibit zero skill for precipitation, both for annual means and multi-year means, despite anthropogenic spinup of the hydrological cycle and other external influences. Sea level pressure (Fig. 25i) behaves differently in that the global ACCs of *persistence*, ranging between 0.3 and 0.5, are consistently higher than those of NorCPM1. Thus, the external forcing seems to have a significant influence on the observed sea level pressure variability, but NorCPM1 fails to capture it. For LY1, the



ACCs of the initialised hindcasts are slightly higher than those of the *historical* experiment, again suggesting skilful

initialisation of ENSO.

**Figure 26: Correlation skill for global means of SST (a), 0–300 m temperature (b), sea surface height (c), surface CO₂ flux (d),
column-integrated primary production (e,f), 2 m air temperature (g) and precipitation (h). The box plots are constructed from**

**1000 bootstrap realisations of the correlations. Potential predictability of primary production (f) is referenced to *assim-i1*. The
plotted correlation range varies for different variables.**

We finally evaluate how well the system constrains the temporal evolutions of global means (Fig. 26). Especially in the

context of climate change attribution, it is of interest whether DA leads to improved representation of global surface

warming, global sea level change and strength of the global hydrological cycle. The initialised hindcasts outperform

*persistence* and *historical* for SST and SAT for LY1. Beyond that, the results show little evidence of initialization benefit,



except a marginal improvement of multi-year mean prediction for the oceanic $CO_2$ flux and a sizable potential predictability benefit for PP. The reanalyses mostly outperform both *persistence* and *historical*. Interestingly the benefit from DA is considerably larger for global precipitation than for global mean SST, possibly indicating a strong control of well constrained tropical—likely ENSO-related—SST variability on large-scale precipitation. DA does not improve the match

with the 16-year short observational record of global sea level, something that warrants further investigation.

## 4 Conclusions

The Norwegian Climate Prediction Model version 1 (NorCPM1) is a new climate prediction system that has contributed with model output to the Decadal Climate Prediction Project as part of the Coupled Model Intercomparison Project phase 6 (CMIP6 DCPP). NorCPM1 combines the Norwegian Earth System Model version 1 (NorESM1) with Ensemble Kalman

Filter (EnKF) anomaly-assimilation of sea surface temperature and hydrographic profile observations. This paper provides a description and evaluation of NorCPM1.

Compared to other dynamical climate prediction systems, NorCPM1 distinguishes itself by its EnKF anomaly assimilation that performs cross-component ocean-to-sea ice updates and is optimised for an ocean vertical density coordinate. The EnKF scheme makes optimal use of the observations by also updating unobserved variables using state-dependent relations from

the model's simulation ensemble. The use of these relations further minimizes shock by ensuring that all variables are updated consistently, to the extent the system behaves linearly. Through performing EnKF anomaly assimilation and accounting for measurement and representation errors in the observations, NorCPM1 aims at synchronising internal variability in a targeted and gentle manner to provide a reliable system (i.e., where the ensemble spread reflects the true internal variability error) that is mostly free of detrimental prediction shock. While on grid-scale this allows certain

mismatch between model and observations, our evaluation of the assimilation experiments shows that the approach accurately synchronises the large-scale variability modes (such as ENSO, PDO, and SPG strength) that are likely to carry multi-year predictability.

The paper assessed the performance of the ESM component of the prediction system. Upgrades of the external forcings from CMIP5 to CMIP6 and minor code changes have only a minor impact on the model's climate representation relative to the

original NorESM1, which contributed to CMIP5. Spatial biases in key climate variables have mostly remained the same, as has the global climate response to external forcings. The conditional bias is hence largely unaltered relative to previous NorCPM configurations. Noteworthy biases are a 50 % too strong Atlantic meridional overturning circulation, excessive Arctic sea ice with cold adjacent continents, warm surface biases in the subpolar North Atlantic and Southern Ocean that are mirrored by cold biases at lower latitudes. In turn, the model's ENSO characteristics and its historical global warming

compare favourably to observations.

The paper assessed the performance of the assimilation capability with two re-analyses products that have been contributed to CMIP6 DCPP. The anomaly assimilation of NorCPM1 does not show any detrimental effects on the climatology and



generally reduces the RMSE of both observed and unobserved state variables (unobserved means not part of observation types that are assimilated) in the assimilation experiments relative to the historical experiment without assimilation. The application of cross-component anomaly assimilation reduces a positive bias in Arctic sea ice thickness and improves synchronisation of sea ice variability and variability of other climate variables, such as Southern Ocean sea surface height.

A challenge unique to anomaly assimilation is how to best construct the anomalies. The paper presents two 30-member climate analysis products that use different reference periods. The choice of reference period has limited impact on their correlation scores with observations, but it has significant impact on mean and long-term trends, e.g., in Atlantic meridional overturning circulation strength and meridional ocean heat transport. Future NorCPM development efforts will explore more sophisticated ways of designing climate anomalies, e.g., following Chikamoto et al. (2019), addressing important issues such as conditional bias and separation of internal variability versus externally forced signals in observations.

The assimilation shows limited success in synchronising variability in ocean biogeochemistry variables like net primary production or air-sea $CO_2$ flux. This result contrasts findings of a perfect model study (Fransner et al., 2020) with the ESM component of NorCPM1 that suggests strong control of the physical state on interannual ocean biogeochemistry variability. Imperfect synchronisation of physical variability, short evaluation periods, errors in observations, and errors in the model representation of ocean biogeochemistry and its interaction with physical processes can contribute to this discrepancy.

The paper assessed the performance of the system to produce first- and multi-year climate predictions. We found robust initialisation benefits for first-year prediction across a range of climate variables that at least is partly related to skilful synchronisation of ENSO variability. Predictability of sea ice extends into the second year in the hindcast product initialised from a reanalysis that more strongly constraints the sea ice state.

While the externally forced trend leads to significant multi-year prediction skill, our evaluation provides limited evidence for robust initialisation benefits on multi-year time scales but also little indication for detrimental effects from initialisation. Multi-year initialisation benefit is mainly confined to SPNA in NorCPM1, where it largely offsets negative skill in unitialized predictions, and leads to modest absolute skill that is significantly lower than the skill from non-dynamical prediction such as persistence forecast. The comparison of two differently initialised hindcast products reveals high sensitivity of the AMOC to the details of the initialisation approach with considerable impact on SPNA temperatures, such as shift in mean state and long-term trend and hindcast drift behaviour. Notwithstanding that both products struggle in predicting the circulation evolution, it indicates the potential for improving SPNA temperature predictions by improving initialisation of hydrographic anomalies that condition the evolution of the large-scale ocean circulation. To realise the full potential, however, would require a model representation of the circulation with realistic mean state, variability and sensitivity to external forcing. Work has started to upgrade NorCPM's model component to NorESM2-MM (Seland et al., 2020) which features a more realistic AMOC and overall reduced climate biases compared to NorESM1. Lead-dependent drift correction removes much of the differences between the two products (including a strong forecast drift in sea ice thickness present in one of the products) and therefore has merits also for anomaly-initialised predictions, in particular if model output is intended as input for climate impact studies.





The initialisation of the physical model states does not robustly benefit ocean biogeochemistry predictions in NorCPM1. This is unsurprising given the aforementioned poor skill of the re-analyses used for hindcast initialisation. Thus, improving and understanding the lack of skill in the re-analyses is paramount to improving NorCPM's ocean biogeochemistry capability.


We found robust transfer of initialisation skill benefit to atmosphere and land for first-year prediction. As current climate models tend to underestimate atmospheric signal-to-noise ratios, more hindcast simulation members are expected to increase first-year skill and enable detection of multi-year signals (Scaife and Smith, 2018; Smith et al., 2020).

In summary, we found demonstrable benefits from initialisation for climate prediction with NorCPM1. The initialisation is virtually free of detrimental effects. At this stage, NorCPM1 primarily serves as a research tool. Based on the forecast quality evaluation presented in this paper, further development is needed to reach multi-year prediction skill at a societally useful level that makes the system more fit for operational use. To this end, the evaluation in this paper will serve as a benchmark for further NorCPM development, such as upgrades to the ESM component and refinements to the assimilation approach with extension to all model components. Deficiencies of NorCPM1 skill identified here will guide future research and model development. The system has demonstrated promising seasonal prediction capabilities (Wang et al., 2019; Kimmritz et al., 2019) and may already contribute to skilful multi-year climate prediction with societal application in a multi-model framework (Smith et al., 2020).



## Appendix A – Choice of DA scheme

There are multiple ways to initialise hindcasts, such as initialisation from existing reanalysis products produced with an independent system (e.g., Chikamoto et al., 2019) or initialisation of the ocean component by running it uncoupled, forced with an atmospheric reanalysis product (Yeager et al., 2018). In NorCPM1, the hindcasts are initialised from a reanalysis produced with the same ESM that assimilates ocean observations with the Ensemble Kalman filter (EnKF; Evensen, 2003). The advantage of using the same ESM is that it avoids initialisation adjustment that occurs when changing the model. The EnKF is an advanced flow dependent data assimilation method where the multivariate corrections are based on a set of observations, their uncertainty and the ensemble of model realisation produced by a Monte Carlo integration from the previous analysis step. Counillon et al. (2016) showed that the upper ocean heat content in the Equatorial and North Pacific, the north Atlantic subpolar gyre region and the Nordic Seas can be well constrained by assimilating SST anomalies with the EnKF. In particular, the vertical covariance shows a pronounced seasonal and decadal variability that highlights the benefit of flow-dependent data assimilation. In NorCPM1 covariances in the ocean are formulated in isopycnal coordinates (the native vertical coordinate of the ocean model), which allows for deeper influence of the assimilated surface observations than when formulating them in standard depth-coordinate (Counillon et al., 2016).




Up to now, climate prediction systems have predominantly assimilated data independently in their respective components, an approach referred to as weakly coupled data assimilation (WCDA; Penny and Hamill, 2017). The other model components



adjust to these individual changes dynamically in between the assimilation cycles. Allowing the assimilation to update across
model components is expected to outperform WCDA because it would enhance dynamical consistency of the initial
condition and expand the influence of the observations across its own component (strongly coupled data assimilation,
SCDA; Penny and Hamill, 2017, 2019). The climate system includes complex, coupled phenomena over wide, separated
spatial and temporal scales of the Earth System components (atmosphere, ocean, land surface, cryosphere). DA procedures,
on the other hand, are mostly designed to deal with a single dominant scale of motion or under the assumption of weak
coupling (Laloyaux et al., 2015, Sun et al., 2020). Joint SCDA of ocean and sea ice has been successful with flow dependent
DA methods such as the EnKF. The scale separation between ocean and sea ice is not as pronounced as between ocean and
atmosphere. The application of flow dependent covariance can handle well the anisotropy and sign reversal of the covariance
at the sea ice front (Lisæter et al., 2003; Sakov et al., 2012) and the update of the multicategory sea ice state (Massonnet et
al., 2015; Kimmritz et al., 2018). Application of the methods has since also been tested successfully in a fully coupled ESM
(Kimmritz et al., 2018) and used for seasonal prediction of Arctic sea ice (Kimmritz et al., 2019). A full SCDA of the ESM
is a more challenging task because of the separation of spatial and temporal scales among atmosphere and ocean. There have
been many advances both theoretically (Lu et al., 2015a; Smith et al., 2015; Tardif et al., 2015; Sluka et al., 2016; Penny et
al. 2017) and on application e.g., the CERA reanalysis (Laloyaux et al., 2016) but no system is yet at the stage of achieving a
full SCDA. For interannual-to-decadal time scale, the largest part of climate predictability resides in the ocean and sea ice
(e.g., Mariotti et al. 2018). Making use of the rich atmospheric observation network will be explored in future NorCPM
versions as it can further improve the initialisation of the slow modes of variability in the ocean where observations are
sparse and generally enhance the consistency of the system.

Climate models have strong biases that are in some places larger than the internal variability (Richter et al., 2014). There are
two common strategies in the climate prediction communities to handle bias: full-field assimilation requiring a subsequent
post-processing to account for the model adjustment back to its own attractor or anomaly assimilation where the observed
anomaly (calculated relative to a reference climatology) are imposed on a biased model climatology (Weber et al., 2015).
Both methods have their advantages and disadvantages. NorCPM1 uses anomaly assimilation because full-field assimilation
is problematic with ensemble DA (Dee, 2005): As models are attracted to their biased climatological states, the model bias in
the observed variables is propagated to the non-observed variables by the multivariate covariance matrix, which leads to a
slow degradation of the system through the consecutive assimilation cycle. A challenge when defining a climatological
reference is to ensure that the climatological reference is accurate and representative of the same variability between the
model and data. Estimating an accurate climatology for observations becomes problematic in regions where observations are
very sparse, limiting the possible span of a reliable climatological period. Furthermore, while it is usually possible for the
model to nullify the internal variability by averaging different ensemble members starting from different initial conditions,
there is only a single realisation of the truth and one must ensure that the climatological period of the observation is long
enough to cancel out internal variability. Finally, it should be added that anomaly assimilation only addresses climatological
biases and conditional biases such as in the variability and in the forced trends.



An emerging number of climate prediction models include ocean biogeochemistry (e.g., Séférian et al. 2014; Li et al., 2016; Lovenduski et al., 2019; Park et al., 2019). Due to technical challenges with implementing ocean biogeochemistry in DA systems related to data sparsity and the non-Gaussian behaviour of many biogeochemical tracers, assimilation of biogeochemical observations is commonly not applied in these models (e.g., Park et al 2019). Instead, the ocean biogeochemistry is treated passively. This has been shown to constrain the biogeochemical variability relatively well (Seferian et al, 2014; Li et al., 2019; Park et al., 2019). There are, however, problems related to the update of physics that introduces artificial mixing between surface and deep waters, leading to excessive surface nutrient concentrations and primary production, especially in the tropics (While et al., 2010; Park et al., 2018). Skilful near-time predictions of 4–7 years of air–sea $CO_2$ exchange (Li et al., 2016, 2019), a couple of years for chlorophyll (Park et al., 2019) and 2–5 years for net primary production (NPP, Seferian et al. 2014) have been achieved by this passive initialization of ocean biogeochemistry. Fransner et al. (2020) showed, in a perfect model framework, that the initial state of ocean biogeochemistry has little impact on the prediction skill beyond LY1, and their work suggested that assimilation of biogeochemical tracers would only give a marginal improvement in the predictive skill of ocean biogeochemistry.

**Appendix B – Skill scores and significance testing**

Following Goddard et al. (2013), we use the anomaly correlation coefficient (ACC) for assessing hindcast and reanalysis performance. We use ΔACC score differences for comparing our reanalysis and hindcast products and for benchmarking against uninitialized predictions and persistence forecast. As in Goddard et al. (2013), we consider lead-year 1 (LY1), lead-year 2-5 (LY2–5) and lead-year 6-9 (LY6–9) forecast ranges using multi-year averages. For example, if a hindcast is initialised in October 1960 then LY1 corresponds to the average of 1961, i.e., the following calendar year.

If the temporal coverage of the observations is shorter than that of the model output, we maximize the use of observations in the ACC computation. For example, if the observations start in 1993 then the ACC computation for LY6–9 will use hindcasts starting at the end of 1983 and later. Consequently, the start dates used in the ACC computation may differ for the different forecast ranges, while the evaluation period is fixed except in the persistence forecast. The LY1 persistence forecast uses the observational average of the previous year, while the LY2–5 and LY6–9 persistence forecasts use the average over the four previous years. This is done because we found the effect of temporal filtering to outweigh the shift towards older observations, resulting in persistence skills consistently higher than if using the last month or last year instead.

Prior to the ACC computation, we interpolate model and observational data to a common, regular 5°×5° grid if not stated otherwise. We do not remove the linear trend or other estimates of the forced response, except when evaluating surface carbon flux. When comparing ACCs of hindcasts (which comprise 10 simulation members) with uninitialized predictions, we only use the first 10 members of *historical* because we want to isolate the benefit of initialisation without confounding it with the effect of ensemble size on the accuracy of the externally forced trend estimate.





We test local and field significance of skill scores and score differences following Yeager et al. (2018). We consider a score
locally significant if the associated p-value (i.e., probability for producing a random score equal or higher than the score
tested) is below $\alpha_{local}:=0.1$ (i.e., 90% confidence). Regions that fail the local significance test are marked with slash / on the
skill score maps (e.g., Fig. 7). We derive the p-values by means of resampling the original data that is interpolated to the
common grid. For each obtained skill score we construct 4,000 bootstrapped scores that capture random uncertainty
stemming from temporal sampling and from having a limited ensemble size. Using the moving block bootstrapping
technique, we resample the data (pairwise model-observation sampling with replacement) in 5-year blocks that may start in
any year but not in the last four years to account for temporal autocorrelation. The blocks are concatenated, and the last
block is truncated such that the bootstrapped time series has the same length as the original series. Additionally, we resample
(with replacement) the ensemble members used in the computation of the ensemble means. While the combination of
members varies between different bootstrapped time series, we keep it fixed within each series. We test significance for both
positive and negative scores. Following Goddard et al. 2013, we estimate the p-value for a particular skill score as the
fraction of bootstrapped scores with opposite sign of that of the score tested (e.g., if the original score is positive and 200 out
of the 4,000 bootstrapped scores are negative then we determine p as 200/4,000 = 0.05). The rationale is to utilise the spread
information from the bootstrapped distribution to calculate the probability for obtaining a score equal or higher than the
score tested, under the null hypothesis that the true score is zero. We verified the bootstrap estimation of p-values on a large
set of artificially constructed series with known true correlation and found good agreement with Monte Carlo estimated p-
values, with $r(p_{bootstrap}, p_{MonteCarlo}) > 0.95$.

Local significance information has particular utility if considering a single location of interest and if the choice of this
location is not informed by the spatial score distribution. Explorative analyses, however, often simultaneously consider
multiple locations of interest and make the selection of locations dependent on the spatial score distribution as they tend to
focus on regions with the most extreme scores. In such cases the use of field significance is more meaningful. Like Yeager et
al. 2018, we test field significance using the false discovery rate (FDR) approach following Wilks (2006, 2016), which has
the practical advantage that it reuses the p-values from the local significance test. The FDR algorithm determines $p_{FDR}$ such
that the false discovery rate in the region where $p < p_{FDR}$ (locations marked with dot · on the maps) becomes approximately
equal to a target FDR of 10%. The value of $p_{FDR}$, stated on all ACC plots, is computed from Equation B1 where N is the
number of p-values, p(i) is the i-th sorted p-value and $\alpha_{FDR}$ a parameter that controls the FDR.

$$p_{FDR} = max_{i=1,...,N}\left[p_{(i)}: p_{(i)} \leq (i/N)\alpha_{FDR}\right], \tag{B1}$$

If $p_{FDR}$ exists, then the test also rejects the global null-hypothesis that the true scores are zero everywhere at 90% confidence
level. Assuming moderate to strong spatial correlation (Wilks 2006), we set $\alpha_{FDR}:=2\alpha_{global}$ and $\alpha_{global}:=\alpha_{local}=0.1$. Consistent
with intuition, $p_{FDR}$ tends to be close to $p_{local}$ if the majority of points are locally significant, while $p_{FDR} \ll p_{local}$ if only few
points are locally significant. In rare situations $p_{FDR}$ can become larger than $p_{local}$ (due to $\alpha_{FDR} > \alpha_{local}$) with the consequence
that scores can be field significant without being locally significant. We consider this an artefact of the ad-hoc adjustment of
$\alpha_{FDR}$ for spatial correlation, and we set $p_{FDR}:=p_{local}$ in such case.

The above significance testing does not account for observational error. Nowadays, observational reanalyses routinely provide ensemble products that span observational uncertainty. While beyond the scope of this study, future skill evaluations should explore ways of utilizing this ensemble information in local and field significance testing. The addition of observational uncertainty should generally lower the p-values, leading to stricter testing.




**Figure C1: Ocean mean-state biases of NorCPM1 and NorESM1-ME. Observations (left row) use surface temperature (SST) from 1971–2000 (Reyn_SmithOIv2; Reynolds et al., 2001), 0–300 m temperature (T300) and salinity (S300) from 1980–2010 (EN4.2.1; Good et al., 2013), and sea level anomaly (SLA) from 1993–2018 (ARMOR3D; Larnicol et al., 2006). Matching time periods are used for computing the climatological biases of NorCPM1 (middle row) and NorESM1-ME (right row). Area-weighted global-mean bias, RMSE, and global-mean bias adjusted RMSE are stated above the panels.**




### Appendix C – Earth system model evaluation

In this section, we evaluated climate mean and variability characteristics, model stability and sensitivity to external forcings of the ESM component of NorCPM1 in absence of data assimilation. Knowledge of the model climate performance can inform attribution of inter-model prediction differences and generally the interpretation of prediction results. The mean climate and variability of the original NorESM1 has been evaluated in previous studies (e.g., Bentsen et al., 2013; Iversen et al., 2013; Tjiputra et al., 2013; Kirkevåg et al., 2013). The updates to the code and forcings have not changed the model characteristics to an extent that would warrant a complete re-evaluation and the evaluation here is therefore kept brief.

Figures C1 and C2 show the climatological biases of NorCPM1 and NorESM1-ME for the ocean and atmosphere, respectively, based on the late 20$^{th}$ Century state of *historical*. NorCPM1's simulated sea surface temperature (SST) is 0.7 K too cold globally, with larger spatial biases (Fig. C1b). The tropical Pacific is slightly too cold, and the subtropics and polar regions are generally too cold by several K. In turn, the Southern Ocean, the subpolar North Atlantic (SPNA) except south of Greenland, coastal upwelling regions, and western boundary current extensions are generally too warm. Upper ocean temperature (T300, 0-300 m averaged) shows a similar pattern (Fig. C1e), indicating that the surface biases largely extend below the surface. For parts of the central North Pacific and Indian Ocean, however, the sign of the T300 biases is opposite compared to the SST biases, indicating a contribution from errors in vertical mixing. The warm bias in the SPNA is accompanied by a negative sea surface height (SSH) bias (Fig. C1k) in excess of 1 m and a positive upper ocean salt (S300, 0-300 m averaged) bias (Fig. C1h) that both extend into the Arctic. This salt bias is mirrored by a fresh bias south of the equator, suggesting large-scale ocean meridional circulation errors as one cause. The simulated Atlantic Meridional Overturning Circulation is indeed too vigorous (Fig. C3c), with a mean strength of 34 Sv versus 19 Sv observational estimate (Cunningham et al., 2007) evaluated at 26.5 °N, and the northward oceanic heat transport in the Atlantic Ocean is too large (Fig. C3b). It should be noted, however, that a similar configuration of NorESM1 featuring a weaker AMOC produces similar temperature and heat transport biases (Guo et al., 2019). Surface air temperature (SAT, evaluated at 2 m) biases (Fig. C2b) match those of SST except over ice-covered regions where the model simulates cold biases in excess of 5 K. The Arctic cold bias extends over much of the North American and Eurasian continents, including North Africa and East Asia. Warm biases are simulated over land adjacent to upwelling coastal upwelling regions and over central Asia, whereas only small temperature biases are simulated over most of Europe and central and eastern part of South America. The Arctic cold bias is accompanied by a negative 500 hPa geopotential height (Z500) bias that is strongest over the Chukchi Sea and Nordic Seas and opposed by positive Z500 biases over the North Pacific and subtropical North Atlantic (Fig. C2k), resulting in a too zonal storm track over the North Atlantic. Similarly, cold surface biases over Antarctica and warm biases over the Southern Ocean are accompanied by negative and positive Z500 biases. The precipitation biases over the ocean reveal the presence of a double inter-tropical convergence zone (Fig. C2h). Over land (Fig. C2k), wet biases are present over much of Africa, the Tibetan plateau, Australia and westerns parts of America, while dry biases are present over the remainder of America, North Africa, India and Northern Eurasia.



**Figure C2: Atmosphere mean-state biases of NorCPM1 and NorESM1-ME. Observations (left row) use 2 m air temperature (SAT), geopotential height at 500 hPa (Z500) and global precipitation from 1979–2019 (ERA5; Hersbach et al., 2020), and precipitation over land from 1979–2018 (CRU TS v4.03; Harris et al., 2020). Matching time periods are used for computing the climatological biases of NorCPM1 (middle row) and NorESM1-ME (right row). Area-weighted global-mean bias, RMSE, and global-mean bias adjusted RMSE are stated above the panels.**


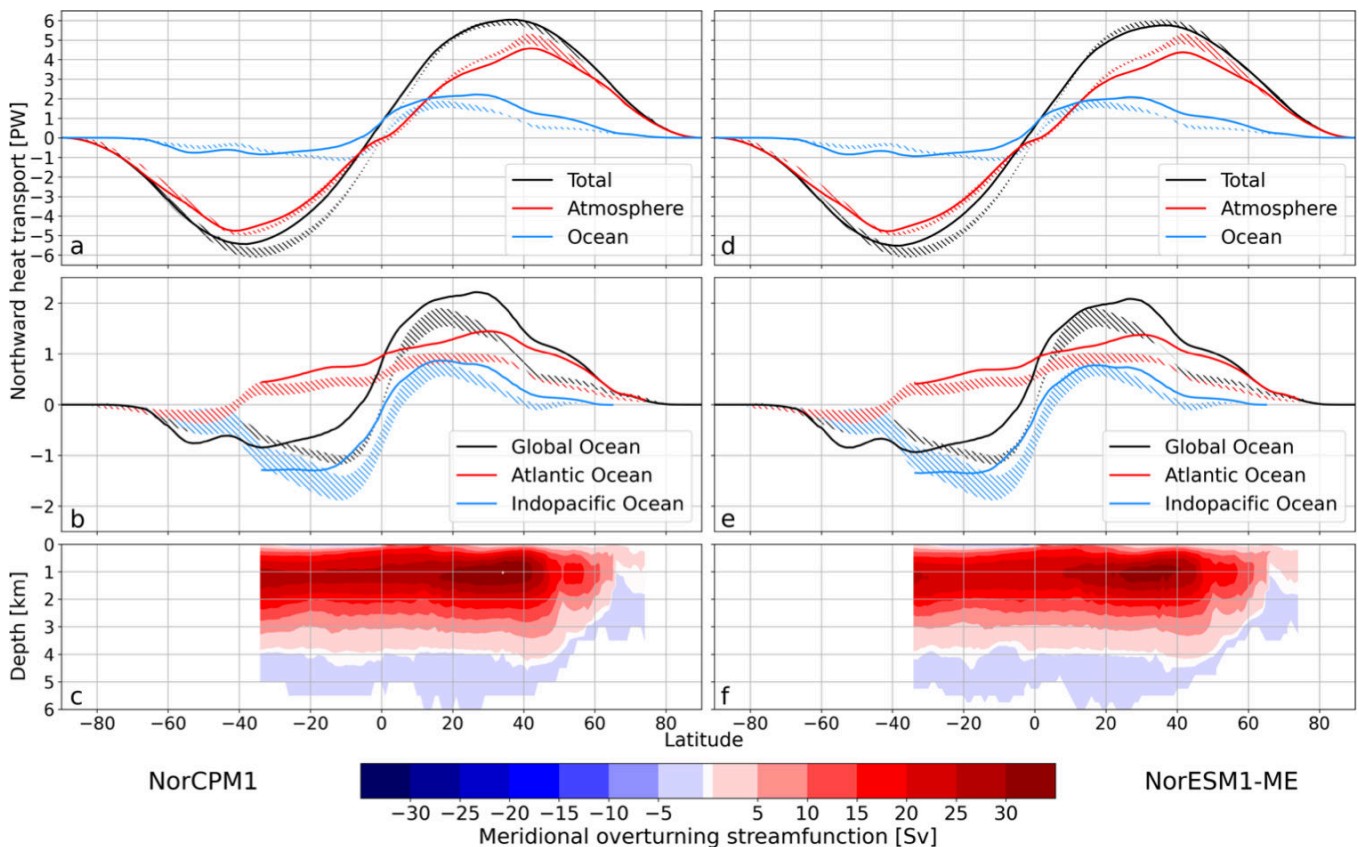

**Figure C3: Atlantic meridional transports of heat and mass. Simulated historical (1976–2005 mean) northward heat transport of NorCPM1 for (a) global atmosphere, ocean, and total, and (b) global ocean, its decomposed Atlantic Ocean and Pacific and Indian oceans, and Atlantic meridional overturning streamfunction (c). The panels (d–f) show the same for NorESM1-ME. The corresponding hatched areas with uncertainties are estimates from Fasullo and Trenberth (2008). In the model estimation, the ocean heat transport is calculated directly from the ocean model, and the atmospheric heat transport is derived by meridional integration of the difference between zonally integrated net TOA and surface heat fluxes.**

While the climate mean-state biases are overall similar to those presented in the original NorESM1 description paper (Bentsen et al., 2013), Figures C1–C3 reveal signs of degradation for NorCPM1, with global mean biases and RMSEs being slightly larger than for NorESM1-ME. Most notable, the Arctic cold and negative Z500 biases are exacerbated. The bias exacerbation is likely due to erroneous transient land-cryosphere surface conditions in NorCPM1, as revealed from additional sensitivity simulations. Each sensitivity simulation has one NorCPM1's forcing update or model modification reverted back to the NorESM1-ME CMIP5 system; these were (1) aerosol code bug, (2) atmospheric cloud tuning, (3) sea ice albedo tuning, (4) land surface forcing, (5) ozone forcing and (6) stratospheric volcanic forcing. We integrated the simulations for 120 years with 1850-level preindustrial forcings, followed by 170 years with transient historical forcings. We evaluate the preindustrial (year-1850) state by averaging the first 120 simulation years and the modern (year-2000) state by





averaging the period corresponding to the forcing years 1879-2019. Figure C4 shows the results for SAT and Z500 averaged over the Arctic region north of 60 °N. Only the simulation with land-use change reverted back to CMIP5 (purple lines) shows a SAT warming trend and Z500 rise comparable to the NorESM1-ME CMIP5 system. The impacts of the other changes are comparably small—the aerosol code bug leads to some high-latitude warming, whereas reverting to CMIP5 ozone forcing or sea ice tuning leads to some cooling—and do not significantly affect the trend. Rectifying NorCPM1's land

surface conditions will be a future development priority. A comparison with results from previous NorCPM versions suggests that the increased biases have a measurable detrimental effect on high latitude prediction skill (Passos et al, in preparation).

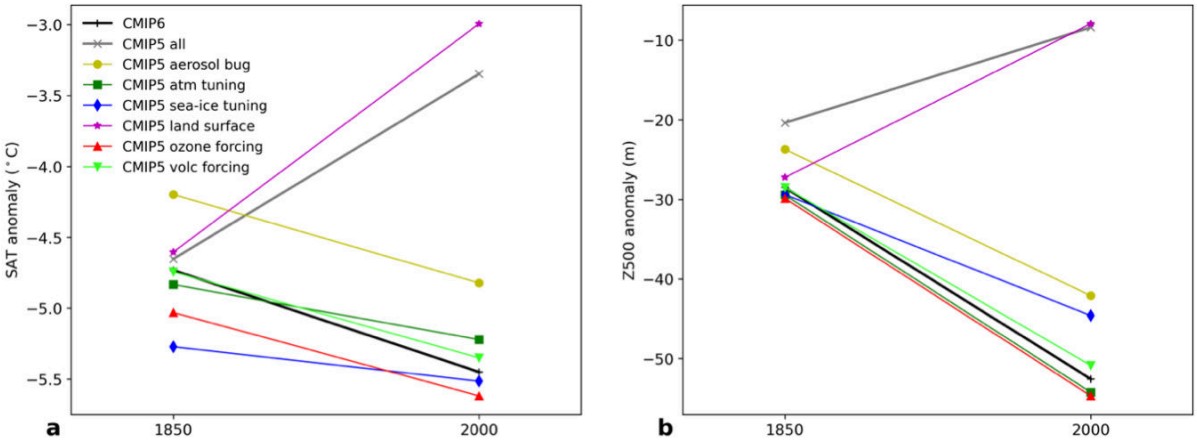

**Figure C4: Arctic SAT (a) and Z500 (b) biases for preindustrial and modern time from supporting sensitivity experiments (see text**

**for details). The sensitivity experiments (colours) are based on NorCPM1 (black), but with individual updates reverted back to NorESM1-ME (grey). Note the differences between a sensitivity experiment to NorCPM1 is opposite to the effect of the specific upgrade (e.g., the effect of the aerosol bug fix is a cooling, not warming).**

Global mean surface temperature (GMST) is commonly used for assessing climate stability and global climate sensitivity (Fig. C5). During the first 100 years, the strong volcanic activity of the late 19[th] and early 20[th] century cools *historical* (blue curve) relative to *piControl*. Their difference reduces in the first half of the 20[th] century and anthropogenic warming emerges

in the 70's when *historical* crosses *piControl*. At the start of the 21[st] century, *historical* is 0.7 K above its 1850–1900 mean and 0.5 K above *piControl*. While these values are lower than the 0.8 K observational estimate (e.g., Peters, 2016), the simulated warming towards the end of the 20[th] century matches the observed trend well and overestimates it during the beginning of the 21[st] century (blue versus black).






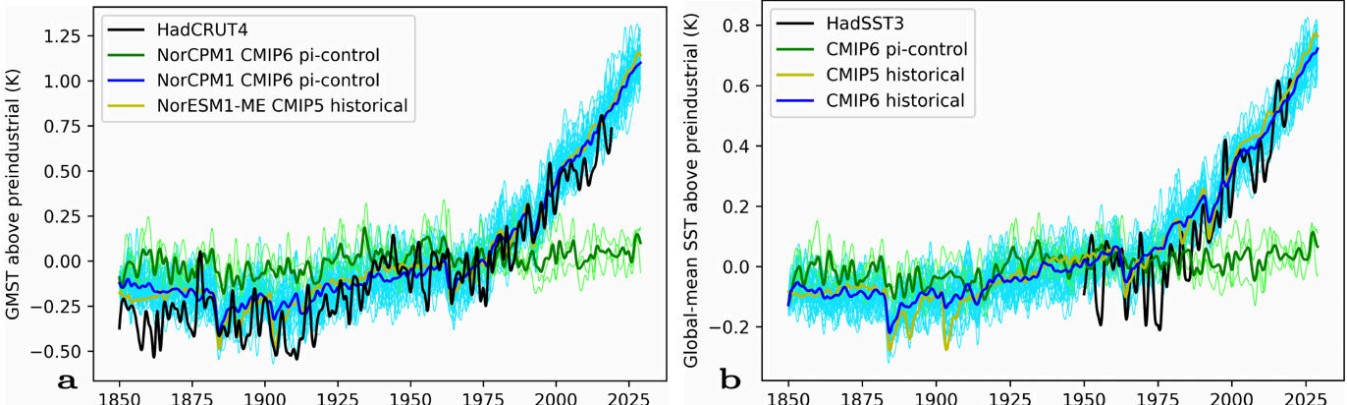

**Figure C5: Global Mean SAT (left) and SST (right) evolutions. Thick lines denote ensemble means, thin lines individual members. CMIP6 data are shifted relative to the mean of CMIP6 *pi-control*. CMIP5 *historical* (30-member mean) is shifted to match the mean of CMIP6 *historical*. Observational estimate derived from HadCRUT4 (Morice et al., 2012) and HadSST 3.1.1 (Kennedy et**
**al., 2011).**

The forced GMST evolution of NorCPM1 behaves very similarly to the one of the CMIP5 NorESM1-ME system (blue versus yellow) despite the model and forcing upgrades. Deviations are most notable in the volcanic climate response at the start of the 20th century and are likely driven by changes in forcing rather than updates to the model. That updates to the model have a limited effect on the model's climate sensitivity is further confirmed with estimates of the effective climate
sensitivity (ECS) to a doubling of atmospheric $CO_2$ relative to pre-industrial level. We estimate the ECS from the output of *abrupt4XCO2* using the linear regression method of Gregory et al. (2004). The 3.06 K estimate for NorCPM1 is close to the 2.94 K for NorESM1-ME. Both estimates derived from single simulations and are statistically indistinguishable given the large sampling uncertainty that results in 5–95% bootstrap confidence intervals of 2.99–3.19 K and 2.86–3.09 K for NorCPM1 and NorESM1-ME, respectively. These estimates are at the low end but well within the 2.1–4.7 K CMIP5
(Andrews et al., 2012) and 1.8–5.6 CMIP6 (Zelinka et al., 2020) model ranges and also within the narrowed observation-based range of 2.6–3.9 K (Sherwood et al., 2020). *piControl* (green curve) exhibits a small positive GMST long-term drift of 0.06 K century$^{-1}$ that is caused by a 0.13 W m$^{-1}$ net radiative transport at the top of atmosphere but is inconsequential for the sensitivity evaluation.

The model produces reasonably realistic ENSO variability in terms of frequencies and variance as diagnosed from the
monthly Niño-3.4 index of *historical* (Fig. C6, C7). The simulated ENSO shows elevated power in the 2–5-year frequency band, similar to observations. The model overestimates the standard-deviation by 20 %.



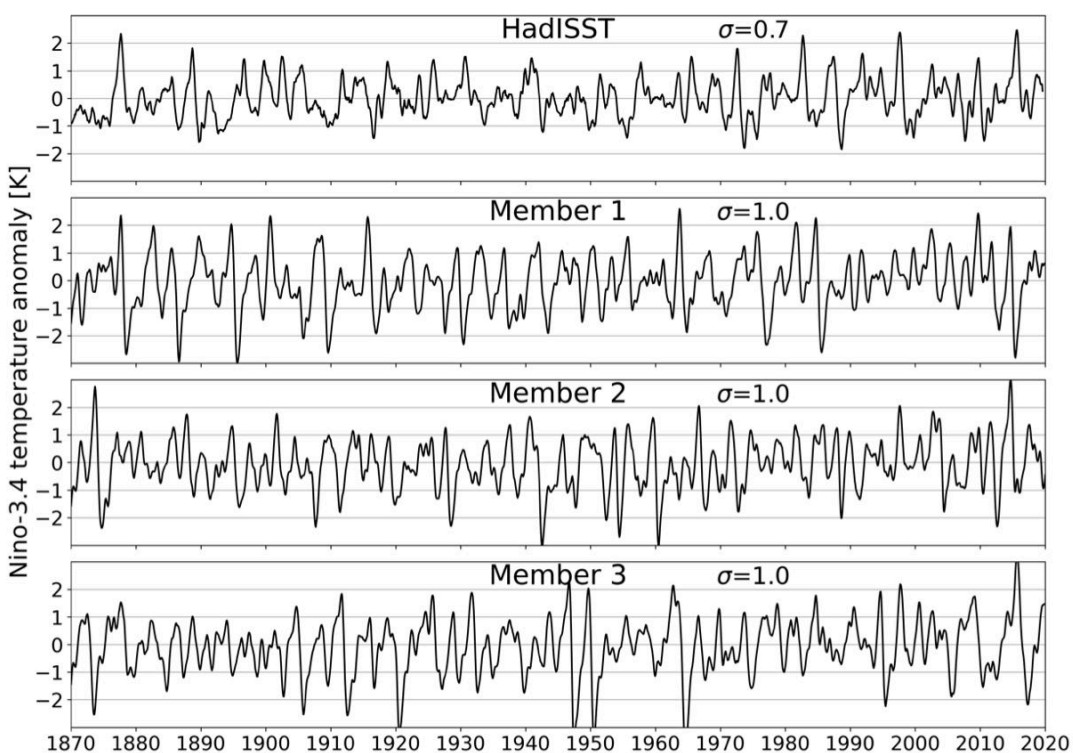

**Figure C6: Niño-3.4 monthly index for HadISST (Rayner et al., 2003) and the first three members of NorCPM1's** *historical*
**experiment.**


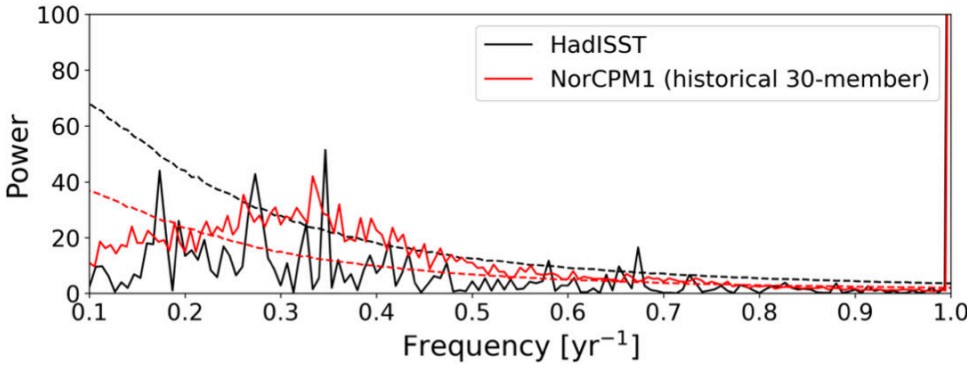

**Figure C7: Niño-3.4 frequency spectrum. The observed estimate (black) uses monthly HadISST data (Rayner et al., 2003) over the**
**period 1870-2019. The model estimate (red) uses monthly data from NorCPM1's** *historical* **experiment and represents the average**
**of the spectra from the 30 members. The stippled lines show the 90th percentile from synthetic red noise time series with lag-1**
**autocorrelation from the HadISST data. Differences in the two red noise estimates are due to the model data having 30 times as**
**many data points compared to the observations.**

**Appendix D – Supporting reanalysis and hindcast evaluation**

The different AMOC evolutions in *assim-i1* versus *assim-i2* indicate that NorCPM1's AMOC initialisation is sensitive to the climatology used for defining anomalies and/or to the assimilation update of sea ice. To further assess the relative

importance of the two factors, we reran the *assim-i1* reanalysis, but this time using the climatological period 1950-2010 that was used in *assim-i2*. Before 1970, the AMOC evolution of the new reanalysis stays close to the strong AMOC evolution of *assim-i2* (Fig. D1). After 1970, however, it separates from assim-i2 and stays somewhere between *assim-i1* and *assim-i2*. This suggests that the assimilation update of sea ice also plays a role in strengthening the AMOC in addition to the climatology, possibly by making the high-latitude oceans more saline (Fig. 3f).

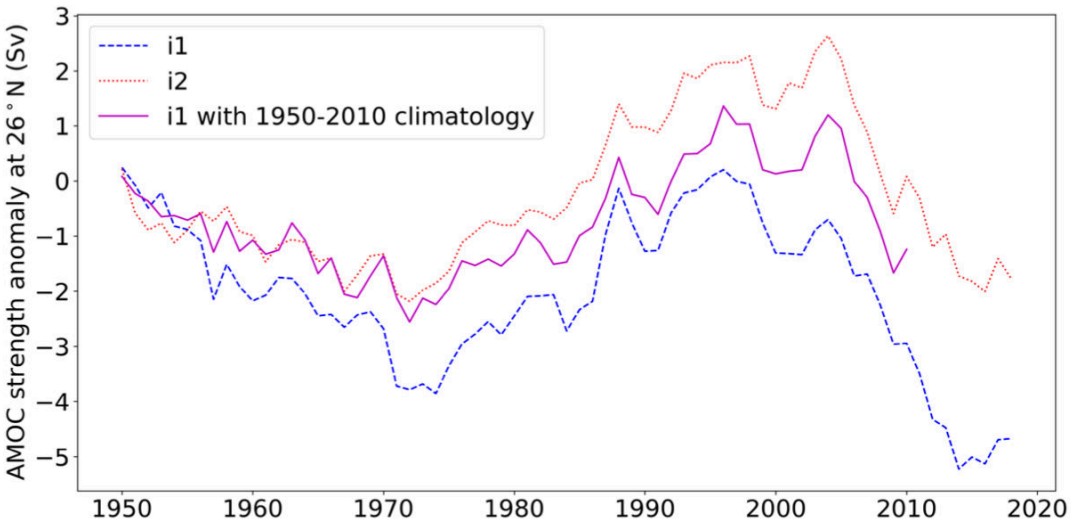


**Figure D1: Attribution of AMOC differences to climatology versus sea ice assimilation. Annual-mean AMOC strength at 26 °N from the ensemble means of *assim-i1* (dotted red), *assim-i1b* (solid purple) and *assim-i2* (stippled blue). The *assim-i1b* reanalysis experiment is identical to *assim-i1* but uses the climatological period 1950–2010 (same as in *assim-i2*) for computing assimilation anomalies.**

Skilful ENSO prediction is a likely driver of NorCPM1's robust initialisation benefit for first-year climate prediction. Figure D2 shows ACCs for the Nino-3.4 index over lead months. The ACCs of the initialised hindcasts stay above 0.8 from November until April, similar to *persistence* but considerably higher than the skill of *historical*. Following, the skill drops sharply, but in contrast to *persistence* the skill of the initialised hindcasts levels out and stays at 0.4-0.5 during summer and autumn, which is slightly above the skill of *historical*. During the subsequent winter and spring, the skill of the initialised

hindcasts drops slightly, but much less than that of *historical*, suggesting a reemergence of initialisation benefit during the second spring. NorCPM1's results for November initialisation agree well with those Wang et al. (2019) obtained with NorCPM assimilating only SST observations (their Fig. 10). For initialization in other seasons, they show that NorCPM markedly outperforms *persistence*.





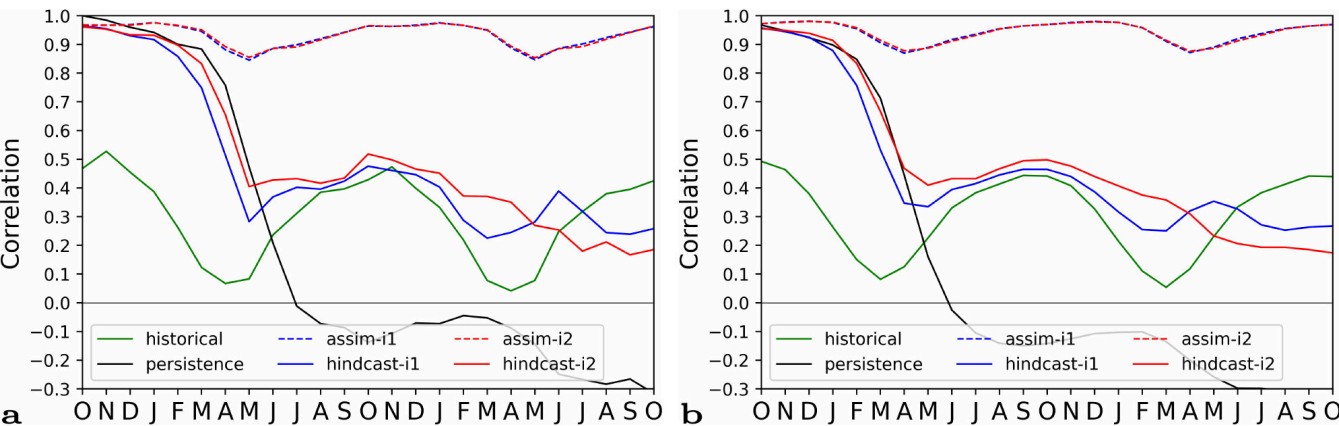

**Figure D2: Prediction skill for Niño-3.4 index. (a) ACC over lead months with observations from HadISST (Rayner et al., 2003) for the period 1950–2019. The observed October value is used as persistence forecast. (b) As a, but applying a 3-month average prior to ACC computation and using the August–October average as persistence forecast.**

Skilful SST prediction is key for skilful prediction skill in atmosphere and over land. However, the consideration of ocean heat content can provide additional clues on its relation to ocean thermal inertia and ocean dynamics. Because the ocean

subsurface state is less directly influenced by unpredictable atmospheric or coupled variability, one would expect to see higher skill there.

The ACCs for 0-300 m temperature (Fig. D3) have a similar pattern as those of SST (Fig. 12) but tend to be smaller and patchier. A caveat is that our observational reference for subsurface temperature is less reliable than that for SST over the 1960-2018 evaluation period. The skill increment from initialisation in SST appears related to the upper ocean heat content

(Fig 12d,i,n; Fig D3d,i,n).

Contrary to temperature, advected salinity signals are not subject to atmospheric feedback-driven surface damping. Predictions of upper ocean salinity can therefore inform temperature predictions by providing additional insights. The ACCs for 0-300 m salinity (Fig. D4) have generally a similar pattern but are lower than those of temperature. Lower coverage of salinity compared to temperature profiles and a weak DA constraint of mixed layer salinity may partly explain the lower

ACCs. Exploiting the profile information of the mixed layer would potentially improve the skill. The absolute ACCs suggest skilful multi-year prediction of salinity over large regions (Fig. D4a,f,k). However, there are fewer places where the initialized dynamical predictions outperform both the persistence forecast (Fig. D4c,h,m) and uninitialized prediction (Fig. D4d,i,n).





**Figure D3: Prediction skill for 0-300 m temperature (T300).** ACC of *hindcast-i1* (a), ΔACC of *hindcast-i1 - assim-i1* (b), *hindcast-i1 - persistence* (c), *hindcast-i1 - historical* (d), and *hindcast-i2 - hindcast-i1* (e) for LY1. Middle and right column show the same but for LY2–5 (f–j) and LY6–9 (k–o). Observations use EN4.2.1 (Good et al., 2013) with coverage 1960–2018. Hatched areas are not locally significant, dotted areas are field significant.





**Figure D4: Prediction skill for 0-300 m salinity (S300). ACC of *hindcast-i1* (a), ΔACC of *hindcast-i1 - assim-i1* (b), *hindcast-i1 - persistence* (c), *hindcast-i1 - historical* (d), and *hindcast-i2 - hindcast-i1* (e) for LY1. Middle and right column show the same but for LY2–5 (f–j) and LY6–9 (k–o). Observations use EN4.2.1 (Good et al., 2013) with coverage 1960–2018. Hatched areas are not locally significant, dotted areas are field significant.**

Skilful decadal prediction of sea level changes has societal value, but the evaluation is complicated by the short satellite record. Nevertheless, the ACC maps for SSH (Fig. D5) reveal that most of the skill comes from the externally forced trend.





The results indicate some benefit from initialisation in the SPNA, but degradation at the inter-gyre region and from there extending into the Nordic Seas. *hindcast-i2* performs better than *hindcast-i1* in the high-latitudes.

**Figure D5: Prediction skill for sea surface height (SSH). ACC of *hindcast-i1* (a), ΔACC of *hindcast-i1 - assim-i1* (b), *hindcast-i1 - persistence* (c), *hindcast-i1 - historical* (d), and *hindcast-i2 - hindcast-i1* (e) for LY1. Middle and right column show the same but for LY2–5 (f–j) and LY6–9 (k–o). Observations use ARMOR-3D (Larnicol et al., 2006) with coverage 1993–2018. Hatched areas are not locally significant, dotted areas are field significant.**



SPNA temperature variability has been linked to meridional ocean heat transport and overturning variability. This relation holds also for the SPNA temperature evolutions in NorCPM1's hindcast (Fig. 13a,d). These can be well approximated from

1435  anomalous northward ocean heat transport into the SPNA region (Fig. D6a,c) and from regressing this transport on AMOC strength (Fig. D6b,d). That the diagnosed temperature changes overestimate the amplitude of the simulated temperature changes is expected, because the surface heat flux acts to dampen the SPNA temperature anomalies in the model.

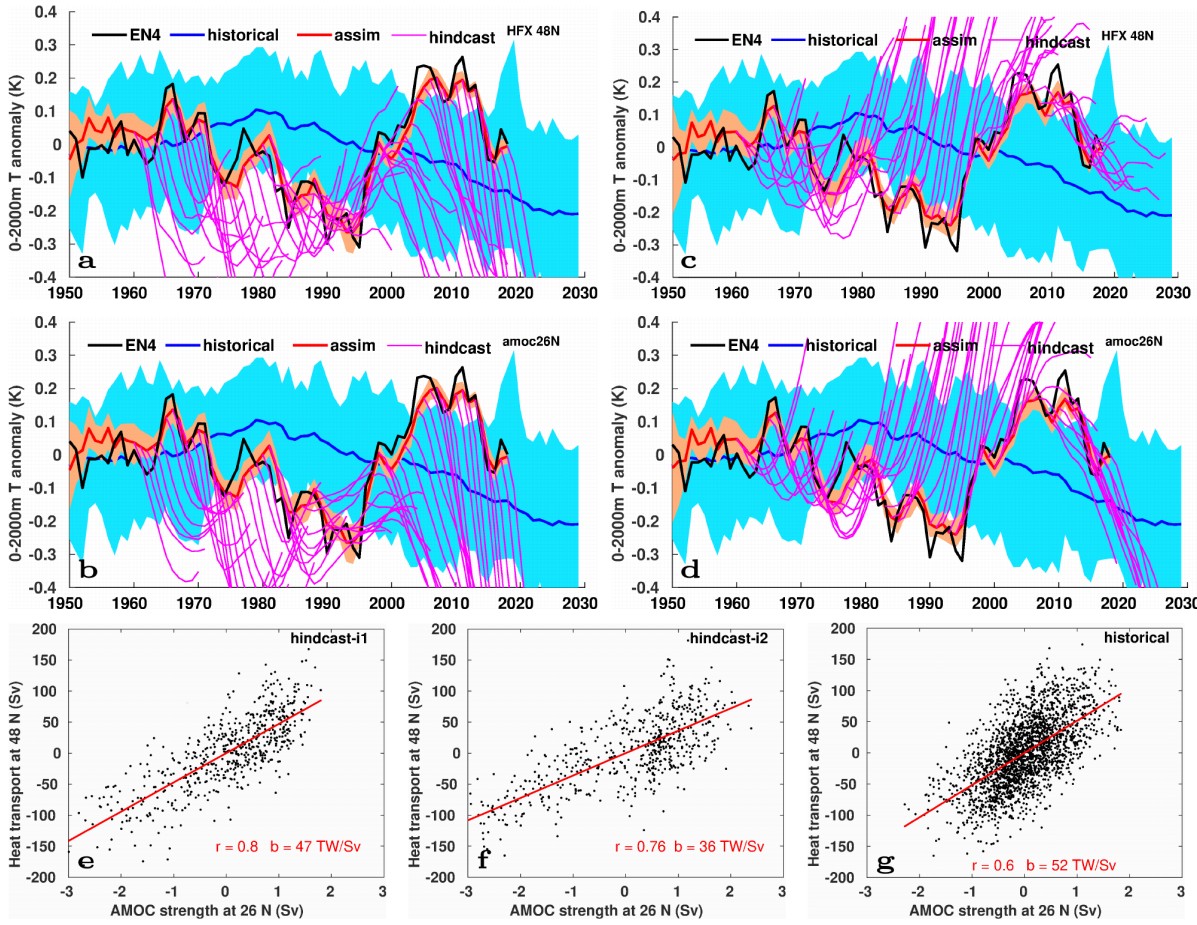

**Figure D6:** SPNA [60-15 °W, 48-65 °N] 0–2000 m temperature (T2000) evolution estimated from anomalous heat transport across

1440  **48 °N (HFX48) (a,c) and from anomalous AMOC strength at 26 °N (MOC26) (b,d) for i1 and i2. Solid lines show ensemble means of *historical* (blue), *assim* (red) and *hindcast* (purple) experiments, with the 1950–2010 average of historical subtracted. Shading denotes ensemble minima and maxima. Estimates shown for *hindcast*, actual temperatures for other experiments. Bottom panels show HFX48 over MOC26 with correlation and regression coefficient from annual values of hindcast-i1 (e) and hindcast-i2 (f) and 5-year averages of historical (g). A factor of 50 TW/Sv has been applied for deriving HFX48 from MOC26 in b,d. The conversion**

1445  **from HFX48 to T2000 tendency disregards temperature changes below 2000 m and heat transport changes through the western-eastern-northern boundaries as well as surface of the SPNA domain.**

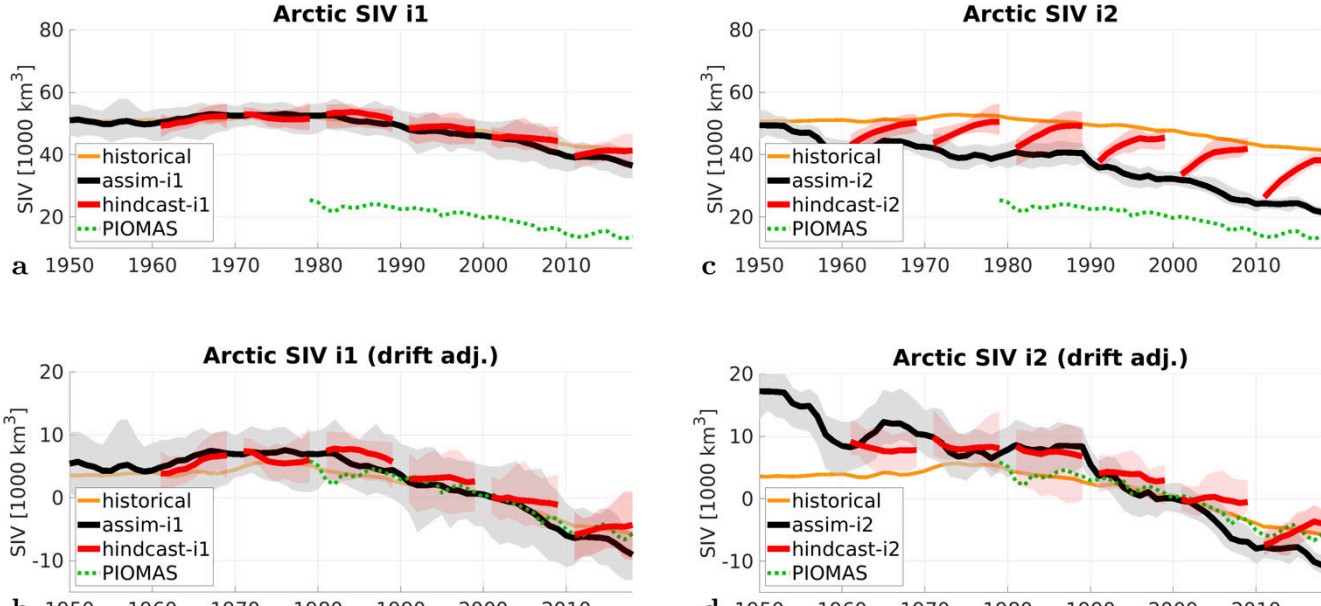

**Figure D7: Time-development of annual Arctic sea ice volume (SIV) of experiment i1 (left) and i2 (right). Black lines are ensemble averages of the respective reanalyses, grey shadings depict the ensemble envelope. Red lines show ensemble averages of the respective decadal hindcasts, red shadings the ensemble envelope, for hindcasts branched in the beginning of each decade. Annual SIV derived from PIOMAS (green) and annual SIV of the ensemble mean of the historical run (orange) are provided as reference. The bottom row shows the drift-adjusted SIV determined as in Yeager et al. (2018) using 1980–2018 as reference period for the adjustment.**

Anomaly initialisation has the advantage that it minimizes forecast drift. The decadal evolutions of the integrated Arctic sea ice volume (Fig. D7) illustrate that the sea ice initialisation for *hindcast-i2* changes the mean state and therefore has a full-field character, causing considerable forecast drift, despite the use of anomaly assimilation. Posteriori drift correction can thus be necessary also for prediction systems that use anomaly assimilation.

*Code availability.* The NorCPM1 code can be downloaded from https://doi.org/10.11582/2021.00014 (Bethke, 2021a) or https://github.com/BjerknesCPU/NorCPM1-CMIP6. The input data needed for running the code can be downloaded from https://doi.org/10.11582/2021.00013 (Bethke, 2021b).

*Data availability.* The CMIP6 output of NorCPM1 is served through the Earth System Grid Federation (ESGF). The output of the CMIP baseline and historical simulations can be accessed at https://doi.org/10.22033/ESGF/CMIP6.10843 (Bethke et al., 2019a) and the output of the DCPP simulations at https://doi.org/10.22033/ESGF/CMIP6.10844 (Bethke et al., 2019b).

*Author contributions.* IB coordinated the writing of this article. YW, IB performed the simulation experiments with support from AG, PC. YW, FC wrote the data assimilation description and physical evaluation; YW produced the figures. FC wrote part of the introduction. FF, AS, JT wrote the biogeochemistry part and evaluation; FF, AS produced the figures. MK produced sea ice analyses and figures. LS contributed to atmospheric evaluation; JV, HL, LP contributed to ocean evaluation. AK, DO, ØS contributed to the implementation of CMIP6 atmospheric forcing, YF to the implementation of CMIP6 land forcing. CG, MB contributed to analysis and visualization of the baseline climate, JV, PC to analysis and visualization of the predictions. NK, TE, MB, JT, FC contributed with project administration and funding acquisition. All co-authors contributed to conceptualization and writing of the article.

*Competing interests.* The authors declare that they have no conflict of interest.

*Acknowledgements.* Computing and storage resources have been provided by UNINETT Sigma2 (nn9039k, ns9039k).

*Financial support.* IB, YW, FC, LP received funding from the Trond Mohn Foundation through the Bjerknes Climate Prediction Unit (BFS2018TMT01). IB received funding from NFR Climate Futures (309562). YW received funding from NFR CoRea (301396). FC received funding from H2020-EU INTAROS (727890). MK received funding from NFR SFE (270733), NordForsk ARCPATH (76654) and H2020-EU.3.5.1 SO-CHIC (821001). FF received funding from NFR Nansen Legacy (276730). MB, JT, AK, DO, ØS, FC and YF received funding from NFR INES (270061).

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
