# Peer review of "NorCPM1 and its contribution to CMIP6 DCPP"

_Geoscientific Model Development, 2021_

## Author Comment (AC1)

Reply to RC1 – Supplement

[Figure]

Figure 3 new: Quality of labels has been improved. Long experiment acronyms are replaced
with shorter ones.

[Figure]

Figure 12 new: Labels are moved outside of panels.

[Figure]

Figure 14 new: a,b,c... labels are moved to lower left corner of panels. Font size of SST/T2000 labels has been reduced in panels e-h.

---

## Author Comment (AC2)

Reply to RC3 – Supplement

[Figure]

Figure D3: Original version showing ACC for 0-300 m ocean temperature.

[Figure]

Figure D3: Alternative version showing MSSS for 0-300 m ocean temperature. The MSSS calculation follows Yeager et al. (2018).

---

## Author Response (AR1)

Dear Editor,

We thank the two reviewers for their valuable comments. We have addressed the comments in the revised version as we outlined in our response to RC1 (https://gmd.copernicus.org/#AC1) and RC3 (https://gmd.copernicus.org/#AC2). Below, we summarize the changes and then give a point-by-point response to all referee comments.

We have added a discussion section (new Section 4) to address general and specific topics that the reviewers brought up. These cover the (1) consideration of atmospheric DA, (2) prospects of improving predictions through ESM versus DA development, (3) use of residual ACC instead of ACC difference as an alternative, more robust method for detecting skill benefit from initialization, (4) use of MSSS as a more robust alternative to ACC. Related to 3 and 4, we present new analyses in the Supplementary Information and have adjusted the text of the main manuscript where necessary.

We have added citations to support our statements at various places. We have clarified and corrected certain parts of the text following the suggestions of the reviewers, e.g.: the updated manuscript states more explicit from the start that our DA does not update the atmospheric component; we note that our definition of "potential predictability" differs from more conventional definitions of potential predictability; we note the apparent discrepancy between the forced skill presented in this paper and findings of a recent multi-model study by Borchert et al. (2021) and offer an explanation.

Following the suggestion of the reviewers, we have tried to make the labelling of the figures more consistent and to improve their quality. As we proposed in our response to RC1, we have moved Appendix C (Earth system model evaluation) and Appendix D (Supporting reanalysis and hindcast evaluation) to a separate Supplementary Information document. This has reduced the overall size of the main paper and allowed us to stay within the GMD journal figure size limit without having to compromise on figure quality.

For convenience of the reviewers, we have combined the main manuscript and Supplementary Information in the diff document but excluded the figures.

Kind regards,
Ingo Bethke (on behalf of the authors)

Point-by-point response to comments from referee 1:

(line numbers in the response refer to the revised manuscript, not to the diff)

Comment 1.1: I take the point that the oceanic DA with an elaborated EnKF scheme carries a lot of the interest to the authors. However, it seems that the combination NorESM+CMIP6 external forcing carries most of the skill in decadal predictions. As we know, this can be seen in other ESMs, too. And of course this usually is a good sign: if the combination ESM + external forcing delivers good results, so that assimilation does not have to repair too many

"shortcomings". To provoke the authors, albeit in a friendly manner: should the prediction community rather invest in better models than in sophisticated assimilation?

Response 1.1: Notwithstanding the emerging importance of external forcing for multiyear climate prediction, we argue that both models and methods to assimilated observations need improvement to provide best possible climate predictions, as has been demonstrated for weather predictions. In the revised manuscript version, we show that initialization benefits can be robustly detected with the residual ACC method proposed by Smith et al. (2019) and that failure of detecting initialization benefits was in part related to the use of ACC differences. We therefore foresee continues benefit from investing in NorCPM's assimilation development, which does not go on expense of the development of NorCPM's ESM component NorESM (which requires very different expertise). In contrary, our aim is that assimilation work can increasingly inform the ESM development to improve aspects of the model relevant to climate prediction. While not addressed in this study, the ESM component of NorCPM is subject to development and upgrades, and the impact of these efforts on climate prediction skill will be documented in future publications. Undoubtedly, improving the ESMs used in climate prediction systems is imperative for achieving more skilful climate predictions and there are limits to the extent more refined assimilation methods can compensate for ESM deficiencies.

Changes 1.1:

Following paragraph has been added to Section 4 in response to the comment:
"While the focus of this study leans towards DA innovations, future skill improvement clearly depends also on improving the ESM component of NorCPM. The dynamical model representation has been demonstrated key to skilful climate prediction (Athanasiadis et al., 2020; Yeager et al., 2018) and recent studies revealed a larger role of external forcing than previously thought (Borchert et al., 2021; Klavans et al., 2021; Liguori et al., 2020). The skill benefit from DA-assisted initialization does not only relate to synchronisation of internal climate variability, but also to correcting the externally forced climate signal at forecast initialization time—which is subject model and forcing errors. We nevertheless expect continuous need for, and benefit from improving NorCPM's assimilation, along with improving its ESM component. We have seen from weather and seasonal forecasting how improvements in both models and methods to assimilate observations (as well as observations and computing power) have continued to lead to enhanced prediction skill (Bauer et al., 2015). Work has started to upgrade NorCPM's ESM component to NorESM2-MM (Seland et al., 2020)—featuring improved physical process parameterizations, a higher atmospheric resolution, a more realistic AMOC, and overall reduced climate biases compared to NorESM1—and results of this effort will be documented in future publications. We envision that the climate prediction evaluation and DA can increasingly inform the development of NorESM, which traditionally focused on long-term climate projections. There is growing evidence that current generation climate models systematically underestimate the influence of SST variations and external forcing variability on extratropical atmospheric variability, particularly related to the North Atlantic Oscillation (e.g., Scaife and Smith, 2018; Athanasiadis et al., 2020). While post-processing methods relying on large ensembles have been proposed to mitigate this shortcoming (Smith et al.

2020), improving this aspect in the next model generation should be a key priority for the prediction community." (L1072-1089)

Comment 1.2: The authors rely on the ocean-atmosphere coupling to transfer observational information to the atmosphere. I am totally fine with this, especially in the view of multi-annual predictions. Nevertheless, what do the authors think about having an atmospheric assimilation as well, at least for the large scale atmospheric state? I would also like the authors to be very clear from the beginning that the EnKF is applied for oceanic/sea ice DA and IS NOT applied in atmospheric assimilation here. This point could be made in l.94 "NorCPM1 further stands out in that it uses an EnKF based anomaly DA scheme..."

Response 1.2: In the revised manuscript, we are clearer from the beginning that the EnKF updates are not applied to the atmosphere and land states. We also include a short discussion on the implications of currently not having atmospheric assimilation and future plans regarding adding it.

Changes 1.2:

As the reviewer proposed, we modified the sentence L99-101 of the introduction in the revised manuscript: "NorCPM1 further stands out in that it uses an Ensemble Kalman Filter (EnKF; Evensen, 2003) based anomaly DA scheme that updates unobserved variables in the ocean and sea ice components (currently DA update is not applied to atmosphere and land)"

The revised manuscript is also clearer that our strongly coupled data assimilation (SCDA) is limited to the ocean and sea ice components L321-322: " To avoid confusion with atmosphere-ocean SCDA (e.g., Penny et al., 2019), we will refer the assim-i2 approach as OSI-SCDA (where OSI stands for "ocean–sea ice")."

We have added a discussion paragraph on plans for assimilating atmospheric observations into NorCPM and expected benefits (L1044-1061 in new Section 4):
"The anomaly assimilation scheme of NorCPM1 currently updates only the ocean and sea ice components, and the atmosphere and land components are only constrained to the extent that they are affected by the surface conditions. Utilizing atmospheric observations and better constraining the atmospheric circulation variability has potential to improve the ocean and sea ice initialization by producing surface fluxes that are more consistent with the SST and SIC anomalies during the assimilation phase. Constraining the atmospheric circulation will also improve atmosphere and land initialization, beneficial for sub-seasonal to seasonal prediction. The success of utilizing initial conditions from forced ocean-sea ice simulations (Yeager et al., 2018) demonstrates the potential in constraining surface fluxes over ocean and sea ice for initializing multiyear climate predictions. Performing EnKF ocean–sea ice assimilation in addition to constraining the atmospheric variability is expected to further improve the predictions (Polkova et al., 2019). Utilization of atmospheric observations in NorCPM's initialization is work in progress. A unified EnKF-based assimilation scheme covering all ESM component would be desirable but is subject to numerous technical and scientific challenges. As an intermediate solution, we are exploring

atmospheric nudging in combination with EnKF-based ocean–sea ice assimilation in NorCPM, a strategy that has been successfully applied in the MPI MiKlip system (Polkova et al., 2019). We will take advantage of the availability of multiple simulation members of the reanalysis products like ERA5 (Hersbach et al., 2020) and CERA-20C (Laloyaux et al., 2018) and nudge the members of the NorCPM analysis to individual members of the reanalysis products to provide a representation of atmospheric observational uncertainties and help generate ensemble spread in the ocean state. We will complement the atmospheric nudging with the leading average cross covariance technique that has been shown to further improve ocean initialization by performing a one-way (from atmosphere to ocean) strongly coupled data assimilation (Lu et al., 2015)."

Comment 1.3: The authors include a lot of figures, and I like this very much. However, the quality of the figure annotations (labels) is sometimes rather poor. I would like to ask the authors to re-assess the annotations, this would greatly help the reader to quickly connect with the figures.

Response 1.3: We have tried to improve the annotations of the figures, including the quality and position of labels. We noticed that the quality degrades when we export from word format to pdf, something that will be less of an issue for the published paper (since the figures are provided separately). For future publications, we will investigate ways to further improve the quality of annotations.

Changes 1.3: We have updated the figures 2-8, 10-15, 19-20, S10-14, S16-19 in the revised manuscript and the Supplementary Information.

Comment 1.4: figures with maps are at the limit in terms of crowded information, but that is still okay.

Response 1.4: We prefer to keep the panel layout of the figures as is. The multi-panel layout allows the reader to visually compare the results for different lead years, prediction benchmarks and fields of interest. Where possible, we tried to use a similar style for the figures with maps, mostly adopted from Yeager et al. (2018). The reader may need to spend some time to understand the first of such figures, but it should require less time to understand successive figures of the same style.

Changes 1.4: -

Comment 1.5: although the maps themselves are in hires, their annotations sometimes look very lowres, e.g. as in Fig. 3

Response & changes 1.5: We have tried to improve the annotations and also converted all maps from vector to bitmap format to avoid rendering issues. Some quality degradation occurred when converting from word format to pdf. The original figure files that will be provided to the journal with the final manuscript version have a higher quality.

Comment 1.6: please put the annotations outside the maps according to rows and columns, similar to Fig.3 (Figs. 7,9-12,15-17,21-24,D3-5)

Response 1.6: Done.

Changes 1.6: Annotations moved outside of maps (Figs. 6, 8, 10, 11, 14-15, 19-20, S10-12, S14, S16-19).

Comment 1.7: huge difference in font size, Fig. 14

Response 1.7: We have reduced the font size of the title labels in panels e-f.

Changes 1.7: Updated Figure 13.

Comment 1.8: Would be good to have all the a,b,c... at a rather similar position throughout all figures. Now they are sometimes in the upper left, lower left, or somewhere within the figure.

Response & changes 1.8: a,b,c... labels moved to the lower left corner in all figures.

Comment 1.9: l.98 "see Section 2.1.1 for details" Is this reference meant for the description of the DA? Then section 2.2 (or subsections) would fit better.

Response & changes 1.9: Text correct to "see Section 2.2.3 for details" (L104).

Point-by-point response to comments from referee 2:

(line numbers in the response always refer to the revised manuscript, not to the diff)

Comment 2.1: I only have one suggestion to the authors that warrants mentioning outside the "specific comments" section. The comparison of skill between the hindcasts and/or assimilation simulations and the historical runs to assess the contribution of the forced response to skill currently relies on skill differences (i.e. ACC(hind)-ACC(hist) and so on). Recently, Smith et al. (2019) demonstrated that this practice does not fully capture the benefit of initialisation due to non-linear interactions in the system, and propose the use of residual ACC, that subtracts the forced signal (hist) from the hindcast signal prior to skill calculation. In their work, Smith et al. show that the use of residuals more accurately describes what the authors set out to assess here. I suggest the authors either present some results using residuals to illustrate this alternative metric, or at least discuss the residual as an alternative (and potentially superior) approach.

Response 2.1: We have followed the reviewer's suggestion and present some results using the residuals to illustrate this alternative metric (we selected the field surface temperature as a candidate because the strong anthropogenic trend makes detection of initialization benefit for temperature particularly challenging). We also added a discussion and modified our conclusions in the light of the new analysis.

Changes 2.1:

In the abstract, we extended the sentence L33-35 to note robust multiyear initialization benefit over land: "External forcings are the primary source of multiyear skills, while added benefit from initialization is demonstrated for the subpolar North Atlantic (SPNA) and its extension to the Arctic, and also for temperature over land if the forced signal is removed."

In Section 3.2.5, we extended the text at L948-950: "For LY2–5 and LY6–9, the difference maps indicate little initialization benefit, implying that most multiyear SAT skill over land stems from the externally forced trend in NorCPM1. However, this result can be sensitive to the ΔACC metric (Fig. S18 and related discussion in Section 4)."

In Section 4, we added following discussion at L1062-1071: "NorCPM1 shows overall high multiyear prediction skill from external forcing, with a modest and regionally limited increase in skill from improving the initial conditions via DA. A caveat with using ACC differences for detecting initialization benefit is that if the absolute ACCs are large, the ACC differences become difficult to robustly detect. Smith et al. (2019) proposed a more robust quantification method for initialization benefit, where the forced signal of the model is regressed out of both the model and observation data and ACCs are computed from the residuals (the result is scaled to account for the smaller variance of the residuals, see Section S2 for more details). Figure S18 compares both methods, with the residual ACCs showing clear initialization benefit for SAT over land regions where ΔACCs are statistically indistinguishable from zero. Like in Yeager et al. (2018), we use ΔACCs in this study to systematically compare against multiple benchmarks. The use of residual ACCs should, however, be of interest for future work, especially for assessing the impact of DA developments on forecast skill."

We have added a new Fig. S18 that compares the results from the two methods for SAT. In Section S2, we briefly describe the residual method and results from it L223-230: "The use of ACC differences for detecting benefit from initialization can be problematic if the skill from the externally forced climate trend is very high and the ACC differences are small (but can nevertheless reflect a significant change in explained variance). Smith et al. (2019) proposed a more robust way of evaluating initialization benefit by regressing out the forced signal of the model—estimated from the ensemble mean of a historical simulation experiment with the same model—from both model and observation output and then computing the ACC of the residuals. The result is scaled with the standard-deviation of the residuals of the observations divided by the standard-deviation of the observations, and an estimated spurious correlation bias is subtracted. Figure S18 compares the ΔACC versus the residual ACC results obtained for SAT. Initialization benefit over land not shown by the ΔACCs is robustly detected by the residual ACCs."

Comment 2.2: l. 24-25 I do not think it is clear at this point what "non-assimilation experiments" are. Maybe just call them "NorESM1 simulations" for clarity?

Response & changes 2.2: We have followed the suggestion and rephrased "non-assimilation experiments" to "NorESM1 simulations" (L24).

Comment 2.3: ll. 54-55 This statement could benefit from one or two publications to back up the claim.

Response 2.3:

We have added the below citations.

Årthun, M., E. W. Kolstad, T. Eldevik, and N. S. Keenlyside (2018), Time Scales and Sources of European Temperature Variability, Geophys Res Lett, 45 (0), doi:10.1002/2018GL077401.

Athanasiadis, P. J., S. Yeager, Y.-O. Kwon, A. Bellucci, D. W. Smith, and S. Tibaldi (2020), Decadal predictability of North Atlantic blocking and the NAO, npj Climate and Atmospheric Science, 3 (1), 20, 10.1038/s41612-020-0120-6.

Sutton, R. T., , and D. L. R. Hodson, 2005: Atlantic Ocean forcing of North American and European summer climate. Science, 309 , 115–118.

Omrani, N. E., N. S. Keenlyside, J. Bader, and E. Manzini (2014), Stratosphere key for wintertime atmospheric response to warm Atlantic decadal conditions, Climate Dynamics, 42 (3-4), 649-663, 10.1007/s00382-013-1860-3.

Changes 2.3: L57-58 "Prediction skill in the ocean gives rise to skill in the atmosphere and over land by affecting the atmospheric circulation or atmospheric transport of anomalous heat and moisture (Årthun et al., 2018; Athanasiadis et al., 2020; Omrani et al., 2014; Sutton and Hodson, 2005)."

Comment 2.4: ll. 63-74 Similarly, this entire paragraph can in my opinion not go without citations. Please add some published work to back up the statements made here.

Response & changes 2.4:

We have added citations to the paragraph L66-79 as indicated below.

Current climate prediction systems are thought to not fully realise the predictive potential on multi-year times scales, although the practical limits of predictability themselves and their regional variations are poorly known (Branstator et al., 2012; Sanchez-Gomez et al., 2015; Smith et al., 2020).

The skill of climate prediction depends on the initialisation of internal climate variability state, the representation of the dynamics and processes that lead to predictability, and the representation of the climate responses to external forcings (Branstator and Teng, 2010; Latif and Keenlyside, 2011; Bellucci et al., 2015; Yeager and Robson, 2017).

Dynamical climate prediction systems typically use Earth system models (initially developed to provide uninitialised long-term climate projections) for representing the dynamics and the responses to external forcings (Meehl et al., 2009; Meehl et al., 2013).

Importantly, the dynamical prediction systems add initialisation capability to the ESMs, adopting a wide range of initialisation strategies (see Section 2.2.1) (Meehl et al., 2021). A better understanding of the three aspects – initialisation, model dynamics, forcing response – is fundamental for better exploiting the climate predictive potential and improving estimates of climate predictability (Keenlyside and Ba, 2010; Cassou et al., 2018; Verfaillie et al., 2021).

The existing climate prediction systems undersample effects of model and initialisation uncertainty and are not necessarily well suited to address questions related to changes in the observing system. The benefits from using advanced data assimilation for initialisation, especially in an ocean density coordinate framework, are not well explored.

Bellucci, A., et al. (2015), Advancements in decadal climate predictability: The role of nonoceanic drivers, Rev Geophys, 53, 165–202, 10.1002/2014RG000473.

Branstator, G., and H. Y. Teng (2010), Two Limits of Initial-Value Decadal Predictability in a CGCM, J Climate, 23(23), 6292-6311

Branstator, G., H. Y. Teng, G. A. Meehl, M. Kimoto, J. R. Knight, M. Latif, and A. Rosati (2012), Systematic Estimates of Initial-Value Decadal Predictability for Six AOGCMs, J Climate, 25(6), 1827-1846

Cassou, C., Y. Kushnir, E. Hawkins, A. Pirani, F. Kucharski, I.-S. Kang, and N. Caltabiano (2018), Decadal Climate Variability and Predictability: Challenges and Opportunities, B Am Meteorol Soc, 99(3), 479-490, 10.1175/BAMS-D-16-0286.1.

Keenlyside, N. S., and J. Ba (2010), Prospects for decadal climate prediction, Wiley Interdisciplinary Reviews: Climate Change, 1(5), 627-635, 10.1002/wcc.69.

Latif, M., and N. S. Keenlyside (2011), A perspective on decadal climate variability and predictability, Deep Sea Research Part II: Topical Studies in Oceanography, 58, 1880-1894

Meehl, G. A., et al. (2009), Decadal Prediction Can It Be Skillful?, B Am Meteorol Soc, 90(10), 1467-1486, https://doi.org/10.1175/2009BAMS2778.1.

Meehl, G. A., et al. (2013), Decadal Climate Prediction: An Update from the Trenches, B Am Meteorol Soc, 10.1175/bams-d-12-00241.1.

Meehl, G. A., et al. (2021), Initialized Earth System prediction from subseasonal to decadal timescales, Nature Reviews Earth & Environment, 2(5), 340-357, 10.1038/s43017-021-00155-x.

Sanchez-Gomez, E., C. Cassou, Y. Ruprich-Robert, E. Fernandez, and L. Terray (2015), Drift dynamics in a coupled model initialized for decadal forecasts, Climate Dynamics, 1-22, 10.1007/s00382-015-2678-y.

Smith, D. M., et al. (2020), North Atlantic climate far more predictable than models imply, Nature, 583(7818), 796-800, 10.1038/s41586-020-2525-0.

Verfaillie, D., F. J. Doblas-Reyes, M. G. Donat, N. Pérez-Zanón, B. Solaraju-Murali, V. Torralba, and S. Wild (2021), How Reliable Are Decadal Climate Predictions of Near-Surface Air Temperature?, J Climate, 34(2), 697-713, 10.1175/JCLI-D-20-0138.1.

Yeager, S. G., and J. I. Robson (2017), Recent Progress in Understanding and Predicting Atlantic Decadal Climate Variability, Current Climate Change Reports, 3(2), 112-127, 10.1007/s40641-017-0064-z.

Comment 2.5: l. 80-81 Take it or leave it: to me, the order historical->assimilation->hindcasts would make more sense.

Response & changes 2.5: We have rephrased it to "two sets of DCPP coupled reanalysis simulations, and two sets of initialized DCPP hindcast simulations that obtain their initial conditions from the two reanalysis sets." (L85-86).

Comment 2.6: l. 131, 134 Naïve question: is NorCPM1 a purely "physical model"?

Response & changes 2.6: Because we realized that the original formulation "physical model" was inaccurate, we have rephrased the formulation to "the ESM component" (L130).

Comment 2.7: l. 302-304 Why the two different baseline periods? This might be obvious, but I am struggling to understand the motivation for this choice.

Response 2.7:

We used two periods primarily because OISST data were not available prior September 1981.

In hindsight, it would have been more consistent if we had extended the OISST back in time with HadISST2 data (like we did for assim-i2) and then used the period 1980-2010. But we checked and found the 1982-2010 OISST-based climatology and the 1980-2010 HadISST2/OISST-based climatology (1980-1981 HadISST, 1982-2010 OISST) are practically

indistinguishable. It is also possible that our computation of climatologies was overly complicated, but we are confident that these choices have only a minor impact.

Changes 2.7: Modified sentence L308-309 for clarification: "... but over the period 1982-2010 when assimilating OISSTV2 data (i.e., beyond 2010) because OISSTV2 was not available before 1982."

Comment 2.8: l. 316 I suppose "The approach" here references the SCDA approach used in assim -i2?

Response 2.8: This is correct.

Changes 2.9: We modified the text to make it clearer L319-324: " In assim-i2, we allow the oceanic observations to update the ocean and the sea ice components. In this case the system is a strongly coupled DA system (SCDA), where the oceanic observations influence the sea ice component of the system both at the DA step and during the model integration. To avoid confusion with atmosphere-ocean SCDA (e.g., Penny et al., 2019), we will refer the assim-i2 approach as OSI-SCDA (where OSI stands for "ocean–sea ice"). The OSI-SCDA assures a more consistent initialization across components and exploits the longer temporal coverage of oceanic observations relative to sea ice observations (see also Appendix A)."

Comment 2.9: ll. 414-416 ACC is known to be sensitive to spurious trends as a skill metric (e.g. Smith et al., 2019). RMSE-based metrics such as MSSS might be more realistic when it comes to skill assessment. While I see why the authors choose ACC for this study, I think a short sentence on potential ACC shortcomings might be in order here or in the discussion.

Response & changes 2.9:

We have added discussion on the use of ACC vs MSSS and added Fig. S19 to provide an example.

In L421, we have added "(for details and discussion of the skill metrics see Appendix B and Section 4)".

In Section 4 (L1094-1099), we have added "The ACC, our primary metric for quantifying skill in this study, is sensitive to random correlation that can occur over the evaluation period as it does not penalize for amplitude errors. The Mean Square Skill Score (MSSS), that penalizes amplitude errors, can be used as an alternative, potentially more robust metric (Goddard et al., 2013 and Section S2). As we found the MSSS results (Fig. S19) comparable to the ACC results (Fig. S10), we decided to use ACC to facilitate comparison with previous works (e.g., Yeager et al., 2018) and because amplitude errors stemming from the model underestimating the forced climate signal can to some extent be corrected posteriori (Smith et al., 2019; Smith et al., 2020).".

In Section S2, we have added "The Mean Square Skill Score (MSSS; Goddard et al., 2013) is an alternative metric to the ACC and computed as 1 - MSE/MSE$_{ref}$, where MSE$_{ref}$ is the Mean Square Error (MSE) of a reference prediction (e.g., climatology, persistence forecast, historical simulation, or other). Unlike the ACC, the MSSS penalizes amplitude errors and has a range from -$\infty$ to 1, with positive values indicating that the predictions outperform the reference prediction. The MSSS results for T300 (Fig. S19) are qualitatively similar to the ACC results (Fig. S3)." (L238-242)

Comment 2.10: Figure 2 Are these global metrics?

Response 2.10: The metrics shown in Figure 2 are global, considering both vertical and horizontal dimensions. They are computed as detailed in Equations 1–6 in Section 3.1.1.

Changes 2.10: For clarification, we have changed the caption of Figure 2 to "Global assimilation statistics (see Section 3.1.1. for definitions)...". We also corrected the numbering of the Equations from 2–7 to 1–6.

Comment 2.11: l. 465 There is a "." Missing after "observations".

Response & changes 2.11: Added dot at end of sentence (L472).

Comment 2.12: l. 507 Is this referencing "changes" due to data assimilation? I suggest being explicit about that.

Response & changes 2.12: We have made it clearer that this paragraph continues to discuss the impact of the assimilation on the mean state and have rephrased the text to "Some impact of DA on the mean state of assim-i1 is also seen..." (L522).

Comment 2.13: ll. 532-533 I am struggling to see the smaller improvement for S300 compared to T300 reflected in figure 7. I suppose the authors reference figures 7f and 7g here, comparing the assimilation simulations' improvement relative to historical simulations for T300 and S300? When I visually compare 7f to 7g, I am struggling to make out a clear "winner". If anything, it appears to me that S300 shows slightly higher values. Could you please comment on that? In general, the text could at times benefit from more clear reference of figure sub-panels when making statements in the text.

Response 2.13:

We made a mistake in our first version of Figure 7, resulting in smaller ACCs for S300. Later, we noticed and rectified this but forgot to correct the manuscript text.

We have removed the sentence "The improvements for S300 are smaller ...".

Also, we have tried to refer more explicitly to sub-panels in the revised version of the manuscript.

Changes 2.13: Corrected sentence " The improvements for S300 are similarly high and largest in the Arctic, albeit showing localised degradation in some coastal regions." (L542-544)

Comment 2.14: Figure 7 The labelling of assim-i1 and assim-i2 as ANA1 and ANA2 in the figures is not optimal, as it is inconsistent with the labelling in the text. I suggest changing the labels in the figures for consistency. This is an issue in all figures that show skill comparison including assimilation simulations.

Response & changes 2.14: The revised manuscript now consistently uses the same acronyms (historical, assim-i1, hindcast-i1 etc) in all figure labels.

Comment 2.15: l. 559 ff. This discussion is reminiscent of work done by Koul et al. (2020) on which SPG index represents which underlying physical processes. I think this part could be shortened by referring to the above mentioned paper.

Response 2.15: The text in question was not intended as discussion but rather to provide some minimum motivation and background for the readers less familiar with the North Atlantic subpolar gyre concept. We therefore prefer to keep the text but have added the citation to Koul et al. (2020) for discussing the limits of our simple index definition.

Changes 2.15: Added sentence "We note that more elaborated index definitions based on principle component analysis of SSH and subsurface density are likely to capture circulation features and associated water mass variability better than our simple index (Koul et al, 2020)." (L570-572)

Comment 2.16: l. 608-610 I think it is important to point out this "small contribution" is insignificant (is it?).

Response 2.16:

We have computed a p-value of 0.085 using block-bootstrapping—with block lengths of 5 years—and re-sampling of members in the computation of the ensemble mean (see Appendix B for details on p-value computation).

Following Yeager et al. (2018), we have used a 10 % significance level throughout the manuscript and it would be inconsistent if we switched to 5 % significance level for only this test. Results presented in Section S2 in Figure S9 are also suggestive for a small, but systematic influence from external forcing on simulated NINO34 SSTs, with positive correlation values for all calendar months.

Changes 2.16:

We have moderated the statement to " The ensemble mean of historical has a smaller amplitude and is only marginally correlated with the observed index (r=0.2, p=0.085, alpha=0.1), suggesting a potential small contribution from external forcing." (L619-621)

Comment 2.17: l. 676-678 At least one citation should be given for the statement on internal vs external causes of the global warming hiatus (e.g. Medhaug et al. 2017?).

Response & changes 2.17: We have cited Medhaug et al. (2017) in the revised manuscript L685-687: "For global scale SAT synchronisation, the global warming hiatus at the beginning of the 21st Century, which has been attributed to both internal variability and external forcing (e.g., Medhaug et al., 2017), makes an interesting test case."

Comment 2.18: l. 696-698 I assume the authors used ACC**2 to calculate explained variance? This should be made explicit.

Response & changes 2.18: We have rephrased the sentence to "One should note that a change in correlation from 0.6 to 0.9 equates to more than doubling in explained variance from 36 % to 81 % (estimated by the square of the correlation)" (L705-707)

Comment 2.19: ll. 741-754 In a recent paper, Borchert et al. (2021) showed increased contribution of forcing to decadal SPNA SST prediction skill in CMIP6 compared to CMIP5, using a multi-model ensemble including NorCPM1. How do the authors square their findings presented here with what Borchert et al. (2021) found?

Response & changes 2.19:
In the revised manuscript, we point out the apparent discrepancy to the findings of Borchert et al. (2021): "This result stands in contrast to multi-model findings (that include NorCPM1) suggesting a positive contribution of the forced signal to SPNA temperature skill over a comparable period (Borchert et al., 2021). We suspect a problem with CMIP6 land use change specification (Fig. 13c and text in Section S1), leading to an unrealistic historical cooling trend over North America in NorCPM1. Via downstream effects, the continental cooling (likely an artifact) may contribute to the SPNA cooling trend shown after 1980, exacerbating the discrepancy between the observed and simulated SPNA temperature evolutions." (L752-757)

Comment 2.20: l. 825 To me, the phrase "potential predictability" refers to the skill of simulations initialised from a piControl simulation with respect to the same piControl simulation. What the authors demonstrate here (skill of hindcasts initialised from a reanalysis-type simulation with respect to a reanalysis that did not directly assimilate biogeochemistry, but produces biogeochemistry that is consistent with observed physical

climate) is to me more than that. The authors might want to re-consider their phrasing here so as to avoid underselling their findings.

Response 2.20:

As the manuscript text states, we adopted the definition of "potential predictability" from Yeager et al. (2018), a key reference for our study.

We agree that our use of "potential predictability" differs from the one in some earlier studies that utilized a piControl simulation (e.g., Collins et al., 2006) and we more clearly point this out in the revised version.

Given the large disparity between our PP skill estimated against real observations and the potential PP skill estimated against assim-i1, it does not seem likely that our potential predictability realistically represents real-world predictability (even if considering observational uncertainties and short temporal coverages). We therefore do not think we are underselling our potential predictability results and are rather cautious about generalizing them. More work is needed to understand and resolve this issue.

Collins, M., Botzet, M., Carril, A. F., Drange, H., Jouzeau, A., Latif, M., Masina, S., Otteraa, O. H., Pohlmann, H., Sorteberg, A., Sutton, R., & Terray, L. (2006). Interannual to Decadal Climate Predictability in the North Atlantic: A Multimodel-Ensemble Study, Journal of Climate, 19(7), 1195-1203.

Changes 2.20: Added sentence "We will refer to this as the potential predictability*, using the asterisk to indicate that it differs from more conventional potential predictability estimates based on self-prediction that typically utilize a preindustrial control simulation (e.g., Collins et al., 2006)." (L836-839)

Comment 2.21: l 888 It should be "in the Pacific sector".

Response & changes 2.21: We have added the article in L892.

Comment 2.22: Figs. 25 & 26 The comparison of mean skill and skill of the mean signal is interesting. However, the authors only mention Fig. 26 briefly in one paragraph. Would 1-2 more sentences on this topic be interesting to a wider readership?

Response 2.22: We have added two sentences on the comparison between the two metrics. However, we prefer not to overly speculate as we currently do not fully understand the differences in the results.

Changes 2.22:

Added sentences "While the initialized hindcasts performed equal or better than historical for globally averaged skill of local SST, T300, and SAT (Fig. 21a,b,g), hindcast-i1 and hindcasti2 show slightly poorer multiyear skill than historical in their global means (Fig. 22a,b,g). Except for SST, the reanalyses mostly outperform both persistence and historical but not as clearly as for the globally averaged skill." (L1032-1035)

Added sentence "Why exactly the globally averaged grid-cell skills (Fig. 21) show more benefit from DA than the skills of the global means (Fig. 22) is something that warrants further investigation." (L1038-1039)

Comment 2.23: l 1253 ff I particularly like this (brief) mention of observational error, particularly in light of the small but important differences between assim-i1 and assim-i2. This might be a candidate for the main text, as this info is currently a little hidden.

Response & changes 2.23: We have moved the text in question from Appendix B to our new discussion section (Section 4)
"The significance testing used in this study (Appendix B) does not account for observational error. Nowadays, observational reanalyses routinely provide ensemble products that span observational uncertainty. While beyond the scope of this study, future skill evaluations should explore ways of utilizing this ensemble information in local and field significance testing. The addition of observational uncertainty should generally lower the p-values, leading to stricter testing." (L1090-1093)

Comment 2.24: l 1281 As far as I can make out, the mean state of AMOC is not shown in Figure C3c, nor is the vigorous nature of simulated AMOC. This might be a phrasing issue, but I was looking for this information in the figure. Rephrase to avoid this in the future?

Response 2.24:

Figure C3c shows the climatological Atlantic meridional overturning circulation (AMOC) streamfunction in units Sv for NorCPM1. To our knowledge, this is the standard way for characterising the mean state of AMOC. Values in excess of 30 Sv clearly indicate the vigorous nature of the simulated AMOC ("vigorous" in the sense of strong, not necessarily variable). While this is somewhat different from the "time-mean AMOC strength evaluated at a fixed latitude", these two characterizations are closely related.

The overly strong AMOC has been a longstanding, well-documented problem of previous NorESM versions (e.g., Bentsen et al., 2013, Cheng et al., 2013) but has been mitigated through updates and re-tuning of the ocean code in more recent NorESM versions (Guo et al., 2019; Seland et al., 2020).

Bentsen, M., Bethke, I., Debernard, J. B., Iversen, T., Kirkevåg, A., Seland, Ø., Drange, H., Roelandt, C., Seierstad, I. A., Hoose, C., and Kristjánsson, J. E.: The Norwegian Earth System Model, NorESM1-M – Part 1: Description and basic evaluation of the physical climate, Geosci. Model Dev., 6, 687-720, https://doi.org/10.5194/gmd-6-687-2013, 2013.

Cheng, W., Chiang, J. C. H., & Zhang, D. (2013). Atlantic Meridional Overturning Circulation (AMOC) in CMIP5 Models: RCP and Historical Simulations, Journal of Climate, 26(18), 7187-7197.

Guo, C., Bentsen, M., Bethke, I., Ilicak, M., Tjiputra, J., Toniazzo, T., Schwinger, J., and Otterå, O. H.: Description and evaluation of NorESM1-F: a fast version of the Norwegian Earth System Model (NorESM), Geosci. Model Dev., 12, 343–362, https://doi.org/10.5194/gmd-12-343-2019, 2019.

Seland, Ø., Bentsen, M., Olivié, D., Toniazzo, T., Gjermundsen, A., Graff, L. S., Debernard, J. B., Gupta, A. K., He, Y.-C., Kirkevåg, A., Schwinger, J., Tjiputra, J., Aas, K. S., Bethke, I., Fan, Y., Griesfeller, J., Grini, A., Guo, C., Ilicak, M., Karset, I. H. H., Landgren, O., Liakka, J., Moseid, K. O., Nummelin, A., Spensberger, C., Tang, H., Zhang, Z., Heinze, C., Iversen, T., and Schulz, M.: Overview of the Norwegian Earth System Model (NorESM2) and key climate response of CMIP6 DECK, historical, and scenario simulations, Geosci. Model Dev., 13, 6165–6200, https://doi.org/10.5194/gmd-13-6165-2020, 2020.

Changes 2.25: We have rephrased the caption text of Fig. S3 from "Atlantic meridional overturning streamfunction" to " Atlantic meridional overturning circulation (AMOC) streamfunction" to be more precise.

---

## Author Response (AR2)

Dear Editor,

As we did not receive any further referee/Editor comments, we have uploaded the manuscript version from the last iteration without applying further changes.

Kind regards,
Ingo Bethke